# VANISHING GRADIENTS IN REINFORCEMENT FINETUNING OF LANGUAGE MODELS

**Noam Razin**[*‡], **Hattie Zhou**[*§], **Omid Saremi**[†], **Vimal Thilak**[†], **Arwen Bradley**[†],

**Preetum Nakkiran**[†], **Joshua Susskind**[†], **Etai Littwin**[†]

[†]Apple  [‡]Tel Aviv University  [§]Mila, Université de Montréal

## ABSTRACT

Pretrained language models are commonly aligned with human preferences and downstream tasks via *reinforcement finetuning (RFT)*, which refers to maximizing a (possibly learned) reward function using policy gradient algorithms. This work identifies a fundamental optimization obstacle in RFT: we prove that the expected gradient for an input vanishes when its reward standard deviation under the model is small, even if the expected reward is far from optimal. Through experiments on an RFT benchmark and controlled environments, as well as a theoretical analysis, we then demonstrate that vanishing gradients due to small reward standard deviation are prevalent and detrimental, leading to extremely slow reward maximization. Lastly, we explore ways to overcome vanishing gradients in RFT. We find the common practice of an initial *supervised finetuning (SFT)* phase to be the most promising candidate, which sheds light on its importance in an RFT pipeline. Moreover, we show that a relatively small number of SFT optimization steps on as few as $1\%$ of the input samples can suffice, indicating that the initial SFT phase need not be expensive in terms of compute and data labeling efforts. Overall, our results emphasize that being mindful for inputs whose expected gradient vanishes, as measured by the reward standard deviation, is crucial for successful execution of RFT.[1]

## 1 INTRODUCTION

The dominant machine learning paradigm for textual data relies on pretraining a language model on vast corpora, and subsequently aligning it with human preferences and downstream tasks via finetuning (see, *e.g.*, Yang et al. (2019); Raffel et al. (2020); Wei et al. (2022); Thoppilan et al. (2022); OpenAI (2023); Touvron et al. (2023); Zhao et al. (2023)). *Supervised finetuning (SFT)* provides a straightforward procedure for adapting the model, in which the cross entropy loss is minimized over pairs of textual inputs and desirable outputs. Despite the simplicity of SFT, the cost of obtaining high-quality labeled data, as well as the difficulty in expressing human preferences through ground truth labels, has led to wide adoption of a reinforcement learning-based approach, herein referred to as *reinforcement finetuning (RFT)* (Wu and Hu, 2018; Ziegler et al., 2019; Stiennon et al., 2020; Ouyang et al., 2022; Bai et al., 2022; OpenAI, 2023; Touvron et al., 2023).

In RFT, instead of minimizing a loss over labeled inputs, policy gradient algorithms are used for maximizing a reward function. The reward function can be learned from human preferences, in which case RFT is known as *reinforcement learning from human feedback (RLHF)*, or tailored to a downstream task. Specifically, denote by $V(\mathbf{x}; \theta)$ the expected reward that a model parameterized by $\theta$ achieves, given as input a sequence of tokens $\mathbf{x}$. For a distribution over inputs, RFT maximizes $\mathbb{E}_{\mathbf{x}}[V(\mathbf{x}; \theta)]$ with respect to $\theta$ by following estimates of its gradient. Proximal Policy Optimization (PPO) (Schulman et al., 2017) — a policy gradient algorithm widely used for RFT — adheres to this scheme, up to replacing the expected reward with a surrogate objective.

In this work, we highlight a fundamental optimization obstacle in RFT: we prove that the expected gradient for an input $\mathbf{x}$, *i.e.* $\nabla_\theta V(\mathbf{x}; \theta)$, vanishes when the reward standard deviation of $\mathbf{x}$ under

---

[*]Work done while interning at Apple.

[1]Code for reproducing our experiments is available at `https://github.com/apple/ml-rlgrad`.

the model is small, even if the expected reward $V(\mathbf{x}; \theta)$ is far from optimal (Section 3). The same holds for PPO, and stems from a combination of the reward maximization objective and the softmax operator used for producing distributions over tokens. In contrast, under SFT the expected gradient does not necessarily vanish in analogous circumstances.

Vanishing gradients in RFT can arise when attempting to reverse an existing behavior of the model or when the text distribution of the downstream task differs from that of the pretraining corpora. In both of these common use cases, over some inputs, the pretrained model is likely to give non-negligible probability only to outputs of roughly the same suboptimal reward. The reward standard deviation for such inputs is small, implying that their expected gradients are near-zero. Thus, RFT is anticipated to be ineffective for achieving a higher reward over them.

Using the GRUE benchmark (Ramamurthy et al., 2023) for RFT of language models, we empirically affirm the prevalence of inputs with small reward standard deviation, for which the expected gradient vanishes, and showcase its harmful effects (Section 4.1). Specifically, we find that three of seven GRUE datasets contain a considerable amount of inputs with near-zero reward standard deviation under pretrained models, while their expected reward is suboptimal. As anticipated, RFT has limited impact on the rewards of such inputs. Moreover, our experiments show that, in datasets where inputs with small reward standard deviation are more common, the reward that RFT achieves compared to SFT is worse. This indicates that vanishing gradients in RFT hinder the ability to maximize the reward function. Controlled experiments that remove possible confounding factors (Section 4.2), such as insufficient exploration, and a theoretical analysis for a simplified setting (Section 4.3), corroborate that vanishing gradients in RFT can lead to extremely slow optimization.

Lastly, we explore ways to overcome vanishing gradients in RFT of language models (Section 5). Conventional heuristics, including increasing the learning rate, using a temperature hyperparameter, and entropy regularization, are shown to be inadequate (Section 5.1). On the other hand, in agreement with prior evidence from Ramamurthy et al. (2023), we find the common practice of an initial SFT phase (*e.g.*, used in Ziegler et al. (2019); Stiennon et al. (2020); Ouyang et al. (2022); Dubois et al. (2023); Touvron et al. (2023)) to be beneficial, both in terms of the resulting reward and in significantly reducing the number of input samples with small reward standard deviation. This sheds light on the importance of SFT in an RFT pipeline — it can help alleviate vanishing gradients.

If the benefits of an initial SFT phase stem, at least partially, from alleviating vanishing gradients for the subsequent RFT phase, as our results suggest, one would expect that a relatively few SFT steps on a small number of labeled inputs can suffice. Meaning, we need only perform SFT to decrease the number of input samples with near-zero reward standard deviation, after which the potency of RFT will substantially improve. We show that this is indeed the case (Section 5.2). Remarkably, for some datasets, SFT over as few as $1\%$ of the input samples allows RFT to reach $96\%$ of the reward achieved when SFT is performed over all input samples. This demonstrates that the initial SFT phase need not be expensive in terms of compute and data labeling efforts, highlighting its effectiveness in addressing vanishing gradients in RFT.

Overall, we identify a fundamental optimization challenge in RFT, demonstrate its prevalence and detrimental effects, and explore possible solutions. Our results emphasize that being mindful for inputs whose expected gradient vanishes, as can be measured by the reward standard deviation, is crucial for successful execution of RFT. We believe further investigating ways to overcome vanishing gradients in RFT, *e.g.*, by modifying the objective (as recently done in Lu et al. (2022); Rafailov et al. (2023); Hu et al. (2023); Dong et al. (2023); Gulcehre et al. (2023)), is a valuable direction for future research.

**Related work:** We discuss related work throughout and defer a concentrated account with additional references to Appendix A.

## 2 PRELIMINARIES: SUPERVISED AND REINFORCEMENT FINETUNING

Contemporary language models, such as GPT-4 (OpenAI, 2023) and Llama2 (Touvron et al., 2023), are trained on large corpora to produce a probability distribution over sequences of tokens. The distribution is modeled in an autoregressive manner, often conditioned on an input sequence, where the output of the model is converted into a distribution over tokens through the *softmax* operator — $\mathrm{softmax}(\mathbf{z})_k := \exp(z_k)/\sum_{k'=1}^{K} \exp(z_{k'})$ for $\mathbf{z} \in \mathbb{R}^K$ and $k \in \{1, \ldots, K\}$.

Formally, let $\mathcal{X}$ be a finite vocabulary of tokens. We consider a model $f$ parameterized by $\theta \in \mathbb{R}^P$ that, conditioned on an input $\mathbf{x} = (x_1, \ldots, x_{L_{in}}) \in \mathcal{X}^{L_{in}}$ of length $L_{in}$, defines a distribution over outputs $\mathbf{y} = (y_1, \ldots, y_{L_{out}}) \in \mathcal{X}^{L_{out}}$ of length $L_{out}$ as follows:

$$p_\theta(\mathbf{y}|\mathbf{x}) = \prod_{l=1}^{L_{out}} p_\theta(y_l|\mathbf{x}, \mathbf{y}_{\leq l-1}) = \prod_{l=1}^{L_{out}} \mathrm{softmax}(f(\mathbf{x}, \mathbf{y}_{\leq l-1}; \theta))_{y_l},$$

where $\mathbf{y}_{\leq l-1} := (y_1, \ldots, y_{l-1})$ and $f(\mathbf{x}, \mathbf{y}_{\leq l-1}; \theta) \in \mathbb{R}^{|\mathcal{X}|}$. In particular, $f$ can stand for a neural network that intakes a sequence of tokens and produces logits for the distribution of the next token.

**Supervised finetuning (SFT):** SFT is a straightforward approach for adapting a pretrained language model to comply with human intent and downstream tasks. Given pairs of textual inputs and ouputs, *e.g.* instructions and desirable completions (Wei et al., 2022; Chung et al., 2022; Wang et al., 2022; Longpre et al., 2023), minimize the cross entropy loss through gradient-based methods. The SFT objective $\mathcal{L}_{\mathrm{SFT}}(\theta)$ is defined by:

$$\mathcal{L}_{\mathrm{SFT}}(\theta) := \mathbb{E}_{\mathbf{x} \sim \mathcal{D}}[\mathcal{L}_{\mathrm{SFT}}(\mathbf{x}; \theta)] \quad , \quad \mathcal{L}_{\mathrm{SFT}}(\mathbf{x}; \theta) := \mathbb{E}_{\mathbf{y} \sim \mathcal{D}(\cdot|\mathbf{x})}\left[-\ln p_\theta(\mathbf{y}|\mathbf{x})\right], \tag{1}$$

where $\mathcal{D}$ is a distribution over inputs, $\mathcal{D}(\cdot|\mathbf{x})$ is a ground truth conditional distribution over outputs, and $\mathcal{L}_{\mathrm{SFT}}(\mathbf{x}; \theta)$ is the loss over a single input $\mathbf{x}$. Although the simplicity of SFT is appealing, it comes with a few drawbacks. Namely: *(i)* human preferences can be difficult to express through ground truth labels, *e.g.*, reducing the toxicity and bias of the model's outputs; and *(ii)* obtaining high-quality labeled data can be expensive.

**Reinforcement finetuning (RFT):** Given a reward function $r : \mathcal{X}^{L_{in}} \times \mathcal{X}^{L_{out}} \to [-1, 1]$, instead of minimizing a loss over labeled inputs, the goal of RFT is to maximize the expected reward $V(\theta)$ with respect to the model:

$$V(\theta) := \mathbb{E}_{\mathbf{x} \sim \mathcal{D}}[V(\mathbf{x}; \theta)] \quad , \quad V(\mathbf{x}; \theta) := \mathbb{E}_{\mathbf{y} \sim p_\theta(\cdot|\mathbf{x})}\left[r(\mathbf{x}, \mathbf{y})\right]. \tag{2}$$

In reinforcement learning terminology, RFT forms a horizon-one (bandit) environment, where each input is a state, each output is an action that the model (*i.e.* policy) can take, and $\mathcal{D}$ is a distribution over initial states. The reward function $r$ can be learned from human preferences, *e.g.*, based on rankings of the model's outputs, which are often easier to obtain than ground truth labels (Ziegler et al., 2019; Stiennon et al., 2020; Ouyang et al., 2022). Alternatively, it may be a task specific metric, such as ROUGE (Lin, 2004), METEOR (Banerjee and Lavie, 2005), or a learned sentiment classifier (Ramamurthy et al., 2023). Policy gradient algorithms are used to maximize $V(\theta)$, by updating the parameters $\theta$ according to sample-based estimates of $\nabla_\theta V(\theta)$. PPO (Schulman et al., 2017) — a policy gradient algorithm widely used for RFT — adheres to this scheme, up to replacing $V(\mathbf{x}; \theta)$ with a surrogate (clipped) objective.

## 3 SMALL REWARD STANDARD DEVIATION IMPLIES VANISHING GRADIENTS IN REINFORCEMENT FINETUNING

We begin by introducing a, yet unnoticed, cause for vanishing gradients in RFT: the expected gradient for an input is near-zero if its reward standard deviation is small.

**Theorem 1.** *For parameters $\theta \in \mathbb{R}^P$ and input $\mathbf{x} \in \mathcal{X}^{L_{in}}$, denote the reward standard deviation of $\mathbf{x}$ under the model by $\mathrm{STD}_{\mathbf{y} \sim p_\theta(\cdot|\mathbf{x})}[r(\mathbf{x}, \mathbf{y})] := \sqrt{\mathbb{E}_{\mathbf{y} \sim p_\theta(\cdot|\mathbf{x})}[(r(\mathbf{x}, \mathbf{y}) - V(\mathbf{x}; \theta))^2]}$, where $V(\mathbf{x}; \theta)$ is the expected reward defined in Equation (2). Then, it holds that:*

$$\|\nabla_\theta V(\mathbf{x}; \theta)\| \leq 6 L_{out} \gamma(\mathbf{x}; \theta) \cdot \mathrm{STD}_{\mathbf{y} \sim p_\theta(\cdot|\mathbf{x})}[r(\mathbf{x}, \mathbf{y})]^{2/3},$$

*where $\gamma(\mathbf{x}; \theta) := \max_{l \in \{1, \ldots, L_{out}\}, \mathbf{y}_{\leq l-1} \in \mathcal{X}^{l-1}} \|J_{f(\mathbf{x}, \mathbf{y}_{\leq l-1}; \theta)}\|_2$ with $J_{f(\mathbf{x}, \mathbf{y}_{\leq l-1}; \theta)}$ standing for the Jacobian of $f(\mathbf{x}, \mathbf{y}_{\leq l-1}; \theta)$ with respect to $\theta$, and $\|\cdot\|$ and $\|\cdot\|_2$ denote the Euclidean and operator norms, respectively.*

The gradient $\nabla_\theta V(\mathbf{x}; \theta)$ vanishes, when the reward standard deviation of $\mathbf{x}$ is small, due to a combination of the reward maximization objective and the use of softmax for producing distributions over output tokens (see proof of Theorem 1 in Appendix D.1). Notably, this occurs regardless of what the expected reward $V(\mathbf{x}; \theta)$ is and, in particular, whether it is nearly optimal or not.

Vanishing gradients in RFT can be problematic when attempting to reverse an existing behavior of the model or when the text distribution of the downstream task differs from that of the pretraining

corpora. In both of these common use cases, over some inputs, the pretrained model is likely to give non-negligible probability only to outputs of roughly the same suboptimal reward. The reward standard deviation for such inputs is small, implying that their expected gradients are near-zero. Thus, RFT is anticipated to be ineffective for achieving higher reward over them. We empirically affirm this expectation in Section 4 through experiments on an RFT benchmark and controlled environments.

We make two important distinctions. First, Theorem 1 considers the expected gradient for an individual input $\mathbf{x}$, as opposed to the gradient across a batch or the population of inputs. Thus, although adaptive optimizers such as Adam (Kingma and Ba, 2015) — the standard optimizer choice for RFT — normalize the batch gradient, the contribution of inputs whose reward standard deviation is small is still negligible compared to the contribution of other inputs. Indeed, the experiments of Section 4 verify that adaptive (batch) gradient normalization does not solve the harmful effects of vanishing gradients in RFT, since they are carried out using the Adam optimizer. Second, in contrast to RFT, under SFT the expected gradient $\nabla_\theta \mathcal{L}_{\mathrm{SFT}}(\mathbf{x}; \theta)$ need not vanish unless $\mathcal{L}_{\mathrm{SFT}}(\mathbf{x}; \theta)$ is nearly minimal (and usually does not, *e.g.*, as demonstrated by the experiments of Section 4.2).

**Vanishing gradients in PPO due to small reward standard deviation:** A common practice for maximizing the reward in RFT is to use PPO, which replaces the expected reward with a surrogate objective. In Appendix B we establish that, if the reward standard deviation for an input is small, then its expected gradient under PPO vanishes, just as its expected reward gradient vanishes.

**Known causes of vanishing gradients:** Focusing mostly on tabular Markov Decision Processes, prior works identified other conditions that lead to vanishing gradients in policy gradient algorithms. Agarwal et al. (2021) showed that the gradient norm can decay exponentially with the horizon when the rewards are sparse. In RFT, however, the horizon is equal to one and the rewards are not sparse. More relevant is the fact that the expected gradient for a softmax parameterized model vanishes if it outputs near-deterministic distributions (Ahmed et al., 2019; Hennes et al., 2019; Schaul et al., 2019; Mei et al., 2020a;b; Agarwal et al., 2021; Garg et al., 2022) — a special case of the reward standard deviation being small, *i.e.* of Theorem 1. Though, this is unlikely to be an issue for RFT of language models since, barring extreme cases, the output distributions produced by language models are not near-deterministic.

## 4 Prevalence and Detrimental Effects of Vanishing Gradients in Reinforcement Finetuning

In this section, we first show that the vanishing gradients phenomenon from Section 3 is prevalent in the GRUE benchmark, using the GPT-2 (Radford et al., 2019) and T5-base (Raffel et al., 2020) pretrained language models considered in Ramamurthy et al. (2023) (Section 4.1). Specifically, among its seven datasets, three contain a substantial amount of inputs with small reward standard deviation under the corresponding pretrained model, *i.e.* their expected gradient vanishes, although their expected reward is suboptimal. As anticipated, RFT has limited impact on the rewards of these inputs compared to its impact on the rewards of other inputs. Moreover, in datasets where inputs with small reward standard deviation are more common, the reward that RFT achieves relative to SFT is worse, indicating that vanishing gradients in RFT hinder the ability to maximize the reward function. Controlled experiments that remove possible confounding factors (Section 4.2), such as insufficient exploration, and a theoretical analysis for a simplified setting with linear models (Section 4.3), further establish that vanishing gradients in RFT can lead to extremely slow reward maximization.

For brevity, we often refer to the reward standard deviation of an input under a pretrained model as the input's "pretrain reward standard deviation."

### 4.1 Vanishing Gradients in the GRUE Benchmark

We followed the experimental setup of Ramamurthy et al. (2023), up to slight adjustments (specified in Appendix F.2.1) for a fairer comparison between RFT and SFT. In particular, we adopted their default hyperparameters and considered the following text generation datasets from GRUE: NarrativeQA (Kočiskỳ et al., 2018), ToTTo (Parikh et al., 2020), CommonGen (Lin et al., 2020), IWSLT 2017 (Cettolo et al., 2017), CNN/Daily Mail (Hermann et al., 2015), DailyDialog (Li et al., 2017), and IMDB (Maas et al., 2011) — see Table 1 in Ramamurthy et al. (2023) for a complete de-

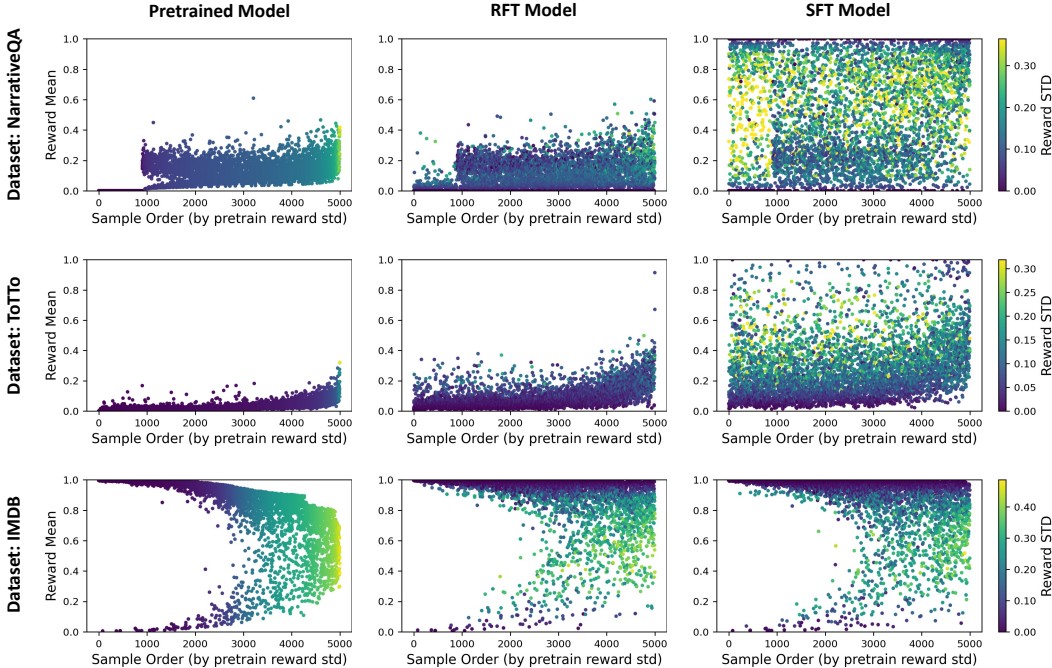

Figure 1: **Inputs with small reward standard deviation under the pretrained model,** *i.e.* **with vanishing expected gradient, are prevalent in the GRUE benchmark.** For randomly chosen subsets of 5000 train samples from the NarrativeQA, ToTTo, and IMDB datasets, presented are the reward means and standard deviations (estimated based on ten generations per input) under the pretrained, RFT, and SFT models. The samples are ordered according to their pretrain reward standard deviation, with each marker representing the reward mean and standard deviation (depicted by color) of an individual sample. Notice that a significant number of samples from NarrativeQA and ToTTo have small pretrain reward standard deviation, while their reward mean is low. Accordingly, RFT struggles to improve the reward of these inputs, especially compared to SFT. In contrast, IMDB does not suffer from this issue, and the effect of RFT and SFT over it is more similar. See Figure 8 in Appendix F.1 for identical experiments with the remaining GRUE datasets.

scription of the datasets. We ran PPO using the Adam optimizer for RFT, with the reward function in each dataset being either a task specific metric or a learned reward model (as specified in Appendix F.2.1). In contrast, SFT was performed based on ground truth completions (*i.e.* labels), also using Adam. Similarly to Ramamurthy et al. (2023), we used GPT-2 as the pretrained language model for DailyDialog and IMDB, and T5-base for the remaining datasets.

Our findings are detailed below. For conciseness, we defer some experiments, *e.g.*, using NLPO (Ramamurthy et al., 2023) instead of PPO for RFT, and implementation details to Appendix F.

**Inputs with small reward standard deviation are prevalent:** As evident from Figure 1, NarrativeQA and ToTTo contain a considerable amount of input samples whose pretrain reward standard deviation is near-zero, *i.e.* their expected gradient vanishes, while their reward mean is low. On the other hand, in IMDB the vast majority of samples have a relatively large pretrain reward standard deviation or a high reward mean. Figure 8 in Appendix F.1 further shows that the remaining datasets lie on a spectrum between these two extremes, with CommonGen also exhibiting a noticeable amount of input samples with small pretrain reward standard deviation.

As was expected (in Section 3), inputs with small pretrain reward standard deviation are more common in datasets whose text distribution differs markedly from the pretraining corpora. Specifically, NarrativeQA consists of questions concatenated to text segments and the task is to output free-form answers based on the texts, while in ToTTo each input is a table and the task is to produce a one-sentence description of it. By manual inspection, unsurprisingly, the pretrained model often generates continuations for the input text instead of obliging to the task at hand. When the desired output does not overlap with a "natural" (according to the pretraining corpora) continuation of an input, the model assigns non-negligible probability only to outputs of roughly the same low reward, leading to a small reward standard deviation and vanishing expected gradient. We provide representative examples of inputs with small and inputs with large pretrain reward standard deviation, including outputs produced by the pretrained, RFT, and SFT models, in Appendix E.

|      | NarrativeQA | ToTTo | IMDB |
|------|-------------|-------|------|
| RFT  | 0.48        | 0.46  | 0.72 |
| SFT  | 0.05        | 0.16  | 0.72 |

Table 1: **RFT affects inputs with small reward standard deviation under the pretrained model less than it affects other inputs.** Per dataset, reported is the Pearson correlation between the pretrain reward standard deviation and the absolute reward mean change due to finetuning, across the train samples. The correlations are computed based on the subsets of train samples from Figure 1. As anticipated, the correlation is considerably higher for RFT compared to SFT on NarrativeQA and ToTTo, which contain a substantial amount of samples with small pretrain reward standard deviation. In contrast, on IMDB the correlation is roughly the same since most samples with a small pretrain reward standard deviation already have a high reward mean, thus, their reward mean does not change much under both RFT and SFT. See Table 3 in Appendix F.1 for the correlations on the remaining GRUE datasets.

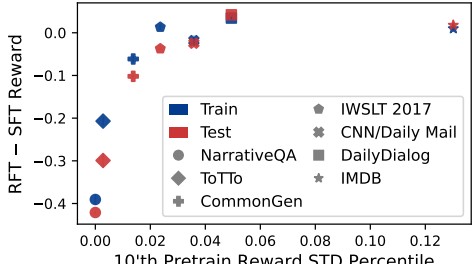

Figure 2: **RFT performance (relative to SFT) is worse when inputs with small reward standard deviation are prevalent.** Per dataset, the difference between the mean reward achieved by RFT and SFT is plotted against the 10'th percentile of the reward standard deviation under the pretrained model. Means are taken over five runs and error bars (indiscernible) mark standard deviations. We exclude inputs with near-optimal reward mean under the pretrained model (higher than 0.9) when computing the percentiles, since small reward standard deviation is only problematic if the reward mean is not high to begin with. Observe that, the lower the pretrain reward standard deviation percentile is, *i.e.* the more train samples have small pretrain reward standard deviation, the worse the reward that RFT achieves relative to SFT.

**RFT affects inputs with small reward standard deviation less than other inputs:** Figure 1 qualitatively demonstrates that the rewards of train samples with small reward standard deviation change less due to RFT than the rewards of other train samples, whereas under SFT the change is more uniform across the samples. To quantify this observation, for each dataset, Table 1 reports the Pearson correlation between the pretrain reward standard deviation and the absolute reward mean change due to finetuning, across the train samples. Indeed, the correlations for RFT are positive and significantly higher than those for SFT.

**RFT performance is worse when inputs with small reward standard deviation are prevalent:** Lastly, we examine whether there is a connection between the prevalence of inputs with small pretrain reward standard deviation, *i.e.* with vanishing expected gradient, and the reward that RFT achieves. A simple way to measure the prevalence of such inputs in a dataset is to examine some percentile of the pretrain reward standard deviation, over the train samples. Remarkably, Figure 2 shows that the lower the 10'th percentile of the pretrain reward standard deviation is, the worse the reward that RFT achieves relative to SFT. This observation is not sensitive to the choice of percentile, as long as it is not too large — Figure 9 in Appendix F.1 demonstrates a similar trend for varying percentile choices.[2]

Our investigation reveals that the extent of vanishing expected gradients in a dataset, as measured by the reward standard deviation, is indicative of how well RFT performs. Yet one cannot infer a causal relation due to possible confounding factors. Mainly, the large output space in language generation introduces the well-known challenge of exploration (Ranzato et al., 2016; Nguyen et al., 2017; Choshen et al., 2020), which in the context of RFT translates to the challenge of obtaining accurate estimates for expected gradients (Greensmith et al., 2004; Schulman et al., 2015; Tucker et al., 2018). A natural question is whether the difficulty of RFT to maximize the rewards of inputs with small reward standard deviation indeed stems from their expected gradients vanishing, or rather solely from insufficient exploration. We address this question via controlled experiments (Section 4.2) and a theoretical analysis (Section 4.3). They show how vanishing gradients in RFT, due to small reward standard deviation, can lead to extremely slow reward maximization even under perfect exploration, *i.e.* even when we have access to expected gradients.

## 4.2 Controlled Experiments

To account for the possible confounding effect of insufficient exploration, *i.e.* of inaccurate gradient estimates, we consider environments where exploration is not an obstacle. Namely, environments

---

[2]For completeness, the train and test rewards that the pretrained, RFT, and SFT models obtain over all datasets are provided in Table 4 of Appendix F.1.

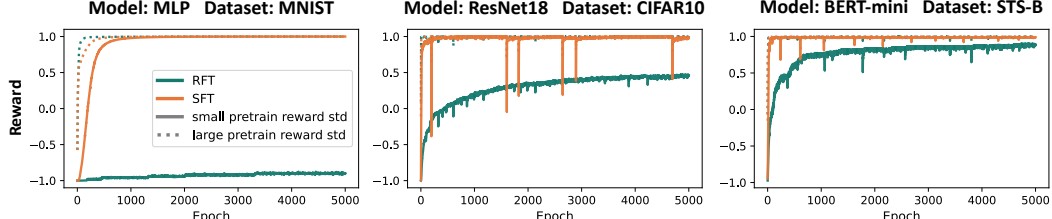

Figure 3: RFT struggles to maximize the reward over inputs with small reward standard deviation under the pretrained model, *i.e.* inputs with vanishing expected gradient, even with perfect exploration. On the contrary, SFT easily leads to maximal reward. For the controlled environments described in Section 4.2, in which RFT has access to expected gradients, displayed is the train reward that RFT and SFT attain throughout optimization, separately for train samples with small and train samples with large pretrain reward standard deviation. In Appendix F.1, Figure 12 provides plots showing the reward standard deviation and gradient norm throughout optimization, and Figure 13 presents identical experiments with stochastic gradient descent instead of Adam.

with a relatively small number of outputs in which one can compute the expected gradient for an input instead of estimating it. We observe that RFT is still unable to achieve high reward over inputs whose pretrain reward standard deviation is small, in stark contrast to SFT which easily does so. This demonstrates that, in the presence of inputs with vanishing expected gradient, RFT can suffer from extremely slow optimization despite perfect exploration. For conciseness, we defer some experiments and implementation details to Appendix F.

We seek to create finetuning environments in which: *(i)* it is possible to compute the expected reward gradient $\nabla_\theta V(\mathbf{x}; \theta)$ for an input $\mathbf{x}$; and *(ii)* there are inputs whose pretrain reward standard deviation is small. To that end, we take a classification dataset with a modest number of labels, *e.g.* STS-B (Cer et al., 2017).[3] This ensures *(i)*: we can compute the expected gradient given an input by simply querying the reward for all labels.

For pretraining, we assign to each train sample multiple ground truth labels by randomly choosing additional labels, and minimize the cross entropy loss starting from a randomly initialized model, *e.g.* a BERT model (Devlin et al., 2018). At the end of this process, given a train sample, the model roughly outputs a uniform distribution over the sample's pretraining labels. During finetuning, every train sample is assigned a (possibly different) single ground truth label, whose reward for RFT is set to 1, whereas the reward of an incorrect label is set to $-1$. For a subset of the samples, we set their finetuning label such that it was not included in their pretraining labels. This guarantees *(ii)*: the pretrain reward standard deviation of these train samples is small since the model (roughly) produces a uniform distribution over labels with the same suboptimal reward. On the other hand, the reward standard deviation is large for the remaining train samples, whose finetuning label existed in their pretraining labels.

We followed the blueprint above for three different model and dataset configurations: a multilayer perceptron (MLP) on MNIST (LeCun, 1998), ResNet18 (He et al., 2016) on CIFAR10 (Krizhevsky et al., 2009), and BERT-mini (Turc et al., 2019) on STS-B (Cer et al., 2017). RFT was performed by maximizing the expected reward (Equation (2)) via the Adam optimizer, and SFT was performed by minimizing the cross entropy loss, also via the Adam optimizer. Figure 3 tracks the train reward that RFT and SFT attain throughout optimization, separately for train samples with small and train samples with large pretrain reward standard deviation. The results confirm that, despite having access to expected gradients and the small output space, RFT is unable to achieve high reward in a reasonable time over inputs with small pretrain reward standard deviation. On the other hand, it does not encounter difficulty for inputs with large pretrain reward standard deviation. The behavior of SFT strikingly differs — it quickly reaches the maximal reward for both types of inputs.

**Insignificance of the pretrain expected reward:** The pretrain expected reward (*i.e.* expected reward under the pretrained model) of inputs with small pretrain reward standard deviation is often close to minimal, as shown on the GRUE benchmark in Section 4.1. This is also the case in the controlled experiments of Figure 3. However, we highlight that the vanishing gradient phenomenon established in Section 3 is independent of the expected reward — the expected gradient for an input vanishes if its reward standard deviation is small, regardless of what the expected reward is. We empirically demonstrate this independence through experiments analogous to those of Figure 3, in

---

[3]We quantize the continuous labels of STS-B, which reside in $[0, 5]$, to six values via rounding.

which, although the initial expected reward of inputs with small pretrain reward standard deviation is relatively high, RFT is still unable to maximize their reward — see Figure 14 in Appendix F.1.

### 4.3 THEORETICAL ANALYSIS: SLOW OPTIMIZATION DUE TO VANISHING GRADIENTS

Through an analysis for a simplified setting, we theoretically demonstrate that vanishing gradients in RFT, due to small reward reward standard deviation, can lead to impractically slow optimization. Specifically, we prove an exponential optimization time separation between RFT and SFT, with respect to the pretrain reward standard deviation of an input $\mathbf{x}$: the time it takes RFT to assign maximal probability for the correct label of $\mathbf{x}$ is $\Omega(1/\operatorname{STD}_{\mathbf{y} \sim p_{\theta^{(0)}}(\cdot|\mathbf{x})}[r(\mathbf{x}, \mathbf{y})]^2)$, where $\theta^{(0)}$ is the pretrained model parameters, while the time it takes SFT to do so is only $\mathcal{O}(\ln(1/\operatorname{STD}_{\mathbf{y} \sim p_{\theta^{(0)}}(\cdot|\mathbf{x})}[r(\mathbf{x}, \mathbf{y})]))$. Our analysis considers linear models trained over orthonormal inputs using a small learning rate. Although the setting is rather simplistic, it gives insight into the detrimental effects of vanishing gradients due to small reward standard deviation. The details of our analysis are given in Appendix C; we regard extending it to more complex settings as an interesting avenue for future work.

## 5 OVERCOMING VANISHING GRADIENTS IN REINFORCEMENT FINETUNING

Section 3 established that small reward standard deviation leads to vanishing gradients in RFT, which Section 4 demonstrated to be detrimental in a real-world benchmark and controlled environments. In this section, we explore ways to overcome vanishing gradients in RFT. We show that the conventional heuristics of increasing the learning rate, applying temperature to the logits, and entropy regularization are inadequate (Section 5.1). On the other hand, the common practice of an initial SFT phase (*e.g.*, used in Ziegler et al. (2019); Stiennon et al. (2020); Ouyang et al. (2022); Dubois et al. (2023); Touvron et al. (2023)) is shown to be highly effective, both in terms of the resulting reward and in reducing the number of inputs with small reward standard deviation. This sheds light on the importance of SFT in an RFT pipeline — it can help alleviate vanishing gradients. Moreover, our experiments reveal that a relatively small number of SFT optimization steps on as few as $1\%$ of the input samples can suffice, indicating that the initial SFT phase need not be expensive in terms of compute and data labeling efforts (Section 5.2).

We focus on three datasets from the GRUE benchmark, which were found in Section 4.1 to suffer the most from vanishing gradients: NarrativeQA, ToTTo, and CommonGen. For conciseness, we defer to Appendix F the results for ToTTo and CommonGen, as well as some implementation details.

### 5.1 INADEQUACY OF CONVENTIONAL HEURISTICS

We validate the resilience of vanishing gradients in RFT to common heuristics. Three sensible methods that may seem suitable for addressing the problem are: increasing the learning rate, applying temperature to the logits, and entropy regularization (which is known theoretically to improve convergence rates in simple settings (Mei et al., 2020b)). However, we do not expect modifying the learning rate to help since it increases the effective gradient norm over a batch of inputs. Thus, the contribution of inputs whose reward standard deviation is small will continue to be negligible compared to the contribution of other inputs. Furthermore, while applying temperature to the logits or entropy regularization may yield non-zero expected gradients, the gradients will likely not be in a direction that aids maximizing the reward due to the large output space of language models.

Based on the default hyperparameters from Ramamurthy et al. (2023), we carried out RFT runs with: *(i)* several values of larger learning rates, temperature, and entropy regularization coefficient; and, for comparison, *(ii)* an initial SFT phase, whereby the pretrained model is first finetuned in a supervised manner before applying RFT. Confirming our expectation, none of the RFT configurations led to a higher train reward compared to the default one. By contrast, in line with prior evidence (Ramamurthy et al., 2023), an initial SFT phase greatly increases the reward that RFT attains — see Table 6 in Appendix F.1. Furthermore, to gauge the effect of the considered heuristics on inputs with small reward standard deviation, Figure 15 in Appendix F.1 presents the reward mean and standard deviation of individual train samples after applying RFT with each of the heuristics. In comparison to the effect of the default RFT configuration and SFT (shown in Figure 1), it is evident that the considered heuristics do not reduce the number of samples with small reward standard deviation nor substantially improve their rewards, whereas SFT is highly effective in doing so.

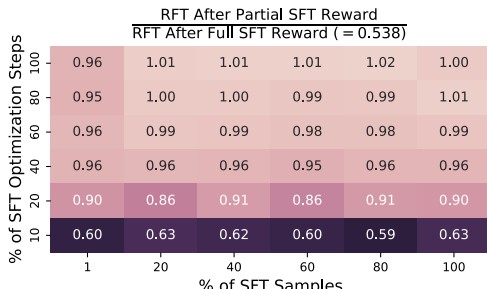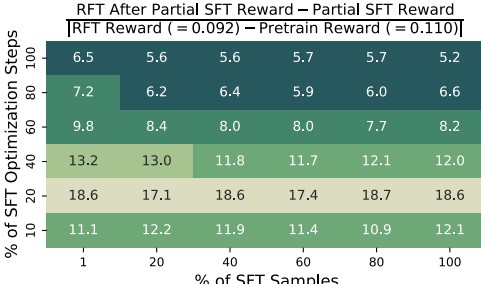

Figure 4: On datasets in which RFT suffers from vanishing gradients, a few initial SFT optimization steps on a small number of labeled inputs substantially boost the efficacy of RFT. For the NarrativeQA dataset, reported metrics are based on the mean train reward achieved when performing RFT after an initial SFT phase with various percentages of optimization steps and labeled inputs (over three random seeds). A "full" SFT phase refers to 100% of the steps and labeled inputs used by Ramamurthy et al. (2023). Observe that the number of optimization steps and labeled inputs can be greatly reduced without causing a significant degradation in reward (left). Furthermore, RFT becomes roughly 5 to 18 times more potent after the initial SFT phase (right). We refer to Appendix F.1 for analogous plots reporting metrics based on the mean test reward (Figure 16), as well as identical experiments on the ToTTo and CommonGen datasets (Figures 18 and 19, respectively).

## 5.2 A FEW SUPERVISED FINETUNING STEPS ON A SMALL NUMBER OF INPUTS SUFFICE

Although an initial SFT phase often yields improved performance, it comes with a major drawback: it requires additional compute and, more importantly, relies on labeled data, which can be especially expensive to obtain for language generation. The results of Section 5.1 suggest that the benefits of an initial SFT phase stem, at least partially, from mitigating vanishing gradients for the subsequent RFT phase. This gives rise to the possibility that a relatively few SFT steps on a small number of labeled inputs suffice. Meaning, we need only perform SFT to decrease the number of input samples with near-zero reward standard deviation, after which the potency of RFT will substantially improve.

We investigate this prospect by decreasing the percentage of optimization steps and labeled inputs used for the initial SFT phase, with 100% corresponding to the default setup from Ramamurthy et al. (2023). As Figure 4 shows, the number of SFT optimization steps and labeled inputs can be greatly reduced without significantly harming the reward achieved by RFT. In particular, using 40% of the optimization steps and as few as 1% of the input samples allows RFT to reach 96% of the reward achieved when performing a "full" initial SFT phase. Moreover, the efficacy of RFT, as quantified by the difference between the reward after and before RFT, exhibits a substantial increase as a result of the initial SFT phase. To understand the cause for this increase, we examine the impact of a partial SFT phase on individual train samples (see Figure 17 in Appendix F.1). We observe that the number of samples with small reward standard deviation, *i.e.* with vanishing expected gradient, is considerably reduced. This supports the perspective that the initial SFT phase is beneficial due to mitigating vanishing gradients for the following RFT phase, and showcases that the SFT phase need not be expensive in terms of compute and data labeling efforts.

## 6 CONCLUSION

In this work, we identified a fundamental optimization challenge in RFT: the expected gradient for an input vanishes when its reward standard deviation is small. We then demonstrated the prevalence and detrimental effects of this phenomenon, as well as explored possible solutions. Overall, our results emphasize that being mindful for inputs whose expected gradient vanishes, as can be measured by the reward standard deviation, is crucial for successful execution of RFT.

**Limitations and future work:** To facilitate efficient experimentation, we considered language models of small to moderate size and did not incorporate reward functions learned from human feedback. An exciting next step is to study the extent to which our empirical findings carry over to more complex finetuning pipelines, including larger models and iterative human feedback (*cf.* Bai et al. (2022); OpenAI (2023); Touvron et al. (2023)). Additionally, Section 5 demonstrated that three common heuristics, which may seem suitable for overcoming vanishing gradients in RFT, are inadequate, whereas performing a few initial SFT steps over a small number of labeled samples is a promising solution. Our investigation of possible solutions is not exhaustive. We believe further exploring ways to overcome vanishing gradients in RFT, *e.g.*, by modifying the objective (as recently done in Lu et al. (2022); Rafailov et al. (2023); Hu et al. (2023); Dong et al. (2023); Gulcehre et al. (2023)), is a valuable direction for future research.

## ACKNOWLEDGEMENTS

We thank Eshbal Hezroni for aid in preparing illustrative figures. NR is supported by the Apple Scholars in AI/ML PhD fellowship.

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

## A    RELATED WORK

**Reinforcement learning for language generation:** Our work highlights an optimization challenge in policy gradient-based RFT, motivated by its widespread use for adapting pretrained language models (Ziegler et al., 2019; Stiennon et al., 2020; Ouyang et al., 2022; Bai et al., 2022; Dubois et al., 2023; OpenAI, 2023; Touvron et al., 2023). Reinforcement learning has been applied more broadly in task specific settings, for example: translation (Ranzato et al., 2016; Bahdanau et al., 2017; Nguyen et al., 2017; Kiegeland and Kreutzer, 2021), summarization (Ranzato et al., 2016; Paulus et al., 2017; Wu and Hu, 2018), and dialog (Li et al., 2016; Zhou et al., 2017; Jaques et al., 2019). Policy gradient algorithms are most commonly the method of choice. Though, value-based methods have also been used (as in Jaques et al. (2019)), which notably do not suffer from the vanishing gradients phenomenon identified in this work.

**Optimization challenges in policy gradient:** The objective landscape of policy gradient is non-convex even in simple settings (*e.g.* with linear models). Thus, deriving conditions under which efficient optimization occurs has proven difficult (Mei et al., 2020a;b; Agarwal et al., 2021; Li et al., 2021). In environments with a large output (*i.e.* action) space, for which language generation is a prominent example, another well-known challenge is that of exploration (see, *e.g.*, Ranzato et al. (2016); Nguyen et al. (2017); Choshen et al. (2020)), which in the context of RFT boils down to obtaining accurate expected gradient estimates (Greensmith et al., 2004; Schulman et al., 2015; Tucker et al., 2018). Interestingly, our results highlight that, for inputs with small reward standard deviation, obtaining more accurate estimates is unlikely to be useful since their expected gradient vanishes.

As discussed in Section 3, the expected gradient for a softmax parameterized model is known to vanish when the model outputs near-deterministic distributions (Mei et al., 2020a;b; Agarwal et al., 2021) — a special case of the reward standard deviation being small, *i.e.* of Theorem 1. Under this special case, it was empirically demonstrated that policy gradient algorithms struggle to maximize the reward, even with perfect exploration, *i.e.* even with access to expected gradients (Ahmed et al., 2019; Hennes et al., 2019; Schaul et al., 2019; Mei et al., 2020a;b; Garg et al., 2022). However, the relevance of near-deterministic output distributions to RFT is limited since language models rarely produce such distributions. In contrast, Section 3 establishes a more general condition leading to vanishing gradients, relevant to RFT, that is demonstrated in Section 4 to result in optimization difficulties.

**Overcoming optimization challenges in policy gradient:** To address the poor optimization landscape of softmax parameterized policy gradient, prior work proposed to replace softmax with an alternative transformation (Mei et al., 2020a) and modify the gradient update step (Hennes et al., 2019; Garg et al., 2022). Examining the utility of these methods for RFT may be an interesting topic for future work.

## B    SMALL REWARD STANDARD DEVIATION IMPLIES VANISHING GRADIENTS IN PPO

Theorem 1 (in Section 3) established that the expected reward gradient $\nabla_\theta V(\mathbf{x}; \theta)$, for an input $\mathbf{x}$, vanishes if the reward standard deviation of $\mathbf{x}$ is small. In this appendix, we prove that the expected gradient under PPO — a policy gradient algorithm widely used for RFT — vanishes as well in that case.

PPO most often replaces the expected reward $V(\mathbf{x}; \theta)$ with a surrogate (clipped) objective (Ouyang et al., 2022; Bai et al., 2022; Ramamurthy et al., 2023; Touvron et al., 2023). That is, it maximizes $V_{\text{PPO}}(\theta) := \mathbb{E}_{\mathbf{x} \sim \mathcal{D}}[V_{\text{PPO}}(\mathbf{x}; \theta)]$ with (*cf.* Schulman et al. (2017)):

$$V_{\text{PPO}}(\mathbf{x}; \theta) := \mathbb{E}_{\mathbf{y} \sim p_{\bar{\theta}}(\cdot|\mathbf{x})}\left[\min\left\{\frac{p_\theta(\mathbf{y}|\mathbf{x})}{p_{\bar{\theta}}(\mathbf{y}|\mathbf{x})}A_{\bar{\theta}}(\mathbf{x}, \mathbf{y}), \text{clip}\left(\frac{p_\theta(\mathbf{y}|\mathbf{x})}{p_{\bar{\theta}}(\mathbf{y}|\mathbf{x})}, 1 - \delta, 1 + \delta\right)A_{\bar{\theta}}(\mathbf{x}, \mathbf{y})\right\}\right],$$

(3)

where $\bar{\theta} \in \mathbb{R}^P$ stands for fixed reference parameters, which are updated to the current parameter values at the start of each PPO iteration, $A_{\bar{\theta}}(\mathbf{x}, \mathbf{y}) := r(\mathbf{x}, \mathbf{y}) - V(\mathbf{x}; \bar{\theta})$ is the advantage function with respect to $\bar{\theta}$, and $\text{clip}(z, 1 - \delta, 1 + \delta)$ projects $z \in \mathbb{R}$ onto the interval $[1 - \delta, 1 + \delta]$ for

$\delta \in (0, 1)$.[4] Another, less common, variant of PPO includes the Kullback–Leibler (KL) divergence between $p_\theta(\cdot|\mathbf{x})$ and $p_{\bar{\theta}}(\cdot|\mathbf{x})$ as a regularization term instead of clipping (*cf.* Schulman et al. (2017)):

$$V_{\text{PPO}-\text{KL}}(\mathbf{x}; \theta) := \mathbb{E}_{\mathbf{y} \sim p_{\bar{\theta}}(\cdot|\mathbf{x})} \left[ \frac{p_\theta(\mathbf{y}|\mathbf{x})}{p_{\bar{\theta}}(\mathbf{y}|\mathbf{x})} A_{\bar{\theta}}(\mathbf{x}, \mathbf{y}) \right] - \lambda \cdot \mathbb{D}_{\text{KL}}(p_{\bar{\theta}}(\cdot|\mathbf{x}) || p_\theta(\cdot|\mathbf{x})), \quad (4)$$

for $\lambda \geq 0$.

Propositions 1 and 2 below establish that, if the reward standard deviation for an input is small, then its expected gradient under both PPO variants vanishes, just as its expected reward gradient vanishes. Specifically, we show that the distance between $\nabla_\theta V(\mathbf{x}; \theta)$ and $\nabla_\theta V_{\text{PPO}}(\mathbf{x}; \theta)$ depends on the total variation distance between $p_\theta(\cdot|\mathbf{x})$ and $p_{\bar{\theta}}(\cdot|\mathbf{x})$, where $\bar{\theta}$ denotes the PPO reference parameters. The same is true for the distance between $\nabla_\theta V(\mathbf{x}; \theta)$ and $\nabla_\theta V_{\text{PPO}-\text{KL}}(\mathbf{x}; \theta)$. At the beginning of each PPO iteration $\bar{\theta}$ is updated to the current parameter values. Hence, the distance between $p_\theta(\cdot|\mathbf{x})$ and $p_{\bar{\theta}}(\cdot|\mathbf{x})$ typically remains small throughout optimization, and is exactly zero at the beginning of each PPO iteration. The proofs of Propositions 1 and 2 are deferred to Appendices D.2 and D.3, respectively.

**Proposition 1.** *Under the notation of Theorem 1, let $\bar{\theta} \in \mathbb{R}^P$ be the reference parameters at some iteration of PPO (Equation (3)). For an input $\mathbf{x} \in \mathcal{X}^{L_{in}}$, we denote the total variation distance between $p_\theta(\cdot|\mathbf{x})$ and $p_{\bar{\theta}}(\cdot|\mathbf{x})$ by $\text{TV}(p_\theta(\cdot|\mathbf{x})||p_{\bar{\theta}}(\cdot|\mathbf{x})) := \frac{1}{2} \sum_{\mathbf{y} \in \mathcal{X}^{L_{out}}} |p_\theta(\mathbf{y}|\mathbf{x}) - p_{\bar{\theta}}(\mathbf{y}|\mathbf{x})|$. Then, it holds that:*

$$\|\nabla_\theta V_{\text{PPO}}(\mathbf{x}; \theta) - \nabla_\theta V(\mathbf{x}; \theta)\| \leq \frac{12 L_{out} \gamma(\mathbf{x}; \theta)}{\delta} \cdot \text{TV}(p_\theta(\cdot|\mathbf{x})||p_{\bar{\theta}}(\cdot|\mathbf{x})),$$

*where $\delta \in (0, 1)$ is the PPO clipping hyperparameter. Consequently, by Theorem 1:*

$$\|\nabla_\theta V_{\text{PPO}}(\mathbf{x}; \theta)\| \leq 6 L_{out} \gamma(\mathbf{x}; \theta) \left( \text{STD}_{\mathbf{y} \sim p_\theta(\cdot|\mathbf{x})}[r(\mathbf{x}, \mathbf{y})]^{2/3} + \frac{2}{\delta} \cdot \text{TV}(p_\theta(\cdot|\mathbf{x})||p_{\bar{\theta}}(\cdot|\mathbf{x})) \right),$$

*where at the beginning of each PPO iteration $\theta = \bar{\theta}$, meaning $\text{TV}(p_\theta(\cdot|\mathbf{x})||p_{\bar{\theta}}(\cdot|\mathbf{x})) = 0$.*

**Proposition 2.** *Under the notation of Theorem 1, let $\bar{\theta} \in \mathbb{R}^P$ be the reference parameters at some iteration of the KL regularized PPO variant (Equation (4)). For an input $\mathbf{x} \in \mathcal{X}^{L_{in}}$, we denote the total variation distance between $p_\theta(\cdot|\mathbf{x})$ and $p_{\bar{\theta}}(\cdot|\mathbf{x})$ by $\text{TV}(p_\theta(\cdot|\mathbf{x})||p_{\bar{\theta}}(\cdot|\mathbf{x})) := \frac{1}{2} \sum_{\mathbf{y} \in \mathcal{X}^{L_{out}}} |p_\theta(\mathbf{y}|\mathbf{x}) - p_{\bar{\theta}}(\mathbf{y}|\mathbf{x})|$. Then, it holds that:*

$$\|\nabla_\theta V_{\text{PPO}-\text{KL}}(\mathbf{x}; \theta) - \nabla_\theta V(\mathbf{x}; \theta)\| \leq 4 L_{out} \gamma(\mathbf{x}; \theta) \lambda \cdot \text{TV}(p_\theta(\cdot|\mathbf{x})||p_{\bar{\theta}}(\cdot|\mathbf{x})),$$

*where $\lambda \geq 0$ is the KL regularization coefficient. Consequently, by Theorem 1:*

$$\|\nabla_\theta V_{\text{PPO}-\text{KL}}(\mathbf{x}; \theta)\| \leq 6 L_{out} \gamma(\mathbf{x}; \theta) \left( \text{STD}_{\mathbf{y} \sim p_\theta(\cdot|\mathbf{x})}[r(\mathbf{x}, \mathbf{y})]^{2/3} + \frac{2\lambda}{3} \cdot \text{TV}(p_\theta(\cdot|\mathbf{x})||p_{\bar{\theta}}(\cdot|\mathbf{x})) \right),$$

*where at the beginning of each PPO-KL iteration $\theta = \bar{\theta}$, meaning $\text{TV}(p_\theta(\cdot|\mathbf{x})||p_{\bar{\theta}}(\cdot|\mathbf{x})) = 0$.*

## C  DETAILS OF THEORETICAL ANALYSIS: SLOW OPTIMIZATION DUE TO VANISHING GRADIENTS

In this appendix, we deliver our formal analysis of slow optimization due to vanishing gradients in RFT, referred to in Section 4.3.

Consider a linear multiclass classification problem with $D$-dimensional inputs and $K \geq 2$ labels, where the model $f : \mathbb{R}^D \times \mathbb{R}^{K \times D} \to \mathbb{R}^K$ is parameterized by a matrix $\theta \in \mathbb{R}^{K \times D}$. Conditioned on an input $\mathbf{x} \in \mathbb{R}^D$, the model produces a distribution over labels $\mathbf{y} \in \{1, \ldots, K\}$ through the softmax operator as follows:

$$p_\theta(\mathbf{y}|\mathbf{x}) = \text{softmax}(f(\mathbf{x}; \theta))_{\mathbf{y}} = \text{softmax}(\theta \mathbf{x})_{\mathbf{y}}.$$

Suppose that we are given parameters $\theta^{(0)} \in \mathbb{R}^{K \times D}$ obtained through some pretraining process, and a finetuning training set $(\mathbf{x}_1, \mathbf{y}_1), \ldots, (\mathbf{x}_N, \mathbf{y}_N) \in \mathbb{R}^D \times \{1, \ldots, K\}$ consisting of $N$ orthonormal

---

[4]We consider a horizon-one formulation of PPO. One can also treat the environment as having a horizon of $L_{out}$, with all intermediate rewards being zero and a discount factor of one.

inputs (meaning, $N \leq D$) and corresponding ground truth labels. A natural way to define a reward function in this case is, given a train sample $\mathbf{x}_n$, set the reward of the ground truth label to 1, *i.e.* $r(\mathbf{x}_n, \mathbf{y}_n) = 1$, and the reward of all other labels to $-1$, *i.e.* $r(\mathbf{x}_n, \mathbf{y}) = -1$ for all $\mathbf{y} \neq \mathbf{y}_n$.

We analyze the optimization dynamics of RFT when carried out via gradient descent with the rewards specified above over the training set, *i.e.* of gradient descent over $-V(\theta)$ (Equation (2)) with $\mathcal{D}$ being the uniform distribution over $\mathbf{x}_1, \ldots, \mathbf{x}_N$. The dynamics of gradient descent are commonly studied via the infinitesimal limit of a small learning rate, known as gradient flow (*cf.* Saxe et al. (2014); Du et al. (2019); Lyu and Li (2020); Razin and Cohen (2020); Elkabetz and Cohen (2021); Frei et al. (2023)), which in the context of RFT boils down to assuming the parameters are governed by the following differential equation:

$$\theta_{\mathrm{RFT}}(0) = \theta^{(0)} \quad , \quad \frac{d}{dt}\theta_{\mathrm{RFT}}(t) := \nabla_\theta V(\theta_{\mathrm{RFT}}(t)) \ , \ t \geq 0 \, ,$$

where $\theta_{\mathrm{RFT}}(t)$ denotes the parameters at time $t \geq 0$ of optimization. Analogously, in SFT, gradient flow for minimizing the cross entropy loss $\mathcal{L}_{\mathrm{SFT}}(\theta)$ (Equation (1)) over the training set amounts to:

$$\theta_{\mathrm{SFT}}(0) = \theta^{(0)} \quad , \quad \frac{d}{dt}\theta_{\mathrm{SFT}}(t) := -\nabla_\theta \mathcal{L}_{\mathrm{SFT}}(\theta_{\mathrm{SFT}}(t)) \ , \ t \geq 0 \, ,$$

with the distribution $\mathcal{D}$ in the definition of $\mathcal{L}_{\mathrm{SFT}}(\theta)$ being uniform over the training set, and $\theta_{\mathrm{SFT}}(t)$ denoting the parameters at time $t \geq 0$.

Theorem 2 establishes an exponential optimization time separation between RFT and SFT with respect to the reward standard deviation of an input under $\theta^{(0)}$. Resonating with the controlled experiments of Section 4.2, we assume that there exists a train sample misclassified by $\theta^{(0)}$, for which $\theta^{(0)}$ assigns the same probability to all incorrect labels.

**Theorem 2.** *Consider the setting described above (in Appendix C), and let $\mathbf{x}_n$ be a train sample that $\theta^{(0)}$ misclassifies, i.e. $p_{\theta^{(0)}}(\mathbf{y}_n|\mathbf{x}_n) \notin \mathrm{argmax}_{\mathbf{y} \in \{1,\ldots,K\}} \, p_{\theta^{(0)}}(\mathbf{y}|\mathbf{x}_n)$, for which $\theta^{(0)}$ assigns the same probability to all incorrect labels, i.e. $p_{\theta^{(0)}}(\mathbf{y}|\mathbf{x}_n) = p_{\theta^{(0)}}(\mathbf{y}'|\mathbf{x}_n)$ for all $\mathbf{y}, \mathbf{y}' \neq \mathbf{y}_n$. Denote by $t_{\mathrm{RFT}}, t_{\mathrm{SFT}} \geq 0$ the initial times at which RFT and SFT classify $\mathbf{x}_n$ correctly, respectively. Formally:*

$$t_{\mathrm{RFT}} := \min\Big\{ t \geq 0 : p_{\theta_{\mathrm{RFT}}(t)}(\mathbf{y}_n|\mathbf{x}_n) \in \mathrm{argmax}_{\mathbf{y} \in \{1,\ldots,K\}} \, p_{\theta_{\mathrm{RFT}}(t)}(\mathbf{y}|\mathbf{x}_n) \Big\} \, ,$$

$$t_{\mathrm{SFT}} := \min\Big\{ t \geq 0 : p_{\theta_{\mathrm{SFT}}(t)}(\mathbf{y}_n|\mathbf{x}_n) \in \mathrm{argmax}_{\mathbf{y} \in \{1,\ldots,K\}} \, p_{\theta_{\mathrm{SFT}}(t)}(\mathbf{y}|\mathbf{x}_n) \Big\} \, .$$

*Then, it holds that:*

$$t_{\mathrm{RFT}} = \Omega\bigg( \frac{1}{\mathrm{STD}_{\mathbf{y} \sim p_{\theta^{(0)}}(\cdot|\mathbf{x}_n)}[r(\mathbf{x}_n, \mathbf{y})]^2} \bigg) \ , \ t_{\mathrm{SFT}} = \mathcal{O}\bigg( \ln\Big( \frac{1}{\mathrm{STD}_{\mathbf{y} \sim p_{\theta^{(0)}}(\cdot|\mathbf{x}_n)}[r(\mathbf{x}_n, \mathbf{y})]} \Big) \bigg) \, .$$

*That is, there exist constants $c_1, c_1' \in \mathbb{R}_{>0}$ and $c_2, c_2' \in \mathbb{R}$ (i.e. $c_1, c_1', c_2, c_2'$ do not depend on the initial parameters $\theta^{(0)}$) such that $t_{\mathrm{RFT}} \geq c_1 \cdot (1/\mathrm{STD}_{\mathbf{y} \sim p_{\theta^{(0)}}(\cdot|\mathbf{x})}[r(\mathbf{x}, \mathbf{y})]^2) + c_2$, whereas $t_{\mathrm{SFT}} \leq c_1' \cdot \ln(1/\mathrm{STD}_{\mathbf{y} \sim p_{\theta^{(0)}}(\cdot|\mathbf{x})}[r(\mathbf{x}, \mathbf{y})]) + c_2'$.*

*Proof sketch (full proof in Appendix D.4).* Since $\mathbf{x}_1, \ldots, \mathbf{x}_N$ are orthogonal, the logits that the model produces for $\mathbf{x}_n$, *i.e.* $f(\mathbf{x}_n; \theta_{\mathrm{RFT}}(t))$ in RFT and $f(\mathbf{x}_n; \theta_{\mathrm{SFT}}(t))$ in SFT, evolve independently of the remaining train samples. Furthermore, the assumption that $\theta^{(0)}$ assigns equal probabilities to all incorrect labels of $\mathbf{x}_n$ implies that they remain equal throughout optimization, and so do the respective logits. Let $\mu(t)$ be the difference between the logit of $\mathbf{y}_n$ and of some $\mathbf{y} \neq \mathbf{y}_n$ at time $t$, given $\mathbf{x}_n$, and note that the initial time at which $\mathbf{x}_n$ is classified correctly is the initial time at which $\mu(t) = 0$. The above facilitate deriving the exact dependence of $t_{\mathrm{RFT}}$ and $t_{\mathrm{SFT}}$ on $f(\mathbf{x}_n; \theta^{(0)})$, *i.e.* on the initial logits produced by the model given $\mathbf{x}_n$, from the differential equation governing $\mu(t)$. The proof concludes by relating $f(\mathbf{x}_n; \theta^{(0)})$ to the reward standard deviation of $\mathbf{x}_n$ under $\theta^{(0)}$. $\qquad\square$

We note that our lower bound on $t_{\mathrm{RFT}}$ is similar in spirit to Theorem 1 of Mei et al. (2020a). For a multi-armed bandit problem, Mei et al. (2020a) lower bounds the time until reaching near-optimal reward, in terms of the initial optimal action probability. The main difference between the results, besides the exact technical conditions, is that we derive the dependence of optimization time on the initial reward standard deviation and contrast it with the optimization time under SFT.

## D DEFERRED PROOFS

### D.1 PROOF OF THEOREM 1

Differentiating $V(\mathbf{x}; \theta)$ with respect to $\theta$ using the log-derivative trick we get:

$$
\begin{aligned}
\nabla_\theta V(\mathbf{x}; \theta) &= \mathbb{E}_{\mathbf{y} \sim p_\theta(\cdot|\mathbf{x})} \Big[ r(\mathbf{x}, \mathbf{y}) \nabla_\theta \ln p_\theta(\mathbf{y}|\mathbf{x}) \Big] \\
&= \mathbb{E}_{\mathbf{y} \sim p_\theta(\cdot|\mathbf{x})} \Big[ r(\mathbf{x}, \mathbf{y}) \sum_{l=1}^{L_{out}} \nabla_\theta \ln p_\theta(y_l | \mathbf{x}, \mathbf{y}_{\leq l-1}) \Big] \\
&= \mathbb{E}_{\mathbf{y} \sim p_\theta(\cdot|\mathbf{x})} \Big[ r(\mathbf{x}, \mathbf{y}) \sum_{l=1}^{L_{out}} \nabla_\theta \ln \mathrm{softmax}(f(\mathbf{x}, \mathbf{y}_{\leq l-1}; \theta))_{y_l} \Big] \\
&= \sum_{\mathbf{y} \in \mathcal{X}^{L_{out}}} p_\theta(\mathbf{y}|\mathbf{x}) r(\mathbf{x}, \mathbf{y}) \sum_{l=1}^{L_{out}} J_{f(\mathbf{x}, \mathbf{y}_{\leq l-1}; \theta)}^\top (\mathbf{e}_{y_l} - p_\theta(\cdot|\mathbf{x}, \mathbf{y}_{\leq l-1})),
\end{aligned}
$$

where $\mathbf{e}_{y_l} \in \mathbb{R}^{|\mathcal{X}|}$ denotes the $y_l$'th standard basis vector, *i.e.* the $y_l$'th entry of $\mathbf{e}_{y_l}$ is one and its other entries are zero, and $p_\theta(\cdot|\mathbf{x}, \mathbf{y}_{\leq l-1})$ stands for the vector holding the next-token probabilities, *i.e.* $p_\theta(\cdot|\mathbf{x}, \mathbf{y}_{\leq l-1}) = \mathrm{softmax}(f(\mathbf{x}, \mathbf{y}_{\leq l-1}; \theta)) \in \mathbb{R}^{|\mathcal{X}|}$.

For $c > 0$ to be determined later, denote by $\mathcal{Y}_c$ the set of outputs whose rewards deviate by more than $c$ from the expected reward, *i.e.*:

$$
\mathcal{Y}_c := \big\{ \mathbf{y} \in \mathcal{X}^{L_{out}} : |r(\mathbf{x}, \mathbf{y}) - V(\mathbf{x}; \theta)| > c \big\} .
$$

Defining a modified reward function $\tilde{r} : \mathcal{X}^{L_{in}} \times \mathcal{X}^{L_{out}} \to [-1, 1]$ by:

$$
\tilde{r}(\mathbf{x}, \mathbf{y}) := \begin{cases} r(\mathbf{x}, \mathbf{y}) & , \mathbf{y} \notin \mathcal{Y}_c \\ V(\mathbf{x}; \theta) & , \mathbf{y} \in \mathcal{Y}_c \end{cases} ,
$$

we may write $\nabla_\theta V(\mathbf{x}; \theta)$ as follows:

$$
\begin{aligned}
&\nabla_\theta V(\mathbf{x}; \theta) \\
&= \underbrace{\sum_{\mathbf{y} \in \mathcal{X}^{L_{out}}} p_\theta(\mathbf{y}|\mathbf{x}) \tilde{r}(\mathbf{x}, \mathbf{y}) \sum_{l=1}^{L_{out}} J_{f(\mathbf{x}, \mathbf{y}_{\leq l-1}; \theta)}^\top (\mathbf{e}_{y_l} - p_\theta(\cdot|\mathbf{x}, \mathbf{y}_{\leq l-1}))}_{(I)} \\
&\quad + \underbrace{\sum_{\mathbf{y} \in \mathcal{Y}_c} p_\theta(\mathbf{y}|\mathbf{x})[r(\mathbf{x}, \mathbf{y}) - \tilde{r}(\mathbf{x}, \mathbf{y})] \sum_{l=1}^{L_{out}} J_{f(\mathbf{x}, \mathbf{y}_{\leq l-1}; \theta)}^\top (\mathbf{e}_{y_l} - p_\theta(\cdot|\mathbf{x}, \mathbf{y}_{\leq l-1}))}_{(II)} .
\end{aligned}
$$

We upper bound the Euclidean norms of $(I)$ and $(II)$ separately. Starting with $(II)$, by Chebyshev's inequality we know that:

$$
p_\theta(\mathcal{Y}_c|\mathbf{x}) \leq \frac{\mathrm{STD}_{\mathbf{y} \sim p_\theta(\cdot|\mathbf{x})}[r(\mathbf{x}, \mathbf{y})]^2}{c^2} .
$$

Notice that $\|\mathbf{e}_{y_l} - p_\theta(\cdot|\mathbf{x}, \mathbf{y}_{\leq l-1})\| \leq \|\mathbf{e}_{y_l} - p_\theta(\cdot|\mathbf{x}, \mathbf{y}_{\leq l-1})\|_1 \leq 2$ for all $\mathbf{y}_{\leq l} \in \mathcal{X}^l$, where $\|\cdot\|_1$ denotes the $\ell_1$ norm. Thus, for any $\mathbf{y} \in \mathcal{Y}_c$ and $l \in \{1, \dots, L_{out}\}$:

$$
\begin{aligned}
\big\| J_{f(\mathbf{x}, \mathbf{y}_{\leq l-1}; \theta)}^\top (\mathbf{e}_{y_l} - p_\theta(\cdot|\mathbf{x}, \mathbf{y}_{\leq l-1})) \big\| &\leq \big\| J_{f(\mathbf{x}, \mathbf{y}_{\leq l-1}; \theta)}^\top \big\|_2 \cdot \|(\mathbf{e}_{y_l} - p_\theta(\cdot|\mathbf{x}, \mathbf{y}_{\leq l-1}))\| \\
&\leq 2\gamma(\mathbf{x}; \theta) .
\end{aligned}
$$

Since rewards lie in $[-1, 1]$, by the triangle inequality this implies that:

$$
\|(II)\| \leq 4 L_{out} \gamma(\mathbf{x}; \theta) \cdot p_\theta(\mathcal{Y}_c|\mathbf{x}) \leq 4 L_{out} \gamma(\mathbf{x}; \theta) \cdot \frac{\mathrm{STD}_{\mathbf{y} \sim p_\theta(\cdot|\mathbf{x})}[r(\mathbf{x}, \mathbf{y})]^2}{c^2} . \tag{5}
$$

As for $(I)$, denoting for $\mathbf{y}_{\leq l-1} \in \mathcal{X}^{l-1}$:

$$
\mathbf{a}^{(\mathbf{y}_{\leq l-1})} := \sum_{\mathbf{y}_{\geq l} \in \mathcal{X}^{L_{out}-l+1}} p_\theta(\mathbf{y}_{\geq l}|\mathbf{x}, \mathbf{y}_{\leq l-1}) \tilde{r}(\mathbf{x}, \mathbf{y})(\mathbf{e}_{y_l} - p_\theta(\cdot|\mathbf{x}, \mathbf{y}_{\leq l-1})) \in \mathbb{R}^{|\mathcal{X}|} ,
$$

where $\mathbf{y}_{\geq l} := (y_l, \dots, y_{L_{out}})$, we have that:

$$
(I) = \sum_{l=1}^{L_{out}} \sum_{\mathbf{y}_{\leq l-1} \in \mathcal{X}^{l-1}} p_\theta(\mathbf{y}_{\leq l-1}|\mathbf{x}) J_{f(\mathbf{x}, \mathbf{y}_{\leq l-1}; \theta)}^\top \mathbf{a}^{(\mathbf{y}_{\leq l-1})} .
$$

The $y$'th entry of $\mathbf{a}^{(\mathbf{y}_{\leq l-1})}$ is given by:

$$
\begin{aligned}
a_y^{(\mathbf{y}_{\leq l-1})} &= \sum_{\mathbf{y}_{\geq l+1} \in \mathcal{X}^{L_{out}-l}} p_\theta(\mathbf{y}_{\geq l+1}|\mathbf{x}, \mathbf{y}_{\leq l-1}, y) p_\theta(y|\mathbf{x}, \mathbf{y}_{\leq l-1}) \tilde{r}(\mathbf{x}, \mathbf{y}_{\leq l-1}, y, \mathbf{y}_{\geq l+1}) \\
&\quad - \sum_{\mathbf{y}_{\geq l} \in \mathcal{X}^{L_{out}-l+1}} p_\theta(\mathbf{y}_{\geq l}|\mathbf{x}, \mathbf{y}_{\leq l-1}) p_\theta(y|\mathbf{x}, \mathbf{y}_{\leq l-1}) \tilde{r}(\mathbf{x}, \mathbf{y}) \\
&= p_\theta(y|\mathbf{x}, \mathbf{y}_{\leq l-1}) \Big( \mathbb{E}_{\mathbf{y}_{\geq l+1} \sim p_\theta(\cdot|\mathbf{x}, \mathbf{y}_{\leq l-1}, y)}[\tilde{r}(\mathbf{x}, \mathbf{y}_{\leq l-1}, y, \mathbf{y}_{\geq l+1})] \\
&\quad - \mathbb{E}_{\mathbf{y}_{\geq l} \sim p_\theta(\cdot|\mathbf{x}, \mathbf{y}_{\leq l-1})}[\tilde{r}(\mathbf{x}, \mathbf{y})] \Big) .
\end{aligned}
$$

Since by the definition of $\tilde{r}$ it holds that $|\tilde{r}(\mathbf{x}, \mathbf{y}) - V(\mathbf{x}; \theta)| \leq c$ for all $\mathbf{y} \in \mathcal{X}^{L_{out}}$, we can bound the difference of expectations in the equation above through adding and subtracting $V(\mathbf{x}; \theta)$ and the triangle inequality:

$$
\begin{aligned}
&\Big| \mathbb{E}_{\mathbf{y}_{\geq l+1} \sim p_\theta(\cdot|\mathbf{x}, \mathbf{y}_{\leq l-1}, y)}[\tilde{r}(\mathbf{x}, \mathbf{y}_{\leq l-1}, y, \mathbf{y}_{\geq l+1})] - \mathbb{E}_{\mathbf{y}_{\geq l} \sim p_\theta(\cdot|\mathbf{x}, \mathbf{y}_{\leq l-1})}[\tilde{r}(\mathbf{x}, \mathbf{y})] \Big| \\
&\leq \Big| \mathbb{E}_{\mathbf{y}_{\geq l+1} \sim p_\theta(\cdot|\mathbf{x}, \mathbf{y}_{\leq l-1}, y)}[\tilde{r}(\mathbf{x}, \mathbf{y}_{\leq l-1}, y, \mathbf{y}_{\geq l+1})] - V(\mathbf{x}; \theta) \Big| \\
&\quad + \Big| V(\mathbf{x}; \theta) - \mathbb{E}_{\mathbf{y}_{\geq l} \sim p_\theta(\cdot|\mathbf{x}, \mathbf{y}_{\leq l-1})}[\tilde{r}(\mathbf{x}, \mathbf{y})] \Big| \\
&\leq 2c .
\end{aligned}
$$

Thus:

$$
\big\| \mathbf{a}^{(\mathbf{y}_{\leq l-1})} \big\| \leq \big\| \mathbf{a}^{(\mathbf{y}_{\leq l-1})} \big\|_1 \leq \sum_{y \in \mathcal{X}} p_\theta(y|\mathbf{x}, \mathbf{y}_{\leq l-1}) \cdot 2c = 2c .
$$

Recalling that:

$$
(I) = \sum_{l=1}^{L_{out}} \sum_{\mathbf{y}_{\leq l-1} \in \mathcal{X}^{l-1}} p_\theta(\mathbf{y}_{\leq l-1}|\mathbf{x}) J_{f(\mathbf{x}, \mathbf{y}_{\leq l-1}; \theta)}^\top \mathbf{a}^{(\mathbf{y}_{\leq l-1})} ,
$$

taking the norm of both sides and applying the triangle inequality yields:

$$
\begin{aligned}
\|(I)\| &\leq \sum_{l=1}^{L_{out}} \sum_{\mathbf{y}_{\leq l-1} \in \mathcal{X}^{l-1}} p_\theta(\mathbf{y}_{\leq l-1}|\mathbf{x}) \gamma(\mathbf{x}; \theta) \big\| \mathbf{a}^{(\mathbf{y}_{\leq l-1})} \big\| \\
&\leq \sum_{l=1}^{L_{out}} 2\gamma(\mathbf{x}; \theta) \cdot c \sum_{\mathbf{y}_{\leq l-1} \in \mathcal{X}^{l-1}} p_\theta(\mathbf{y}_{\leq l-1}|\mathbf{x}) \\
&= 2L_{out} \gamma(\mathbf{x}; \theta) \cdot c .
\end{aligned}
\tag{6}
$$

Combining the bounds on the norms of $(I)$ and $(II)$ (Equations (5) and (6)) then leads to:

$$
\|\nabla_\theta V(\mathbf{x}; \theta)\| \leq 4L_{out}\gamma(\mathbf{x}; \theta) \cdot \frac{\mathrm{STD}_{\mathbf{y} \sim p_\theta(\cdot|\mathbf{x})}[r(\mathbf{x}, \mathbf{y})]^2}{c^2} + 2L_{out}\gamma(\mathbf{x}; \theta) \cdot c .
$$

Lastly, choosing $c = \mathrm{STD}_{\mathbf{y} \sim p_\theta(\cdot|\mathbf{x})}[r(\mathbf{x}, \mathbf{y})]^{2/3}$ concludes the proof:

$$
\|\nabla_\theta V(\mathbf{x}; \theta)\| \leq 6L_{out}\gamma(\mathbf{x}; \theta) \cdot \mathrm{STD}_{\mathbf{y} \sim p_\theta(\cdot|\mathbf{x})}[r(\mathbf{x}, \mathbf{y})]^{2/3} .
$$

$\square$

### D.2 PROOF OF PROPOSITION 1

Define:

$$
\mathcal{R}_\delta^+ := \left\{ \mathbf{y} \in \mathcal{X}^{L_{out}} : \frac{p_\theta(\mathbf{y}|\mathbf{x})}{p_{\bar{\theta}}(\mathbf{y}|\mathbf{x})} > 1 + \delta , \ A_{\bar{\theta}}(\mathbf{x}, \mathbf{y}) > 0 \right\} ,
$$

$$
\mathcal{R}_\delta^- := \left\{ \mathbf{y} \in \mathcal{X}^{L_{out}} : \frac{p_\theta(\mathbf{y}|\mathbf{x})}{p_{\bar{\theta}}(\mathbf{y}|\mathbf{x})} < 1 - \delta , \ A_{\bar{\theta}}(\mathbf{x}, \mathbf{y}) < 0 \right\} ,
$$

$$
\mathcal{R}_\delta := \left\{ \mathbf{y} \in \mathcal{X}^{L_{out}} : \left| \frac{p_\theta(\mathbf{y}|\mathbf{x})}{p_{\bar{\theta}}(\mathbf{y}|\mathbf{x})} - 1 \right| > \delta \right\} ,
$$

*i.e.* $\mathcal{R}_\delta^+$ is the set of outputs for which PPO will clip the objective when the probability ratio is strictly greater than $1 + \delta$, $\mathcal{R}_\delta^-$ is the set of outputs for which clipping occurs when the probability ratio is

strictly less than $1 - \delta$, and $\mathcal{R}_\delta$ is the set of outputs for which the probability ratio is either strictly greater than $1 + \delta$ or strictly less than $1 - \delta$. Note that $\mathcal{R}_\delta^+ \cup \mathcal{R}_\delta^- \subseteq \mathcal{R}_\delta$.

We can decompose $V_{\text{PPO}}(\mathbf{x}; \theta)$ according to $\mathcal{R}_\delta^+, \mathcal{R}_\delta^+$, and the remaining outputs as follows:

$$V_{\text{PPO}}(\mathbf{x}; \theta) = \sum\nolimits_{\mathbf{y} \in \mathcal{X}^{L_{out}} \setminus (\mathcal{R}_\delta^+ \cup \mathcal{R}_\delta^-)} p_\theta(\mathbf{y}|\mathbf{x}) A_{\bar\theta}(\mathbf{x}, \mathbf{y})$$
$$+ \sum\nolimits_{\mathbf{y} \in \mathcal{R}_\delta^+} p_{\bar\theta}(\mathbf{y}|\mathbf{x})(1 + \delta) A_{\bar\theta}(\mathbf{x}, \mathbf{y})$$
$$+ \sum\nolimits_{\mathbf{y} \in \mathcal{R}_\delta^-} p_{\bar\theta}(\mathbf{y}|\mathbf{x})(1 - \delta) A_{\bar\theta}(\mathbf{x}, \mathbf{y}).$$

Since the second and third sums in the equation above do not depend on $\theta$ we have that:

$$\nabla_\theta V_{\text{PPO}}(\mathbf{x}; \theta) = \sum\nolimits_{\mathbf{y} \in \mathcal{X}^{L_{out}} \setminus (\mathcal{R}_\delta^+ \cup \mathcal{R}_\delta^-)} A_{\bar\theta}(\mathbf{x}, \mathbf{y}) \nabla_\theta p_\theta(\mathbf{y}|\mathbf{x}).$$

Adding and subtracting $\sum_{\mathbf{y} \in \mathcal{R}_\delta^+ \cup \mathcal{R}_\delta^-} A_{\bar\theta}(\mathbf{x}, \mathbf{y}) \nabla_\theta p_\theta(\mathbf{y}|\mathbf{x})$ yields:

$$\nabla_\theta V_{\text{PPO}}(\mathbf{x}; \theta) = \sum\nolimits_{\mathbf{y} \in \mathcal{X}^{L_{out}}} A_{\bar\theta}(\mathbf{x}, \mathbf{y}) \nabla_\theta p_\theta(\mathbf{y}|\mathbf{x}) - \sum\nolimits_{\mathbf{y} \in \mathcal{R}_\delta^+ \cup \mathcal{R}_\delta^-} A_{\bar\theta}(\mathbf{x}, \mathbf{y}) \nabla_\theta p_\theta(\mathbf{y}|\mathbf{x}). \quad (7)$$

Recall that $A_{\bar\theta}(\mathbf{x}, \mathbf{y}) = r(\mathbf{x}, \mathbf{y}) - V(\mathbf{x}; \bar\theta)$. Because $V(\mathbf{x}; \bar\theta)$ does not depend on $\mathbf{y}$ we get that:

$$\sum\nolimits_{\mathbf{y} \in \mathcal{X}^{L_{out}}} A_{\bar\theta}(\mathbf{x}, \mathbf{y}) \nabla_\theta p_\theta(\mathbf{y}|\mathbf{x}) = \sum\nolimits_{\mathbf{y} \in \mathcal{X}^{L_{out}}} r(\mathbf{x}, \mathbf{y}) \nabla_\theta p_\theta(\mathbf{y}|\mathbf{x}) - V(\mathbf{x}; \bar\theta) \sum\nolimits_{\mathbf{y} \in \mathcal{X}^{L_{out}}} \nabla_\theta p_\theta(\mathbf{y}|\mathbf{x})$$
$$= \nabla_\theta V(\mathbf{x}; \theta) - V(\mathbf{x}; \bar\theta) \nabla_\theta \sum\nolimits_{\mathbf{y} \in \mathcal{X}^{L_{out}}} p_\theta(\mathbf{y}|\mathbf{x})$$
$$= \nabla_\theta V(\mathbf{x}; \theta) - V(\mathbf{x}; \bar\theta) \underbrace{\nabla_\theta 1}_{=0}$$
$$= \nabla_\theta V(\mathbf{x}; \theta).$$

Plugging this into Equation (7), rearranging the equality, and taking the Euclidean norm of both sides, it follows that:

$$\left\| \nabla_\theta V_{\text{PPO}}(\mathbf{x}; \theta) - \nabla_\theta V(\mathbf{x}; \theta) \right\| \leq \left\| \sum\nolimits_{\mathbf{y} \in \mathcal{R}_\delta^+ \cup \mathcal{R}_\delta^-} A_{\bar\theta}(\mathbf{x}, \mathbf{y}) \nabla_\theta p_\theta(\mathbf{y}|\mathbf{x}) \right\|.$$

Rewards lie in the interval $[-1, 1]$, hence, $|A_{\bar\theta}(\mathbf{x}, \mathbf{y})| \leq 2$. So, applying the triangle inequality we obtain:

$$\left\| \nabla_\theta V_{\text{PPO}}(\mathbf{x}; \theta) - \nabla_\theta V(\mathbf{x}; \theta) \right\| \leq 2 \sum\nolimits_{\mathbf{y} \in \mathcal{R}_\delta^+ \cup \mathcal{R}_\delta^-} \left\| \nabla_\theta p_\theta(\mathbf{y}|\mathbf{x}) \right\|$$
$$= 2 \sum\nolimits_{\mathbf{y} \in \mathcal{R}_\delta^+ \cup \mathcal{R}_\delta^-} \left\| p_\theta(\mathbf{y}|\mathbf{x}) \nabla_\theta \ln p_\theta(\mathbf{y}|\mathbf{x}) \right\|$$
$$= 2 \sum\nolimits_{\mathbf{y} \in \mathcal{R}_\delta^+ \cup \mathcal{R}_\delta^-} p_\theta(\mathbf{y}|\mathbf{x}) \cdot \left\| \sum\nolimits_{l=1}^{L_{out}} \nabla_\theta \ln p_\theta(y_l|\mathbf{x}, \mathbf{y}_{\leq l-1}) \right\|,$$

where the first equality is by the log-derivative trick. Since

$$p_\theta(y_l|\mathbf{x}, \mathbf{y}_{\leq l-1}) = \text{softmax}(f(\mathbf{x}, \mathbf{y}_{l-1}; \theta))_{y_l},$$

for any $l \in \{1, \ldots, L_{out}\}$ and $\mathbf{y} \in \mathcal{X}^{L_{out}}$ we have:

$$\nabla_\theta \ln p_\theta(y_l|\mathbf{x}, \mathbf{y}_{\leq l-1}) = J_{f(\mathbf{x}, \mathbf{y}_{\leq l-1}; \theta)}^\top (\mathbf{e}_{y_l} - p_\theta(\cdot|\mathbf{x}, \mathbf{y}_{\leq l-1})),$$

where $\mathbf{e}_{y_l} \in \mathbb{R}^{|\mathcal{X}|}$ denotes the $y_l$'th standard basis vector, *i.e.* the $y_l$'th entry of $\mathbf{e}_{y_l}$ is one and its other entries are zero, and $p_\theta(\cdot|\mathbf{x}, \mathbf{y}_{\leq l-1}) := \text{softmax}(f(\mathbf{x}, \mathbf{y}_{\leq l-1}; \theta)) \in \mathbb{R}^{|\mathcal{X}|}$. Noticing that $\|\mathbf{e}_{y_l} - p_\theta(\cdot|\mathbf{x}, \mathbf{y}_{\leq l-1})\| \leq \|\mathbf{e}_{y_l} - p_\theta(\cdot|\mathbf{x}, \mathbf{y}_{\leq l-1})\|_1 \leq 2$, where $\|\cdot\|_1$ denotes the $\ell_1$ norm, we have:

$$\|\nabla_\theta \ln p_\theta(y_l|\mathbf{x}, \mathbf{y}_{\leq l-1})\| \leq 2\gamma(\mathbf{x}; \theta).$$

By the triangle inequality this implies that:

$$\left\| \nabla_\theta V_{\text{PPO}}(\mathbf{x}; \theta) - \nabla_\theta V(\mathbf{x}; \theta) \right\| \leq 2 \sum\nolimits_{\mathbf{y} \in \mathcal{R}_\delta^+ \cup \mathcal{R}_\delta^-} p_\theta(\mathbf{y}|\mathbf{x}) \cdot \sum\nolimits_{l=1}^{L_{out}} \|\nabla_\theta \ln p_\theta(y_l|\mathbf{x}, \mathbf{y}_{\leq l-1})\|$$
$$\leq 4 L_{out} \gamma(\mathbf{x}; \theta) \sum\nolimits_{\mathbf{y} \in \mathcal{R}_\delta^+ \cup \mathcal{R}_\delta^-} p_\theta(\mathbf{y}|\mathbf{x})$$
$$= 4 L_{out} \gamma(\mathbf{x}; \theta) \cdot p_\theta(\mathcal{R}_\delta^+ \cup \mathcal{R}_\delta^-|\mathbf{x})$$
$$= 4 L_{out} \gamma(\mathbf{x}; \theta) \cdot \left( p_\theta(\mathcal{R}_\delta^+|\mathbf{x}) + p_\theta(\mathcal{R}_\delta^-|\mathbf{x}) \right), \quad (8)$$

where the last transition is due to $\mathcal{R}_\delta^+$ and $\mathcal{R}_\delta^-$ being disjoint. The proof concludes by showing that both $p_\theta(\mathcal{R}_\delta^+|\mathbf{x}) \leq 2\delta^{-1} \cdot \mathrm{TV}(p_\theta(\cdot|\mathbf{x})||p_{\bar{\theta}}(\cdot|\mathbf{x}))$ and $p_\theta(\mathcal{R}_\delta^-|\mathbf{x}) \leq \delta^{-1} \cdot \mathrm{TV}(p_\theta(\cdot|\mathbf{x})||p_{\bar{\theta}}(\cdot|\mathbf{x}))$.

Focusing on $p_\theta(\mathcal{R}_\delta^+|\mathbf{x})$, by the definition of the total variation distance:

$$\left| p_\theta\big(\mathcal{R}_\delta^+|\mathbf{x}\big) - p_{\bar{\theta}}\big(\mathcal{R}_\delta^+|\mathbf{x}\big) \right| \leq \mathrm{TV}(p_\theta(\cdot|\mathbf{x})||p_{\bar{\theta}}(\cdot|\mathbf{x})) \,,$$

from which it follows that:

$$p_\theta\big(\mathcal{R}_\delta^+|\mathbf{x}\big) \cdot \left| 1 - \frac{p_{\bar{\theta}}\big(\mathcal{R}_\delta^+|\mathbf{x}\big)}{p_\theta\big(\mathcal{R}_\delta^+|\mathbf{x}\big)} \right| \leq \mathrm{TV}(p_\theta(\cdot|\mathbf{x})||p_{\bar{\theta}}(\cdot|\mathbf{x})) \,,$$

where note that $p_\theta(\mathcal{R}_\delta^+|\mathbf{x}) \neq 0$ due to the softmax parameterization, under which the probability of any output cannot be equal to zero. For each $\mathbf{y} \in \mathcal{R}_\delta^+$ we know that $p_\theta(\mathbf{y}|\mathbf{x})/p_{\bar{\theta}}(\mathbf{y}|\mathbf{x}) > 1 + \delta$. Thus, $p_\theta(\mathcal{R}_\delta^+|\mathbf{x})/p_{\bar{\theta}}(\mathcal{R}_\delta^+|\mathbf{x}) > 1 + \delta$ and $p_{\bar{\theta}}(\mathcal{R}_\delta^+|\mathbf{x})/p_\theta(\mathcal{R}_\delta^+|\mathbf{x}) < 1 - \frac{\delta}{1+\delta}$. Plugging this into the inequality above, we have that:

$$p_\theta\big(\mathcal{R}_\delta^+|\mathbf{x}\big) \cdot \frac{\delta}{1+\delta} < p_\theta\big(\mathcal{R}_\delta^+|\mathbf{x}\big) \cdot \left| 1 - \frac{p_{\bar{\theta}}\big(\mathcal{R}_\delta^+|\mathbf{x}\big)}{p_\theta\big(\mathcal{R}_\delta^+|\mathbf{x}\big)} \right| \leq \mathrm{TV}(p_\theta(\cdot|\mathbf{x})||p_{\bar{\theta}}(\cdot|\mathbf{x})) \,,$$

and so:

$$p_\theta\big(\mathcal{R}_\delta^+|\mathbf{x}\big) \leq \frac{1+\delta}{\delta} \cdot \mathrm{TV}(p_\theta(\cdot|\mathbf{x})||p_{\bar{\theta}}(\cdot|\mathbf{x})) \leq \frac{2}{\delta} \cdot \mathrm{TV}(p_\theta(\cdot|\mathbf{x})||p_{\bar{\theta}}(\cdot|\mathbf{x})) \,. \tag{9}$$

Turning our attention to $p_\theta(\mathcal{R}_\delta^-|\mathbf{x})$, by an analogous derivation we get:

$$p_\theta\big(\mathcal{R}_\delta^-|\mathbf{x}\big) \cdot \left| 1 - \frac{p_{\bar{\theta}}\big(\mathcal{R}_\delta^-|\mathbf{x}\big)}{p_\theta\big(\mathcal{R}_\delta^-|\mathbf{x}\big)} \right| \leq \mathrm{TV}(p_\theta(\cdot|\mathbf{x})||p_{\bar{\theta}}(\cdot|\mathbf{x})) \,.$$

For each $\mathbf{y} \in \mathcal{R}_\delta^-$ we know that $p_\theta(\mathbf{y}|\mathbf{x})/p_{\bar{\theta}}(\mathbf{y}|\mathbf{x}) < 1 - \delta$, implying that $p_\theta(\mathcal{R}_\delta^-|\mathbf{x})/p_{\bar{\theta}}(\mathcal{R}_\delta^-|\mathbf{x}) < 1 - \delta$ and $p_{\bar{\theta}}(\mathcal{R}_\delta^-|\mathbf{x})/p_\theta(\mathcal{R}_\delta^-|\mathbf{x}) > 1 + \frac{\delta}{1-\delta}$. Consequently:

$$p_\theta\big(\mathcal{R}_\delta^-|\mathbf{x}\big) \cdot \frac{\delta}{1-\delta} < p_\theta\big(\mathcal{R}_\delta^-|\mathbf{x}\big) \cdot \left| 1 - \frac{p_{\bar{\theta}}\big(\mathcal{R}_\delta^-|\mathbf{x}\big)}{p_\theta\big(\mathcal{R}_\delta^-|\mathbf{x}\big)} \right| \leq \mathrm{TV}(p_\theta(\cdot|\mathbf{x})||p_{\bar{\theta}}(\cdot|\mathbf{x})) \,,$$

from which it follows that:

$$p_\theta\big(\mathcal{R}_\delta^-|\mathbf{x}\big) \leq \frac{1-\delta}{\delta} \cdot \mathrm{TV}(p_\theta(\cdot|\mathbf{x})||p_{\bar{\theta}}(\cdot|\mathbf{x})) \leq \frac{1}{\delta} \cdot \mathrm{TV}(p_\theta(\cdot|\mathbf{x})||p_{\bar{\theta}}(\cdot|\mathbf{x})) \,. \tag{10}$$

Finally, combining Equations (8), (9), and (10) establishes the sought-after result:

$$\|\nabla_\theta V_{\mathrm{PPO}}(\mathbf{x};\theta) - \nabla_\theta V(\mathbf{x};\theta)\| \leq 4 L_{out} \gamma(\mathbf{x};\theta) \cdot \big( p_\theta\big(\mathcal{R}_\delta^+|\mathbf{x}\big) + p_\theta\big(\mathcal{R}_\delta^-|\mathbf{x}\big) \big)$$
$$\leq \frac{12 L_{out} \gamma(\mathbf{x};\theta)}{\delta} \cdot \mathrm{TV}(p_\theta(\cdot|\mathbf{x})||p_{\bar{\theta}}(\cdot|\mathbf{x})) \,.$$

$\square$

### D.3  PROOF OF PROPOSITION 2

We may write the PPO-KL objective (Equation (4)) as:

$$V_{\mathrm{PPO-KL}}(\mathbf{x};\theta) = \mathbb{E}_{\mathbf{y} \sim p_{\bar{\theta}}(\cdot|\mathbf{x})} \left[ \frac{p_\theta(\mathbf{y}|\mathbf{x})}{p_{\bar{\theta}}(\mathbf{y}|\mathbf{x})} A_{\bar{\theta}}(\mathbf{x},\mathbf{y}) \right] - \lambda \cdot \mathbb{D}_{\mathrm{KL}}(p_{\bar{\theta}}(\cdot|\mathbf{x})||p_\theta(\cdot|\mathbf{x}))$$
$$= \mathbb{E}_{\mathbf{y} \sim p_\theta(\cdot|\mathbf{x})}[A_{\bar{\theta}}(\mathbf{x},\mathbf{y})] - \lambda \cdot \mathbb{D}_{\mathrm{KL}}(p_{\bar{\theta}}(\cdot|\mathbf{x})||p_\theta(\cdot|\mathbf{x})) \,.$$

Since $A_{\bar{\theta}}(\mathbf{x},\mathbf{y}) = r(\mathbf{x},\mathbf{y}) - V(\mathbf{x};\bar{\theta})$, it holds that $\mathbb{E}_{\mathbf{y} \sim p_\theta(\cdot|\mathbf{x})}[A_{\bar{\theta}}(\mathbf{x},\mathbf{y})] = V(\mathbf{x};\theta) - V(\mathbf{x};\bar{\theta})$, where note that $V(\mathbf{x};\bar{\theta})$ does not depend on $\theta$. Differentiating with respect to $\theta$ thus yields:

$$\nabla_\theta V_{\mathrm{PPO-KL}}(\mathbf{x};\theta) = \nabla_\theta V(\mathbf{x};\theta) - \lambda \cdot \nabla_\theta \mathbb{D}_{\mathrm{KL}}(p_{\bar{\theta}}(\cdot|\mathbf{x})||p_\theta(\cdot|\mathbf{x})) \,,$$

and so:
$$\|\nabla_\theta V_{\text{PPO}-\text{KL}}(\mathbf{x};\theta) - \nabla_\theta V(\mathbf{x};\theta)\| = \lambda \cdot \|\nabla_\theta \mathbb{D}_{\text{KL}}(p_{\bar\theta}(\cdot|\mathbf{x})||p_\theta(\cdot|\mathbf{x}))\|.$$

We therefore need only show that $\|\nabla_\theta \mathbb{D}_{\text{KL}}(p_{\bar\theta}(\cdot|\mathbf{x})||p_\theta(\cdot|\mathbf{x}))\|$ is upper bounded by $4L_{out}\gamma(\mathbf{x};\theta) \cdot$ $\text{TV}(p_\theta(\cdot|\mathbf{x})||p_{\bar\theta}(\cdot|\mathbf{x}))$. Writing the KL divergence term explicitly:

$$\nabla_\theta \mathbb{D}_{\text{KL}}(p_{\bar\theta}(\cdot|\mathbf{x})||p_\theta(\cdot|\mathbf{x})) = \nabla_\theta \sum_{\mathbf{y}\in\mathcal{X}^{L_{out}}} p_{\bar\theta}(\mathbf{y}|\mathbf{x}) \ln \frac{p_{\bar\theta}(\mathbf{y}|\mathbf{x})}{p_\theta(\mathbf{y}|\mathbf{x})}$$
$$= -\nabla_\theta \sum_{\mathbf{y}\in\mathcal{X}^{L_{out}}} p_{\bar\theta}(\mathbf{y}|\mathbf{x}) \ln p_\theta(\mathbf{y}|\mathbf{x})$$
$$+ \underbrace{\nabla_\theta \sum_{\mathbf{y}\in\mathcal{X}^{L_{out}}} p_{\bar\theta}(\mathbf{y}|\mathbf{x}) \ln p_{\bar\theta}(\mathbf{y}|\mathbf{x})}_{=0}$$
$$= -\sum_{\mathbf{y}\in\mathcal{X}^{L_{out}}} p_{\bar\theta}(\mathbf{y}|\mathbf{x}) \sum_{l=1}^{L_{out}} \nabla_\theta \ln p_\theta(y_l|\mathbf{x},\mathbf{y}_{\leq l-1}).$$

For any $l \in \{1,\dots,L_{out}\}$ and $\mathbf{y} \in \mathcal{X}^{L_{out}}$ we have that:

$$\nabla_\theta \ln p_\theta(y_l|\mathbf{x},\mathbf{y}_{\leq l-1}) = J^\top_{f(\mathbf{x},\mathbf{y}_{\leq l-1};\theta)}(\mathbf{e}_{y_l} - p_\theta(\cdot|\mathbf{x},\mathbf{y}_{\leq l-1})),$$

where $\mathbf{e}_{y_l} \in \mathbb{R}^{|\mathcal{X}|}$ denotes the $y_l$'th standard basis vector, *i.e.* the $y_l$'th entry of $\mathbf{e}_{y_l}$ is one and its other entries are zero, and $p_\theta(\cdot|\mathbf{x},\mathbf{y}_{\leq l-1}) := \text{softmax}(f(\mathbf{x},\mathbf{y}_{\leq l-1};\theta)) \in \mathbb{R}^{|\mathcal{X}|}$. Thus, a straightforward computation shows:

$$\nabla_\theta \mathbb{D}_{\text{KL}}(p_{\bar\theta}(\cdot|\mathbf{x})||p_\theta(\cdot|\mathbf{x}))$$
$$= -\sum_{l=1}^{L_{out}} \sum_{\mathbf{y}_{\leq l-1}\in\mathcal{X}^{l-1}} p_{\bar\theta}(\mathbf{y}_{\leq l-1}|\mathbf{x}) \cdot J^\top_{f(\mathbf{x},\mathbf{y}_{\leq l-1};\theta)}(p_{\bar\theta}(\cdot|\mathbf{x},\mathbf{y}_{\leq l-1}) - p_\theta(\cdot|\mathbf{x},\mathbf{y}_{\leq l-1})).$$

Then, by the triangle inequality and the fact that:

$$\left\| J^\top_{f(\mathbf{x},\mathbf{y}_{\leq l-1};\theta)}(p_{\bar\theta}(\cdot|\mathbf{x},\mathbf{y}_{\leq l-1}) - p_\theta(\cdot|\mathbf{x},\mathbf{y}_{\leq l-1})) \right\|$$
$$\leq \left\| J^\top_{f(\mathbf{x},\mathbf{y}_{\leq l-1};\theta)} \right\|_2 \cdot \|(p_{\bar\theta}(\cdot|\mathbf{x},\mathbf{y}_{\leq l-1}) - p_\theta(\cdot|\mathbf{x},\mathbf{y}_{\leq l-1}))\|$$
$$\leq \gamma(\mathbf{x};\theta) \cdot \|(p_{\bar\theta}(\cdot|\mathbf{x},\mathbf{y}_{\leq l-1}) - p_\theta(\cdot|\mathbf{x},\mathbf{y}_{\leq l-1}))\|_1,$$

where $\|\cdot\|_1$ denotes the $\ell_1$ norm, we obtain:

$$\|\nabla_\theta \mathbb{D}_{\text{KL}}(p_{\bar\theta}(\cdot|\mathbf{x})||p_\theta(\cdot|\mathbf{x}))\|$$
$$\leq \gamma(\mathbf{x};\theta) \sum_{l=1}^{L_{out}} \sum_{\mathbf{y}_{\leq l-1}\in\mathcal{X}^{l-1}} p_{\bar\theta}(\mathbf{y}_{\leq l-1}|\mathbf{x}) \cdot \|p_{\bar\theta}(\cdot|\mathbf{x},\mathbf{y}_{\leq l-1}) - p_\theta(\cdot|\mathbf{x},\mathbf{y}_{\leq l-1})\|_1. \tag{11}$$

Notice that for any $l \in \{1,\dots,L_{out}\}$ and $\mathbf{y}_{\leq l-1} \in \mathcal{X}^{l-1}$:

$$p_{\bar\theta}(\mathbf{y}_{\leq l-1}|\mathbf{x}) \cdot \|p_{\bar\theta}(\cdot|\mathbf{x},\mathbf{y}_{\leq l-1}) - p_\theta(\cdot|\mathbf{x},\mathbf{y}_{\leq l-1})\|_1$$
$$= p_{\bar\theta}(\mathbf{y}_{\leq l-1}|\mathbf{x}) \sum_{y_l\in\mathcal{X}} |p_{\bar\theta}(y_l|\mathbf{x},\mathbf{y}_{\leq l-1}) - p_\theta(y_l|\mathbf{x},\mathbf{y}_{\leq l-1})|$$
$$= \sum_{y_l\in\mathcal{X}} |p_{\bar\theta}(\mathbf{y}_{\leq l}|\mathbf{x}) - p_{\bar\theta}(\mathbf{y}_{\leq l-1}|\mathbf{x})p_\theta(y_l|\mathbf{x},\mathbf{y}_{\leq l-1})|$$
$$\leq \sum_{y_l\in\mathcal{X}} \Big[ |p_{\bar\theta}(\mathbf{y}_{\leq l}|\mathbf{x}) - p_\theta(\mathbf{y}_{\leq l}|\mathbf{x})| + p_\theta(y_l|\mathbf{x},\mathbf{y}_{\leq l-1}) \cdot |p_\theta(\mathbf{y}_{\leq l-1}|\mathbf{x}) - p_{\bar\theta}(\mathbf{y}_{\leq l-1}|\mathbf{x})| \Big]$$
$$= |p_\theta(\mathbf{y}_{\leq l-1}|\mathbf{x}) - p_{\bar\theta}(\mathbf{y}_{\leq l-1}|\mathbf{x})| + \sum_{y_l\in\mathcal{X}} |p_{\bar\theta}(\mathbf{y}_{\leq l}|\mathbf{x}) - p_\theta(\mathbf{y}_{\leq l}|\mathbf{x})|,$$

where the penultimate transition is by adding and subtracting $p_\theta(\mathbf{y}_{\leq l-1}|\mathbf{x})p_\theta(y_l|\mathbf{x},\mathbf{y}_{\leq l-1})$ in each summand and the triangle inequality. Summing over $\mathbf{y}_{\leq l-1} \in \mathcal{X}^{l-1}$ we thus get:

$$\sum_{\mathbf{y}_{\leq l-1}\in\mathcal{X}^{l-1}} p_{\bar\theta}(\mathbf{y}_{\leq l-1}|\mathbf{x}) \cdot \|p_{\bar\theta}(\cdot|\mathbf{x},\mathbf{y}_{\leq l-1}) - p_\theta(\cdot|\mathbf{x},\mathbf{y}_{\leq l-1})\|_1$$
$$\leq \sum_{\mathbf{y}_{\leq l-1}\in\mathcal{X}^{l-1}} |p_\theta(\mathbf{y}_{\leq l-1}|\mathbf{x}) - p_{\bar\theta}(\mathbf{y}_{\leq l-1}|\mathbf{x})| + \sum_{\mathbf{y}_{\leq l}\in\mathcal{X}^l} |p_{\bar\theta}(\mathbf{y}_{\leq l}|\mathbf{x}) - p_\theta(\mathbf{y}_{\leq l}|\mathbf{x})|.$$

Both sums in right hand side of the inequality above are equal to twice the total variation distance between marginal distributions of $p_\theta(\cdot|\mathbf{x})$ and $p_{\bar\theta}(\cdot|\mathbf{x})$. Hence, each one of them is upper bounded by $2\mathrm{TV}(p_\theta(\cdot|\mathbf{x})||p_{\bar\theta}(\cdot|\mathbf{x}))$, *i.e.*:

$$\sum\nolimits_{\mathbf{y}_{\le l-1}\in\mathcal{X}^{l-1}} p_{\bar\theta}(\mathbf{y}_{\le l-1}|\mathbf{x}) \cdot \|p_{\bar\theta}(\cdot|\mathbf{x},\mathbf{y}_{\le l-1}) - p_\theta(\cdot|\mathbf{x},\mathbf{y}_{\le l-1})\|_1 \le 4\mathrm{TV}(p_\theta(\cdot|\mathbf{x})||p_{\bar\theta}(\cdot|\mathbf{x})).$$

Going back to Equation (11):

$$\|\nabla_\theta\mathbb{D}_{\mathrm{KL}}(p_{\bar\theta}(\cdot|\mathbf{x})||p_\theta(\cdot|\mathbf{x}))\| \le 4L_{out}\gamma(\mathbf{x};\theta) \cdot \mathrm{TV}(p_\theta(\cdot|\mathbf{x})||p_{\bar\theta}(\cdot|\mathbf{x})),$$

from which it readily follows that

$$\begin{aligned}\|\nabla_\theta V_{\mathrm{PPO-KL}}(\mathbf{x};\theta) - \nabla_\theta V(\mathbf{x};\theta)\| &\le \lambda \cdot \|\nabla_\theta\mathbb{D}_{\mathrm{KL}}(p_{\bar\theta}(\cdot|\mathbf{x})||p_\theta(\cdot|\mathbf{x}))\| \\ &\le 4\lambda L_{out}\gamma(\mathbf{x};\theta) \cdot \mathrm{TV}(p_\theta(\cdot|\mathbf{x})||p_{\bar\theta}(\cdot|\mathbf{x})).\end{aligned}$$

$\square$

### D.4 PROOF OF THEOREM 2

The proof is sectioned into two parts. We first derive the lower bound on the time until correct classification of $\mathbf{x}_n$ for RFT, after which we prove the upper bound for SFT. For simplicity, we assume without loss of generality that $\mathbf{x}_1 = \mathbf{e}_1, \ldots, \mathbf{x}_N = \mathbf{e}_N$, with $\mathbf{e}_1, \ldots, \mathbf{e}_N$ being the first $N$ standard basis vectors of $\mathbb{R}^D$. The analysis for arbitrary orthonormal inputs $\mathbf{x}_1, \ldots, \mathbf{x}_N$ follows from analogous arguments, by tracking the evolution of inner products between rows of $\theta$ and $\mathbf{x}_1, \ldots, \mathbf{x}_N$ instead of individual entries of $\theta$.

#### D.4.1 OPTIMIZATION TIME LOWER BOUND FOR RFT

Denote by $w_{\mathbf{y},d}(t) \in \mathbb{R}$ the $(\mathbf{y},d)$'th entry of $\theta_{\mathrm{RFT}}(t)$, for $\mathbf{y} \in \{1,\ldots,K\}, d \in \{1,\ldots,D\}$. Under this notation, since $\mathbf{x}_{n'} = \mathbf{e}_{n'}$ we have that $f(\mathbf{x}_{n'};\theta_{\mathrm{RFT}}(t)) = \theta_{\mathrm{RFT}}(t)\mathbf{x}_{n'} = (w_{1,n'}(t), \ldots, w_{K,n'}(t))^\top$ for $n' \in \{1,\ldots,N\}$. Now, recalling that $r(\mathbf{x}_{n'},\mathbf{y}_{n'}) = 1$ and $r(\mathbf{x}_{n'},\mathbf{y}) = -1$ for all $\mathbf{y} \ne \mathbf{y}_{n'}$, the expected reward at time $t \ge 0$ can be written as:

$$\begin{aligned}V(\theta_{\mathrm{RFT}}(t)) &= \frac{1}{N}\sum\nolimits_{n'=1}^N \sum\nolimits_{\mathbf{y}=1}^K p_{\theta_{\mathrm{RFT}}(t)}(\mathbf{y}|\mathbf{x}_{n'})r(\mathbf{x}_{n'},\mathbf{y}) \\ &= \frac{1}{N}\sum\nolimits_{n'=1}^N\big[p_{\theta_{\mathrm{RFT}}(t)}(\mathbf{y}_{n'}|\mathbf{x}_{n'}) - \big(1 - p_{\theta_{\mathrm{RFT}}(t)}(\mathbf{y}_{n'}|\mathbf{x}_{n'})\big)\big] \\ &= \frac{1}{N}\sum\nolimits_{n'=1}^N\left[\frac{\exp(w_{\mathbf{y}_{n'},n'}(t))}{\sum_{\mathbf{y}=1}^K\exp(w_{\mathbf{y},n'}(t))} - \left(1 - \frac{\exp(w_{\mathbf{y}_{n'},n'}(t))}{\sum_{\mathbf{y}=1}^K\exp(w_{\mathbf{y},n'}(t))}\right)\right] \\ &= \frac{1}{N}\sum\nolimits_{n'=1}^N\left[\frac{2\exp(w_{\mathbf{y}_{n'},n'}(t))}{\sum_{\mathbf{y}=1}^K\exp(w_{\mathbf{y},n'}(t))} - 1\right].\end{aligned}$$

By introducing the shorthand $z_{n'}(t) := \sum_{\mathbf{y}=1}^K\exp(w_{\mathbf{y},n'}(t))$, we have that:

$$V(\theta_{\mathrm{RFT}}(t)) = \frac{1}{N}\sum\nolimits_{n'=1}^N\left[\frac{2\exp(w_{\mathbf{y}_{n'},n'}(t))}{z_{n'}(t)} - 1\right].$$

Focusing on $\mathbf{x}_n = \mathbf{e}_n$, under gradient flow the parameters $w_{1,n}(t), \ldots, w_{K,n}(t)$, which determine the output distribution given $\mathbf{x}_n$, obey the following differential equations:

$$\frac{d}{dt}w_{\mathbf{y},n}(t) = \nabla_{w_{\mathbf{y},n}}V(\theta_{\mathrm{RFT}}(t)) = \begin{cases}\frac{2}{N}\left[\frac{\exp(w_{\mathbf{y}_n,n}(t))}{z_n(t)}\left(1 - \frac{\exp(w_{\mathbf{y}_n,n}(t))}{z_n(t)}\right)\right] &, \mathbf{y} = \mathbf{y}_n \\ -\frac{2}{N}\cdot\frac{\exp(w_{\mathbf{y}_n,n}(t))\exp(w_{\mathbf{y},n}(t))}{z_n(t)^2} &, \mathbf{y} \ne \mathbf{y}_n\end{cases}, \quad (12)$$

for all $\mathbf{y} \in \{1,\ldots,K\}$. As can be seen from Equation (12), the evolution of $w_{1,n}(t), \ldots, w_{K,n}(t)$ is independent of the remaining parameters (this is a result of the orthogonality assumption on $\mathbf{x}_1, \ldots, \mathbf{x}_N$). Furthermore, the differential equation governing each parameter $w_{\mathbf{y},n}(t)$ corresponding to an incorrect label $\mathbf{y} \ne \mathbf{y}_n$ is the same. Since, by assumption, the softmax probabilities of the incorrect labels for $\mathbf{x}_n$ are equal under $\theta_{\mathrm{RFT}}(0)$, we know that $w_{\mathbf{y},n}(0) = w_{\mathbf{y}',n}(0)$ for all

$\mathbf{y}, \mathbf{y}' \neq \mathbf{y}_n$. Hence, the uniqueness of the solution to Equation (12) for an initial condition implies that $w_{\mathbf{y},n}(t) = w_{\mathbf{y}',n}(t)$ for all $t \geq 0$ and $\mathbf{y}, \mathbf{y}' \neq \mathbf{y}_n$. Meaning, the parameters corresponding to incorrect labels of $\mathbf{x}_n$ are equal throughout optimization.

Given this observation, we may denote $v(t) := w_{\mathbf{y},n}(t)$ for any $\mathbf{y} \neq \mathbf{y}_n$. The dynamics for the $n$'th column of $\theta_{\mathrm{RFT}}(t)$ thus boil down to:

$$\frac{d}{dt}w_{\mathbf{y}_n,n}(t) = \frac{2(K-1)\exp(w_{\mathbf{y}_n,n}(t) + v(t))}{N[\exp(w_{\mathbf{y}_n,n}(t)) + (K-1)\exp(v(t))]^2},$$

$$\frac{d}{dt}v(t) = -\frac{2\exp(w_{\mathbf{y}_n,n}(t) + v(t))}{N[\exp(w_{\mathbf{y}_n,n}(t)) + (K-1)\exp(v(t))]^2},$$
(13)

where we used the fact that $z(t) = \exp(w_{\mathbf{y}_n,n}(t)) + (K-1)\exp(v(t))$.

Let $\mu(t) := w_{\mathbf{y}_n,n}(t) - v(t)$. Dividing the nominator and denominator of the terms on the right hand side in Equation (13) by $\exp(2v(t))$, we may describe the dynamics of $\mu(t)$ through a single differential equation:

$$\frac{d}{dt}\mu(t) = \frac{d}{dt}w_{\mathbf{y}_n,n}(t) - \frac{d}{dt}v(t) = \frac{2K\exp(\mu(t))}{N[\exp(\mu(t)) + K - 1]^2}.$$

Equivalently, this can be written as:

$$\frac{d}{dt}\mu(t) = \frac{2K}{N[\exp(\mu(t)) + 2(K-1) + (K-1)^2\exp(-\mu(t))]}.$$
(14)

It can be verified by differentiation that the solution to the above differential equation satisfies:

$$\frac{2K}{N}t + \exp(\mu(0)) + 2(K-1)\mu(0) - (K-1)^2\exp(-\mu(0))$$
$$= \exp(\mu(t)) + 2(K-1)\mu(t) - (K-1)^2\exp(-\mu(t)).$$

Notice that by Equation (14) we have $\frac{d}{dt}\mu(t) > 0$, and so $\mu(t)$ is monotonically increasing. Since $\theta^{(0)}$ misclassifies $\mathbf{x}_n$, *i.e.* $\mu(0) < 0$, this implies that $t_{\mathrm{RFT}}$ — the initial time at which RFT classifies $\mathbf{x}_n$ correctly — is the unique time at which $\mu(t) = 0$. Such a time must exist since $\frac{d}{dt}\mu(t)$ is bounded away from zero when $\mu(t) \in [\mu(0), 0]$. It therefore follows that:

$$t_{\mathrm{RFT}} = \frac{N(K-1)^2}{2K}\exp(-\mu(0)) - \frac{N(K-1)^2}{2K} + \frac{N}{2K}(1 - \exp(\mu(0))) - \frac{N(K-1)}{K}\mu(0)$$
$$\geq \frac{N(K-1)^2}{2K} \cdot (\exp(-\mu(0)) - 1),$$
(15)

where the inequality is due to $\mu(0) < 0$ and $1 - \exp(\mu(0)) > 0$.

Now, recalling that $\exp(w_{\mathbf{y}_n,n}(t))/z_n(t)$ is the probability for outputing the correct class, which receives a reward of 1, while incorrect labels receive a reward of $-1$, the standard deviation of the reward for $\mathbf{x}_n$ at time $t$ is given by:

$$\mathrm{STD}_{\mathbf{y} \sim p_{\theta_{\mathrm{RFT}}(t)}(\mathbf{y}|\mathbf{x}_n)}[r(\mathbf{x}_n, \mathbf{y})] = \sqrt{1 - \left[\frac{\exp(w_{\mathbf{y}_n,n}(t))}{z_n(t)} - \left(1 - \frac{\exp(w_{\mathbf{y}_n,n}(t))}{z_n(t)}\right)\right]^2}$$
$$= \sqrt{\frac{4(K-1)\exp(w_{\mathbf{y}_n,n}(t) + v(t))}{z_n(t)^2}}$$
$$= \sqrt{\frac{4(K-1)}{\exp(\mu(t)) + 2(K-1) + (K-1)^2\exp(-\mu(t))}},$$

where the first transition is due to fact that the standard deviation of a random variable that is 1 with probability $q$ and $-1$ otherwise is $\sqrt{1 - (q - (1-q))^2}$, and the last transition is by dividing the nominator and denominator of the fraction by $\exp(w_{\mathbf{y}_n,n}(t) + v(t))$ and plugging in $\mu(t) = w_{\mathbf{y}_n,n}(t) - v(t)$. Specifically, at time $t = 0$:

$$\frac{1}{\mathrm{STD}_{\mathbf{y} \sim p_{\theta^{(0)}}(\mathbf{y}|\mathbf{x}_n)}[r(\mathbf{x}_n, \mathbf{y})]} = \sqrt{\frac{\exp(\mu(0)) + 2(K-1) + (K-1)^2\exp(-\mu(0))}{4(K-1)}}.$$
(16)

Applying $\exp(\mu(0)) < 1 < 2(K-1)$ and squaring both sides of the inequality leads to:

$$\frac{1}{\text{STD}_{\mathbf{y} \sim p_{\theta(0)}(\mathbf{y}|\mathbf{x}_n)}[r(\mathbf{x}_n, \mathbf{y})]^2} < 1 + \frac{(K-1)}{4}\exp(-\mu(0)),$$

which implies:

$$\exp(-\mu(0)) > \frac{4}{K-1} \cdot \left( \frac{1}{\text{STD}_{\mathbf{y} \sim p_{\theta(0)}(\mathbf{y}|\mathbf{x}_n)}[r(\mathbf{x}_n, \mathbf{y})]^2} - 1 \right).$$

Combining the inequality above with Equation (15), we may conclude:

$$\begin{aligned} t_{\text{RFT}} &\geq \frac{2N(K-1)}{K} \cdot \left( \frac{1}{\text{STD}_{\mathbf{y} \sim p_{\theta(0)}(\cdot|\mathbf{x}_n)}[r(\mathbf{x}_n, \mathbf{y})]^2} - 1 - \frac{K-1}{4} \right) \\ &= \Omega\left( \frac{1}{\text{STD}_{\mathbf{y} \sim p_{\theta(0)}(\cdot|\mathbf{x}_n)}[r(\mathbf{x}_n, \mathbf{y})]^2} \right). \end{aligned}$$

### D.4.2 OPTIMIZATION TIME UPPER BOUND FOR SFT

Similarly to the analysis for RFT, we denote by $w_{\mathbf{y},d}(t) \in \mathbb{R}$ the $(\mathbf{y}, d)$'th entry of $\theta_{\text{SFT}}(t)$, for $\mathbf{y} \in \{1, \ldots, K\}, d \in \{1, \ldots, D\}$, and let $z_{n'}(t) = \sum_{\mathbf{y}=1}^{K} \exp(w_{\mathbf{y},n'}(t))$ for $n' \in \{1, \ldots, N\}$. Under this notation, the SFT loss is:

$$\begin{aligned} \mathcal{L}_{\text{SFT}}(\theta_{\text{SFT}}(t)) &= -\frac{1}{N}\sum_{n'=1}^{N}\ln\big(p_{\theta_{\text{SFT}}(t)}(\mathbf{y}_n|\mathbf{x}_{n'})\big) \\ &= -\frac{1}{N}\sum_{n'=1}^{N}\ln\left( \frac{\exp(w_{\mathbf{y}_{n'},n'}(t))}{z_{n'}(t)} \right). \end{aligned}$$

Focusing on $\mathbf{x}_n = \mathbf{e}_n$, the parameters $w_{1,n}(t), \ldots, w_{K,n}(t)$, which determine the output distribution given $\mathbf{x}_n$, obey the following differential equations:

$$\frac{d}{dt}w_{\mathbf{y},n}(t) = -\nabla_{w_{\mathbf{y},n}}\mathcal{L}_{\text{SFT}}(\theta_{\text{SFT}}(t)) = \begin{cases} \frac{1}{N}\left( 1 - \frac{\exp(w_{\mathbf{y}_n,n}(t))}{z_n(t)} \right) & , \mathbf{y} = \mathbf{y}_n \\ -\frac{1}{N} \cdot \frac{\exp(w_{\mathbf{y},n}(t))}{z_n(t)} & , \mathbf{y} \neq \mathbf{y}_n \end{cases}, \quad (17)$$

for all $\mathbf{y} \in \{1, \ldots, K\}$. As in RFT, the evolution of $w_{1,n}(t), \ldots, w_{K,n}(t)$ is independent of the remaining parameters due to the orthogonality assumption on $\mathbf{x}_1, \ldots, \mathbf{x}_N$. Furthermore, every $w_{\mathbf{y},n}(t)$ corresponding to an incorrect label $\mathbf{y} \neq \mathbf{y}_n$ is governed by the same differential equation, and, by assumption, initially $w_{\mathbf{y},n}(0) = w_{\mathbf{y}',n}(0)$ for all $\mathbf{y}, \mathbf{y}' \neq \mathbf{y}_n$. As a result, the uniqueness of the solution to Equation (17) for an initial condition implies that $w_{\mathbf{y},n}(t) = w_{\mathbf{y}',n}(t)$ for all $t \geq 0$ and $\mathbf{y}, \mathbf{y}' \neq \mathbf{y}_n$.

Denoting $v(t) := w_{\mathbf{y},n}(t)$ for some $\mathbf{y} \neq \mathbf{y}_n$, the dynamics for the $n$'th column of $\theta_{\text{SFT}}(t)$ are therefore given by:

$$\begin{aligned} \frac{d}{dt}w_{\mathbf{y}_n,n}(t) &= \frac{(K-1)\exp(v(t) - w_{\mathbf{y}_n,n}(t))}{N[1 + (K-1)\exp(v(t) - w_{\mathbf{y}_n,n}(t))]}, \\ \frac{d}{dt}v(t) &= -\frac{\exp(v(t) - w_{\mathbf{y}_n,n}(t))}{N[1 + (K-1)\exp(v(t) - w_{\mathbf{y}_n,n}(t))]}, \end{aligned} \quad (18)$$

Let $\mu(t) := w_{\mathbf{y}_n,n}(t) - v(t)$. We can express the dynamics of $\mu(t)$ through:

$$\frac{d}{dt}\mu(t) = \frac{d}{dt}w_{\mathbf{y}_n,n}(t) - \frac{d}{dt}v(t) = \frac{K}{N[\exp(\mu(t)) + (K-1)]},$$

for which it can be verified by differentiation that the unique solution satisfies:

$$\frac{K}{N}t + (K-1)\mu(0) + \exp(\mu(0)) = (K-1)\mu(t) + \exp(\mu(t)). \quad (19)$$

We are now in a position to extract from Equation (19) an upper bound on $t_{\text{SFT}}$ — the initial time at which SFT classifies $\mathbf{x}_n$ correctly. Notice that $\frac{d}{dt}\mu(t) > 0$, and so $\mu(t)$ is monotonically increasing. Since $\theta^{(0)}$ misclassifies $\mathbf{x}_n$, *i.e.* $\mu(0) < 0$, this implies that $t_{\text{SFT}}$ is the unique time at which $\mu(t) = 0$. Note that such a time must exist since $\frac{d}{dt}\mu(t)$ is bounded away from zero when $\mu(t) \in [\mu(0), 0]$. Thus, from Equation (19) it follows that:

$$
\begin{aligned}
t_{\text{SFT}} &= \frac{N[1 - (K-1)\mu(0) - \exp(\mu(0))]}{K} \\
&< \frac{N}{K} - \frac{N(K-1)}{K}\mu(0)\,.
\end{aligned}
\tag{20}
$$

where the inequality is due to $1 - \exp(\mu(0)) < 1$. Recall from Equation (16) that:

$$
\begin{aligned}
\frac{1}{\text{STD}_{\mathbf{y} \sim p_{\theta^{(0)}}(\mathbf{y}|\mathbf{x}_n)}[r(\mathbf{x}_n, \mathbf{y})]} &= \sqrt{\frac{\exp(\mu(0)) + 2(K-1) + (K-1)^2 \exp(-\mu(0))}{4(K-1)}} \\
&\geq \sqrt{\frac{(K-1)\exp(-\mu(0))}{4}}\,.
\end{aligned}
$$

Taking the natural logarithm of both sides and rearranging the inequality yields:

$$
-\mu(0) \leq \ln\left(\frac{4}{(K-1)\,\text{STD}_{\mathbf{y} \sim p_{\theta^{(0)}}(\mathbf{y}|\mathbf{x}_n)}[r(\mathbf{x}_n, \mathbf{y})]^2}\right)\,.
$$

Hence, combined with Equation (20), we arrive at the desired result:

$$
\begin{aligned}
t_{\text{SFT}} &< \frac{N}{K} + \frac{N(K-1)}{K} \cdot \ln\left(\frac{4}{(K-1)\,\text{STD}_{\mathbf{y} \sim p_{\theta^{(0)}}(\mathbf{y}|\mathbf{x}_n)}[r(\mathbf{x}_n, \mathbf{y})]^2}\right) \\
&= \mathcal{O}\left(\ln\left(\frac{1}{\text{STD}_{\mathbf{y} \sim p_{\theta^{(0)}}(\cdot|\mathbf{x}_n)}[r(\mathbf{x}_n, \mathbf{y})]}\right)\right)\,.
\end{aligned}
$$

$\square$

# E  EXAMPLES OF INPUTS WITH SMALL AND INPUTS WITH LARGE REWARD STANDARD DEVIATION UNDER THE PRETRAINED LANGUAGE MODEL

Figures 5, 6, and 7 give examples from the NarrativeQA, ToTTo, and CommonGen datasets, respectively, of inputs with small and inputs with large pretrain reward standard deviation. We also include outputs produced by the pretrained, RFT, and SFT models along with their rewards. The examples showcase how: *(i)* the pretrained model often generates continuations for the input text instead of obliging to the task at hand; and *(ii)* RFT struggles to maximize the reward of inputs with a small pretrain reward standard deviation compared to those with a large pretrain reward standard deviation.

# F  FURTHER EXPERIMENTS AND IMPLEMENTATION DETAILS

## F.1  FURTHER EXPERIMENTS

Listed below are additional experiments, omitted from Sections 4 and 5.

**Vanishing gradients in the GRUE benchmark (Section 4.1):**

- Figure 8 supplements Figure 1, by plotting the reward mean and standard deviation of train samples, under the pretrained, RFT, and SFT models, for the remaining GRUE datasets: CommonGen, IWSLT 2017, CNN/Daily Mail, and DailyDialog.
- Table 3 supplements Table 1 by reporting, for the remaining GRUE datasets (see above), the Pearson correlation between the reward standard deviation under the pretrained model and the absolute reward mean change due to finetuning, across the train samples.

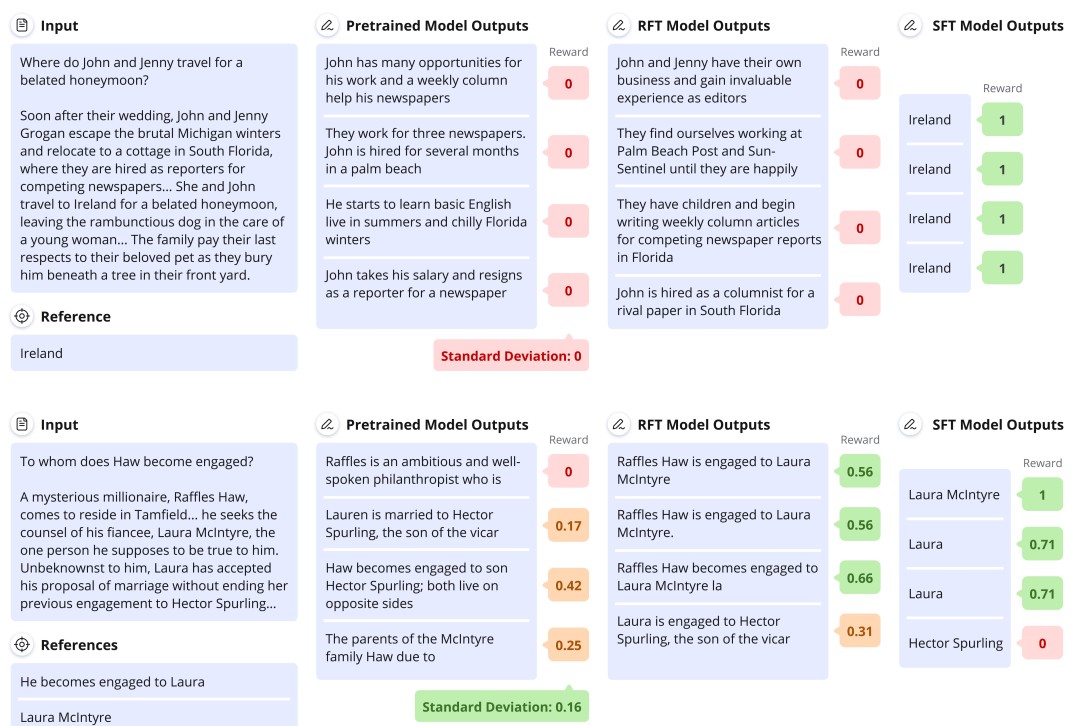

Figure 5: Representative example of an input with small (top) and an input with large (bottom) reward standard deviation under the pretrained model from the NarrativeQA dataset.

- Figure 9 supplements Figure 2, by showing that the correlation between the pretrain reward standard deviation percentile and the reward difference between RFT and SFT is not sensitive to the choice of percentile.
- Table 4 reports the train and test rewards that the pretrained, RFT, and SFT models obtain over each of the considered GRUE datasets.
- Figure 10, Table 5, and Figure 11 extend Figure 1, Table 1, and Figure 2, respectively, by considering RFT runs carried out using NLPO (Ramamurthy et al., 2023), an adaptation of PPO for language generation tasks, instead of PPO.

**Controlled experiments (Section 4.2):**

- Figure 12 supplements Figure 3 by reporting the reward standard deviation and gradient norm throughout optimization, in addition to the reward.
- Figure 13 presents experiments identical to Figure 12, except that RFT and SFT were optimized using stochastic gradient descent (SGD) as opposed to Adam.
- Figure 14 is identical to Figure 12, except that the pretrain expected reward of inputs with small pretrain reward standard deviation is $0.5$ instead of $-1$, *i.e.* it is relatively high at the start of finetuning.

**Overcoming vanishing gradients in RFT (Section 5):**

- Table 6 compares the train and test rewards obtained by RFT with an increased learning rate, temperature, and entropy regularization coefficient, to the rewards obtained by the default RFT configuration and RFT after an initial SFT phase, on the NarrativeQA dataset.
- Figure 15 presents reward means and standard deviations of individual train samples from NarrativeQA (analogously to Figure 1) after applying RFT with an increased learning rate, temperature, or entropy regularization coefficient.
- Table 7 supplements Table 6 by including results for identical experiments on the ToTTo and CommonGen datasets.

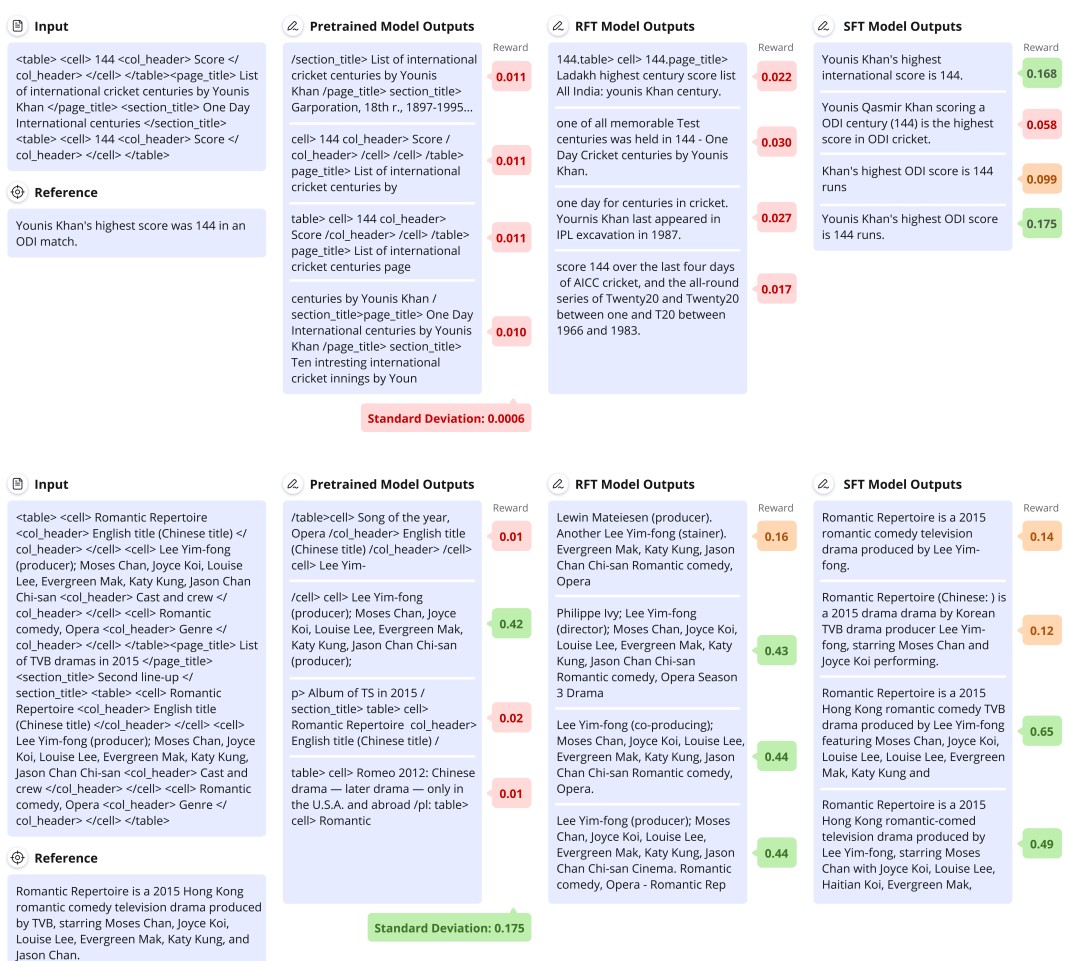

Figure 6: Representative example of an input with small (top) and an input with large (bottom) reward standard deviation under the pretrained model from the ToTTo dataset.

- Figure 16 supplements Figure 4 by reporting the same metrics based on the mean test reward, in addition to the metrics based on the mean train reward, for the partial SFT experiments on NarrativeQA.

- Figure 17 compares the reward means and standard deviations of individual train samples (analogously to Figure 1), for the NarrativeQA dataset, after performing a full SFT phase and after a partial SFT phase with 40% of the optimization steps and 1% of the labeled samples.

- Figures 18 and 19 supplement Figure 4 by including results for identical experiments on the ToTTo and CommonGen datasets, respectively.

## F.2 FURTHER IMPLEMENTATION DETAILS

We provide implementation details omitted from our experimental reports (Sections 4 and 5 and Appendix F.1). Code for reproducing our results, based on the PyTorch (Paszke et al., 2017), HuggingFace (Wolf et al., 2019), and RL4LMs (Ramamurthy et al., 2023) libraries, can be found at https://github.com/apple/ml-rlgrad. Each finetuning experiment was run on four Nvidia A100 80GB GPUs, and each experiment from Section 4.2 was run on a single Nvidia V100 32GB GPU, except for those with MLPs for which we used a standard laptop.

We note that, since the test sets of ToTTo and CommonGen are not publicly available, we report results over their validation sets instead. This does not pose an issue as our interest lies in optimization, *i.e.* the ability to achieve a higher reward over the train set, and we do not conduct any hyperparameter tuning beyond taking the default values from Ramamurthy et al. (2023).

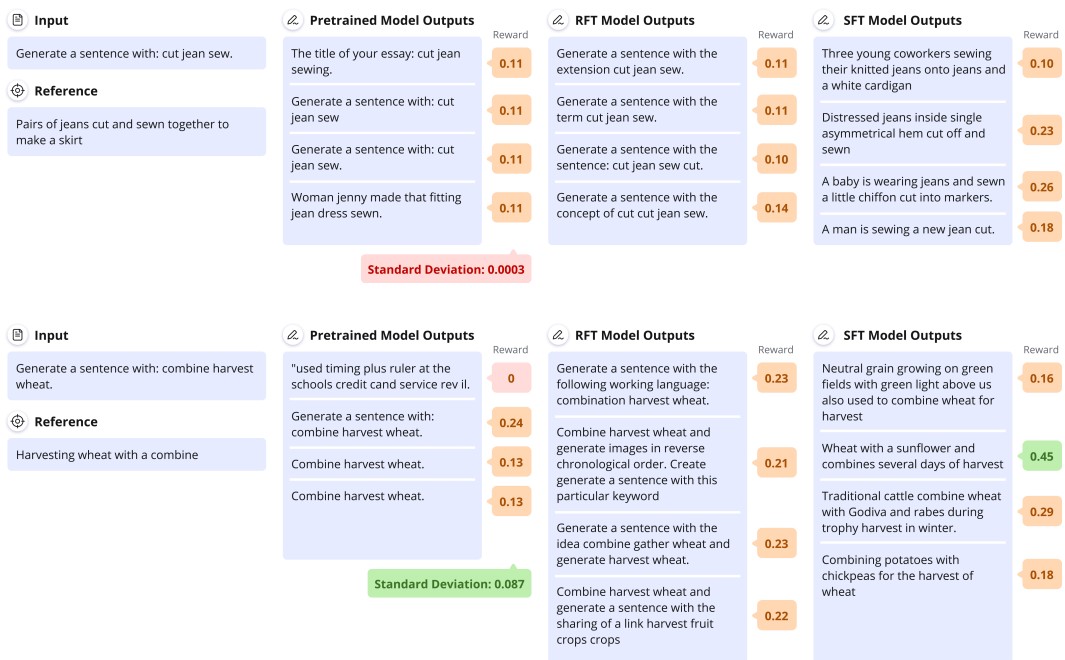

Figure 7: Representative example of an input with small (top) and an input with a relatively large (bottom) reward standard deviation under the pretrained model from the CommonGen dataset.

Table 2: The reward functions used in our experiments for datasets from the GRUE benchmark.

| Dataset | Reward |
|---|---|
| NarrativeQA | ROUGE-L-Max (Lin, 2004) |
| ToTTo | SacreBLEU (Post, 2018) |
| CommonGen | METEOR (Banerjee and Lavie, 2005) |
| IWSLT 2017 | SacreBLEU |
| CNN/Daily Mail | METEOR |
| DailyDialog | METEOR |
| IMDB | Learned sentiment classifier (Ramamurthy et al., 2023) |

### F.2.1 EXPERIMENTS ON THE GRUE BENCHMARK (SECTIONS 4.1 AND 5)

**Modifications to the default setup from Ramamurthy et al. (2023):** In our experiments with the GRUE benchmark we followed the experimental setup of Ramamurthy et al. (2023), including default hyperparameters for RFT and SFT, up to the following adjustments.

- For the NarrativeQA dataset, Ramamurthy et al. (2023) use an SFT model that was finetuned on multiple question answering datasets. In contrast, we finetune it only on NarrativeQA for a fairer comparison with RFT, which does not have access to the additional datasets.

- For the IMDB dataset, Ramamurthy et al. (2023) perform SFT using positively labeled train samples only, whereas RFT is performed over the whole training set. For a fairer comparison, we use for RFT the same train samples used for SFT.

- For the CommonGen dataset, we do not apply a repeat penalty to the reward function.

**Reward functions:** The GRUE benchmark suggests a few reward functions for each dataset, from which we used those specified in Table 2.

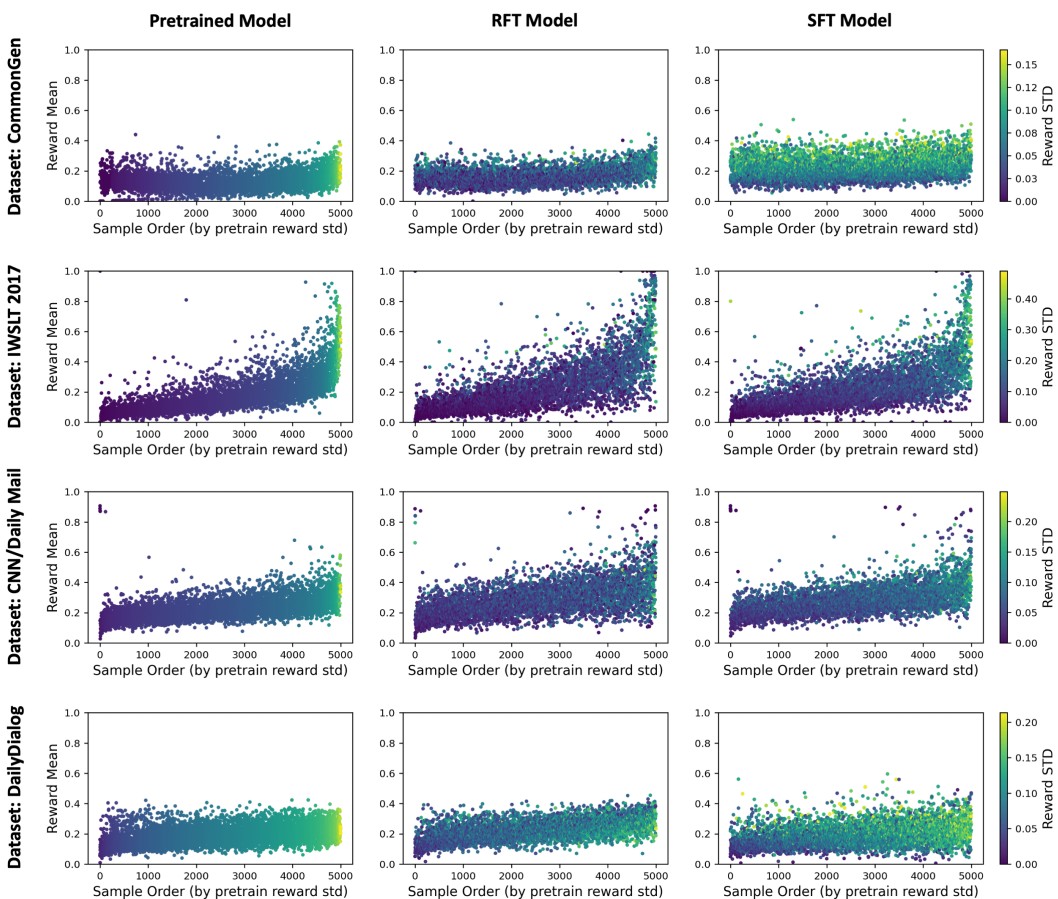

Figure 8: Inputs with small reward standard deviation under the pretrained model, *i.e.* with vanishing expected gradient, are prevalent in the GRUE benchmark. This figure supplements Figure 1, by including identical plots for the remaining GRUE datasets. For further details see caption of Figure 1.

### F.2.2 CONTROLLED EXPERIMENTS (SECTION 4.2)

**Models:** We used the default ResNet18 and BERT-mini implementations from the PyTorch and HuggingFace libraries, respectively. The MLP had two hidden layers, of sizes 250 and 100, with ReLU non-linearity.

**Data:** To facilitate more efficient experimentation, we subsampled the MNIST and CIFAR10 training sets, choosing uniformly at random 1000 samples from MNIST and 10,000 from CIFAR10. For pretraining, we assigned 4 additional labels for each sample in MNIST and CIFAR10, *i.e.* each sample had a total of 5 pretraining labels, while each sample in STS-B was assigned 2 additional labels, *i.e.* a total of 3 pretraining labels. During finetuning, every train sample was assigned a single, randomly chosen, ground truth label. For $10\%$ of the samples the label was chosen such that it was not included in their pretraining labels. As a result, their pretrain reward standard deviation was small. The rest of the samples were assigned a label such that it was one of their pretraining labels. Meaning, their pretrain reward standard deviation was relatively large.

**Optimization:** In the experiments of Figures 3, 12, and 14, the cross-entropy loss was minimized via the Adam optimizer with learning rate $0.0001$, default $\beta_1, \beta_2$ coefficients, and a batch size of 512. Pretraining and finetuning were carried out for 1000 and 5000 epochs, respectively. For Figure 13 we used an identical setup, except that Adam was replaced by SGD with learning rate $0.01$ for the MLP and ResNet18 models. To improve optimization stability, for BERT-mini we reduced the learning rate to $0.001$. Varying the learning rate (for both experiments with Adam and SGD) did not yield a noticeable improvement in the ability of RFT to maximize the reward of train samples with small pretrain reward standard deviation.

Table 3: RFT affects inputs with small reward standard deviation under the pretrained model less than it affects other inputs. This table supplements Table 1 by reporting, for the remaining GRUE datasets, the Pearson correlation between the pretrain reward standard deviation and the absolute reward mean change due to finetuning, across the train samples. As anticipated, the correlation is considerably higher for RFT compared to SFT. For further details see caption of Table 1.

|  | CommonGen | IWSLT 2017 | CNN/Daily Mail | DailyDialog |
|---|---|---|---|---|
| RFT | 0.18 | 0.51 | 0.33 | 0.33 |
| SFT | 0.07 | 0.37 | 0.23 | 0.21 |

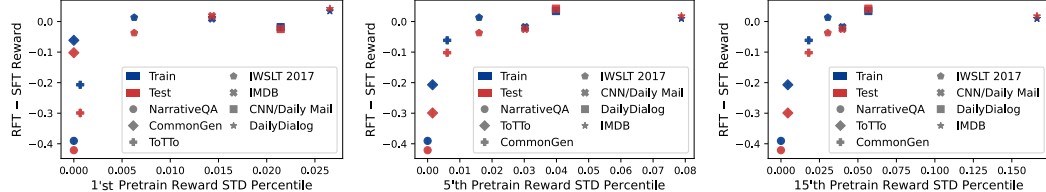

Figure 9: RFT performance (relative to SFT) is worse when inputs with small reward standard deviation are prevalent. This figure supplements Figure 2, by including identical plots with different choices for the measured pretrain reward standard deviation percentile: $1\%, 5\%$, and $15\%$ (instead of $10\%$). As can be seen, the correlation between the pretrain reward standard deviation percentile and the reward difference between RFT and SFT is not sensitive to reasonable choices of the percentile. For further details see caption of Figure 2.

Table 4: Mean reward attained by the pretrained, RFT, and SFT models over each of the considered GRUE datasets. Means and standard deviations are taken over five different random seeds per model and dataset. Since the test sets of ToTTo and CommonGen are not publicly available, reported are the rewards over their validation sets instead. This does not pose an issue as our interest lies in optimization, *i.e.* the ability to achieve a higher reward over the train set, and we do not conduct any hyperparameter tuning beyond taking the default values from Ramamurthy et al. (2023).

|  | Train Reward | | | Test Reward | | |
|---|---|---|---|---|---|---|
|  | Pretrained Model | RFT Model | SFT Model | Pretrained Model | RFT Model | SFT Model |
| NarrativeQA | $0.110 \pm 0.000$ | $0.102 \pm 0.012$ | $0.493 \pm 0.003$ | $0.117 \pm 0.000$ | $0.116 \pm 0.002$ | $0.537 \pm 0.001$ |
| ToTTo | $0.024 \pm 0.000$ | $0.098 \pm 0.014$ | $0.305 \pm 0.005$ | $0.036 \pm 0.000$ | $0.144 \pm 0.019$ | $0.443 \pm 0.003$ |
| CommonGen | $0.130 \pm 0.000$ | $0.163 \pm 0.007$ | $0.224 \pm 0.000$ | $0.179 \pm 0.000$ | $0.206 \pm 0.005$ | $0.308 \pm 0.001$ |
| IWSLT 2017 | $0.175 \pm 0.000$ | $0.239 \pm 0.005$ | $0.226 \pm 0.000$ | $0.307 \pm 0.000$ | $0.299 \pm 0.002$ | $0.337 \pm 0.000$ |
| CNN/Daily Mail | $0.227 \pm 0.000$ | $0.260 \pm 0.011$ | $0.279 \pm 0.000$ | $0.255 \pm 0.000$ | $0.284 \pm 0.010$ | $0.308 \pm 0.000$ |
| DailyDialog | $0.191 \pm 0.000$ | $0.224 \pm 0.003$ | $0.189 \pm 0.005$ | $0.190 \pm 0.001$ | $0.222 \pm 0.004$ | $0.180 \pm 0.005$ |
| IMDB | $0.831 \pm 0.001$ | $0.890 \pm 0.005$ | $0.880 \pm 0.001$ | $0.498 \pm 0.002$ | $0.565 \pm 0.009$ | $0.547 \pm 0.002$ |

Table 5: RFT (using NLPO) affects inputs with small reward standard deviation under the pretrained model less than it affects other inputs. This table supplements Tables 1 and 3, by including the Pearson correlations for RFT performed via NLPO (Ramamurthy et al., 2023) instead of PPO. Similarly to RFT with PPO, the correlation between the pretrain reward standard deviation of a train sample and its absolute reward mean change for RFT with NLPO is considerably higher than for SFT. For further details see caption of Table 1.

|  | NarrativeQA | ToTTo | IMDB | CommonGen | IWSLT 2017 | CNN/Daily Mail | DailyDialog |
|---|---|---|---|---|---|---|---|
| RFT (NLPO) | 0.42 | 0.48 | 0.71 | 0.19 | 0.51 | 0.34 | 0.29 |
| RFT | 0.48 | 0.46 | 0.72 | 0.18 | 0.51 | 0.33 | 0.33 |
| SFT | 0.05 | 0.16 | 0.72 | 0.07 | 0.37 | 0.23 | 0.21 |

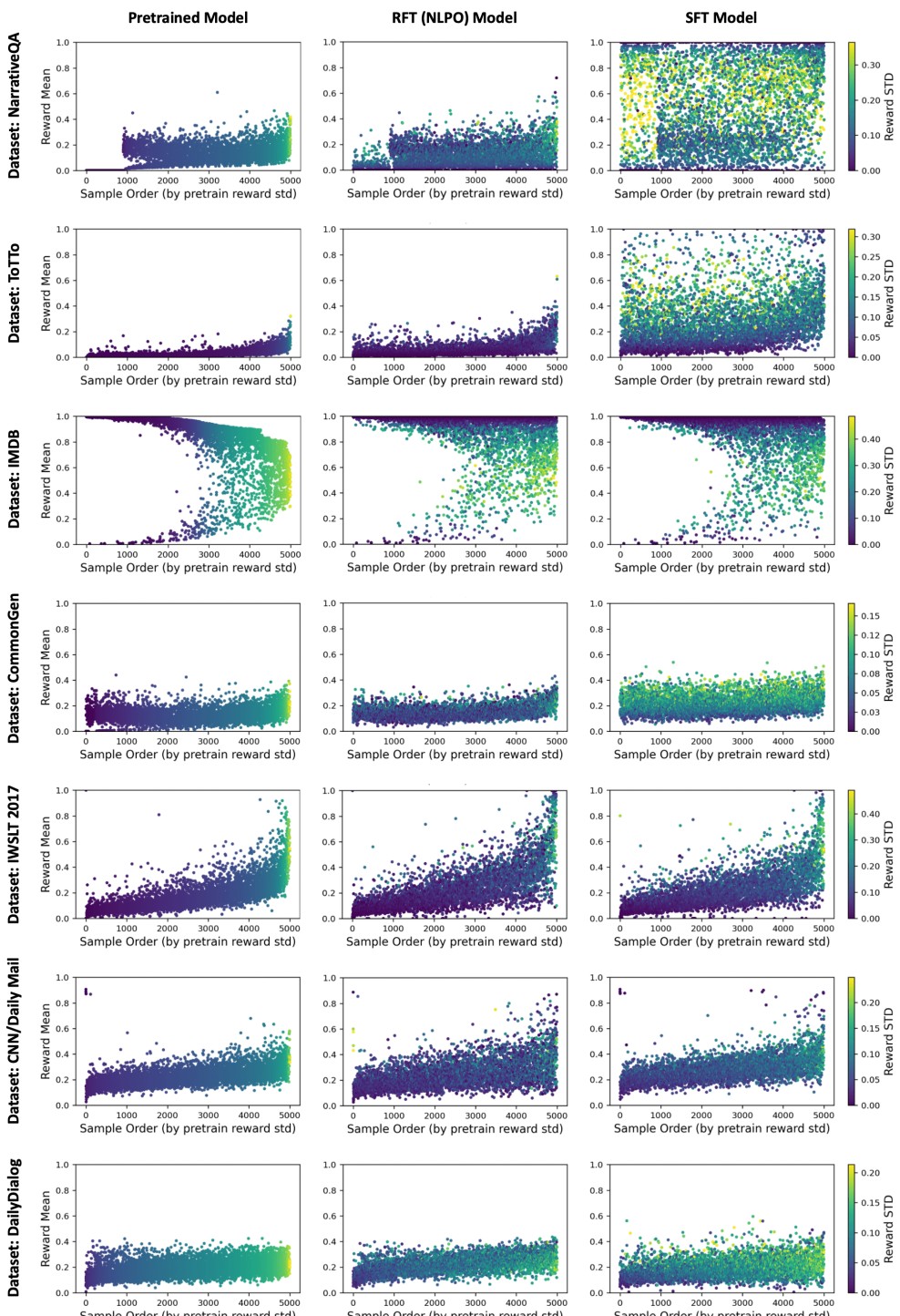

Figure 10: Inputs with small reward standard deviation under the pretrained model, *i.e.* with vanishing expected gradient, are prevalent in the GRUE benchmark. This figure is identical to Figures 1 and 8, except that NLPO (Ramamurthy et al., 2023) is used instead of PPO for RFT. The pretrained model (left) and SFT model (right) columns are repeated here for ease of comparison. Notice that, the effect of RFT with NLPO on the rewards of train samples is qualitatively similar to that of RFT with PPO. For further details see caption of Figure 1.

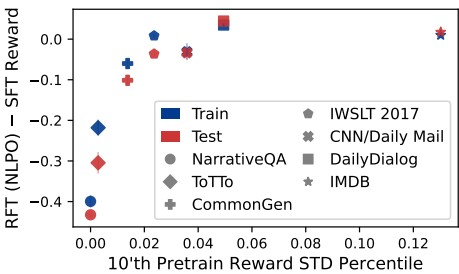

Figure 11: RFT (using NLPO) performance (relative to SFT) is worse when inputs with small reward standard deviation are prevalent. This figure is identical to Figure 2, except that NLPO (Ramamurthy et al., 2023) is used instead of PPO for RFT. For further details see caption of Figure 2.

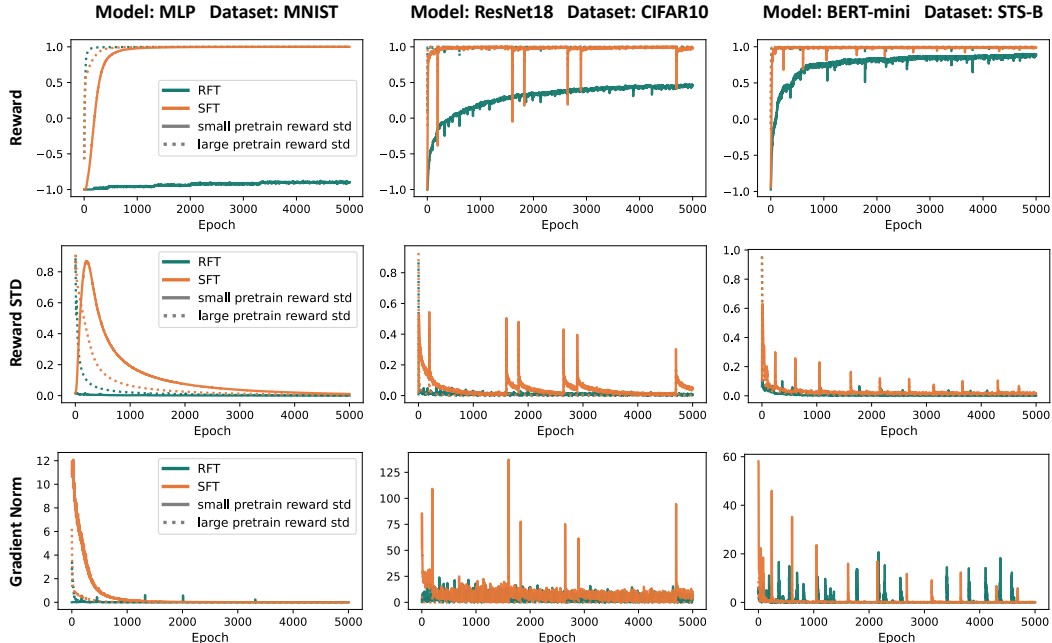

Figure 12: RFT struggles to maximize the reward over inputs with small reward standard deviation under the pretrained model, *i.e.* inputs with vanishing expected gradient, even with perfect exploration. On the contrary, SFT easily leads to maximal reward. This figure supplements Figure 3 by reporting the reward standard deviation and gradient norm, in addition to the reward (separately for train samples with small and train samples with large pretrain reward standard deviation). The top row is identical to Figure 3, and is repeated here for convenience. Notice that, in accordance with Theorem 1, the gradient norm under RFT vanishes when the reward standard deviation is small, whereas under SFT the gradient norm vanishes only when the reward is near-maximal. For further details see caption of Figure 3.

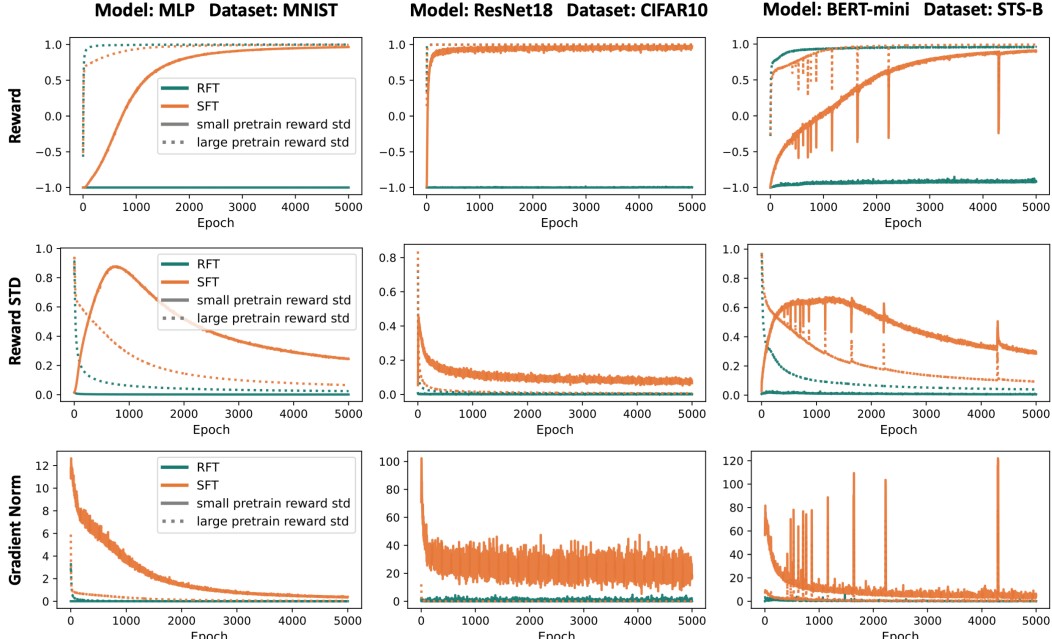

Figure 13: RFT struggles to maximize the reward over inputs with small reward standard deviation under the pretrained model, *i.e.* inputs with vanishing expected gradient, even with perfect exploration (using stochatic gradient descent). On the contrary, SFT easily leads to maximal reward. This figure is identical to Figure 12, except that RFT and SFT were optimized using stochastic gradient descent (SGD) as opposed to Adam (note that in RFT we use SGD to minimize the negative expected reward). As can be seen, the inability of RFT to maximize the rewards of inputs with small reward standard deviation is even more severe when using SGD instead of Adam. For further details see caption of Figure 12.

Table 6: Conventional heuristics are inadequate for overcoming vanishing gradients in RFT. For the NarrativeQA dataset, which was shown in Section 4.1 to suffer from vanishing gradients, reported are means and standard deviations of train and test rewards, taken over three random seeds, with best result in bold. While the common heuristics of increasing the learning rate, temperature, and entropy regularization coefficient are unable to improve upon the default RFT configuration, an initial SFT phase is highly effective. See Table 7 in Appendix F.1 for analogous experiments on the ToTTo and CommonGen datasets.

Dataset: NarrativeQA

|  | Train Reward | Test Reward |
|---|---|---|
| RFT[*] | $0.101 \pm 0.009$ | $0.116 \pm 0.000$ |
| SFT + RFT | **$0.537 \pm 0.005$** | **$0.544 \pm 0.003$** |
| RFT with learning rate $2 \cdot 10^{-5}$ | $0.012 \pm 0.010$ | $0.020 \pm 0.017$ |
| RFT with learning rate $2 \cdot 10^{-4}$ | $0.053 \pm 0.010$ | $0.048 \pm 0.016$ |
| RFT with learning rate $2 \cdot 10^{-3}$ | $0.039 \pm 0.018$ | $0.012 \pm 0.020$ |
| RFT with temperature 1.5 | $0.077 \pm 0.021$ | $0.118 \pm 0.002$ |
| RFT with temperature 2 | $0.060 \pm 0.018$ | $0.104 \pm 0.009$ |
| RFT with temperature 2.5 | $0.044 \pm 0.004$ | $0.088 \pm 0.009$ |
| RFT with entropy regularization 0.01 | $0.080 \pm 0.006$ | $0.113 \pm 0.002$ |
| RFT with entropy regularization 0.1 | $0.024 \pm 0.005$ | $0.019 \pm 0.007$ |
| RFT with entropy regularization 1 | $0.011 \pm 0.000$ | $0.013 \pm 0.010$ |

[*]With default hyperparameters: learning rate $2 \cdot 10^{-6}$, temperature 1, entropy regularization 0.

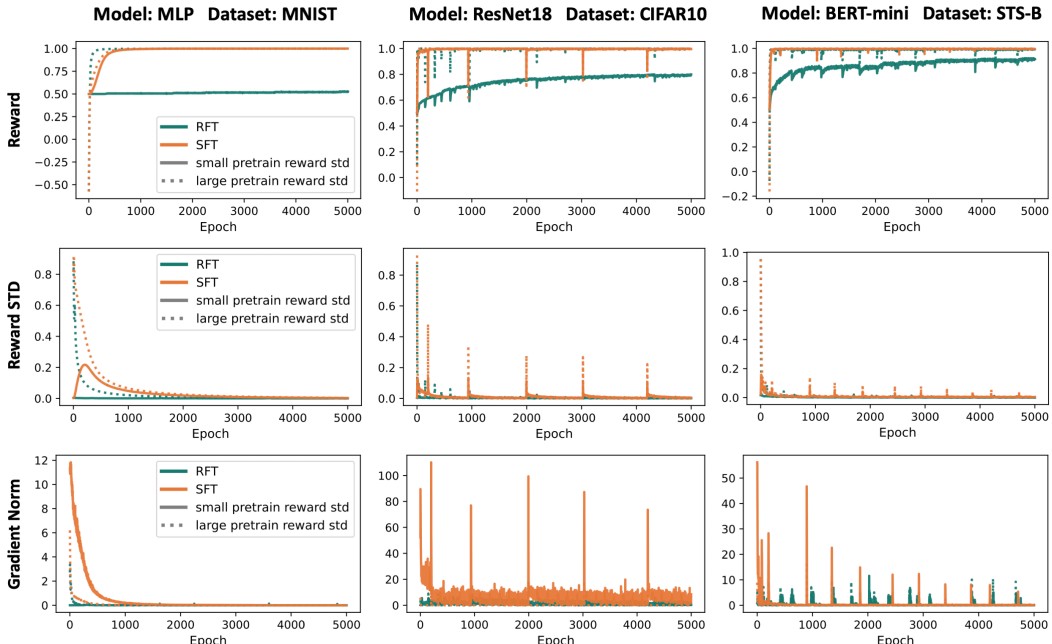

Figure 14: RFT struggles to maximize the reward over inputs with small reward standard deviation under the pretrained model, *i.e.* inputs with vanishing expected gradient, even with perfect exploration and when their expected reward under the pretrained model is relatively high. On the contrary, SFT easily leads to maximal reward. This figure is identical to Figure 12, except that for inputs with small pretrain reward standard deviation (*i.e.* inputs for which the finetuning label was not included in their pretraining labels), the reward of an incorrect label is set to 0.5 as opposed to −1. Consequently, the expected reward of such inputs at the beginning of finetuning is 0.5, while it is −1 in the experiments of Figure 12. Nonetheless, RFT struggles to increase their reward, with the reward dynamics mimicking those from Figure 12. For further details see caption of Figure 12.

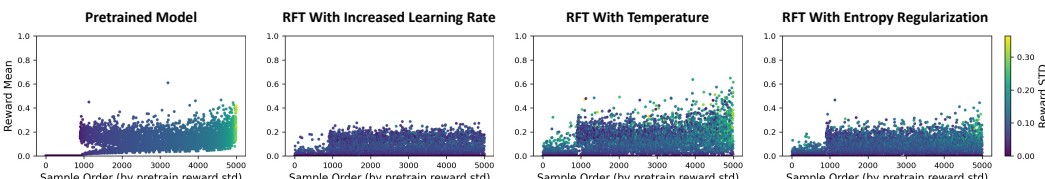

Figure 15: Conventional heuristics are inadequate for overcoming vanishing gradients in RFT. For the same randomly chosen NarrativeQA train samples from Figure 1, presented are the reward means and standard deviations (estimated based on ten generations per input) under the pretrained model and RFT models optimized with an increased learning rate, temperature, or entropy regularization coefficient. The learning rate, temperature, and entropy regularization values with which the plots were created are those achieving the highest train reward (excluding the default values), as reported in Table 6. Notice that the effect of these RFT runs is qualitatively similar to that of the default RFT configuration (shown in Figure 1). Meaning, the heuristics do not alleviate the difficulty of RFT to improve the reward of inputs with small pretrain reward standard deviation.

Table 7: Conventional heuristics are inadequate for overcoming vanishing gradients in RFT. This table supplements Table 6 by including results for the ToTTo and CommonGen datasets. Reported are means and standard deviations of train and test rewards, taken over three different random seeds, with best result in bold. While the common heuristics of increasing the learning rate, temperature, and entropy regularization coefficient are unable to improve upon the default RFT configuration, an initial SFT phase is highly effective.

Dataset: ToTTo

|  | Train Reward | Validation Reward |
|---|---|---|
| RFT[*] | $0.095 \pm 0.026$ | $0.137 \pm 0.037$ |
| SFT + RFT | $\mathbf{0.342 \pm 0.001}$ | $\mathbf{0.439 \pm 0.001}$ |
| RFT with learning rate $2 \cdot 10^{-5}$ | $0.065 \pm 0.005$ | $0.131 \pm 0.003$ |
| RFT with learning rate $2 \cdot 10^{-4}$ | $0.000 \pm 0.000$ | $0.000 \pm 0.000$ |
| RFT with learning rate $2 \cdot 10^{-3}$ | $0.000 \pm 0.000$ | $0.000 \pm 0.000$ |
| RFT with temperature 1.5 | $0.059 \pm 0.007$ | $0.096 \pm 0.022$ |
| RFT with temperature 2 | $0.043 \pm 0.004$ | $0.073 \pm 0.005$ |
| RFT with temperature 2.5 | $0.019 \pm 0.001$ | $0.038 \pm 0.002$ |
| RFT with entropy regularization 0.01 | $0.061 \pm 0.013$ | $0.139 \pm 0.014$ |
| RFT with entropy regularization 0.1 | $0.000 \pm 0.000$ | $0.018 \pm 0.015$ |
| RFT with entropy regularization 1 | $0.000 \pm 0.000$ | $0.000 \pm 0.000$ |

[*]With default hyperparameters: learning rate $2 \cdot 10^{-6}$, temperature 1, entropy regularization 0

Dataset: CommonGen

|  | Train Reward | Validation Reward |
|---|---|---|
| RFT[*] | $0.164 \pm 0.004$ | $0.209 \pm 0.005$ |
| SFT + RFT | $\mathbf{0.265 \pm 0.001}$ | $\mathbf{0.319 \pm 0.008}$ |
| RFT with learning rate $2 \cdot 10^{-5}$ | $0.098 \pm 0.019$ | $0.134 \pm 0.015$ |
| RFT with learning rate $2 \cdot 10^{-4}$ | $0.112 \pm 0.094$ | $0.157 \pm 0.072$ |
| RFT with learning rate $2 \cdot 10^{-3}$ | $0.055 \pm 0.029$ | $0.106 \pm 0.048$ |
| RFT with temperature 1.5 | $0.139 \pm 0.004$ | $0.190 \pm 0.017$ |
| RFT with temperature 2 | $0.147 \pm 0.004$ | $0.189 \pm 0.000$ |
| RFT with temperature 2.5 | $0.140 \pm 0.007$ | $0.182 \pm 0.006$ |
| RFT with entropy regularization 0.1 | $0.009 \pm 0.000$ | $0.083 \pm 0.020$ |
| RFT with entropy regularization 1 | $0.002 \pm 0.000$ | $0.004 \pm 0.003$ |
| RFT with entropy regularization 10 | $0.001 \pm 0.000$ | $0.000 \pm 0.000$ |

[*]With default hyperparameters: learning rate $2 \cdot 10^{-6}$, temperature 1, entropy regularization 0.01

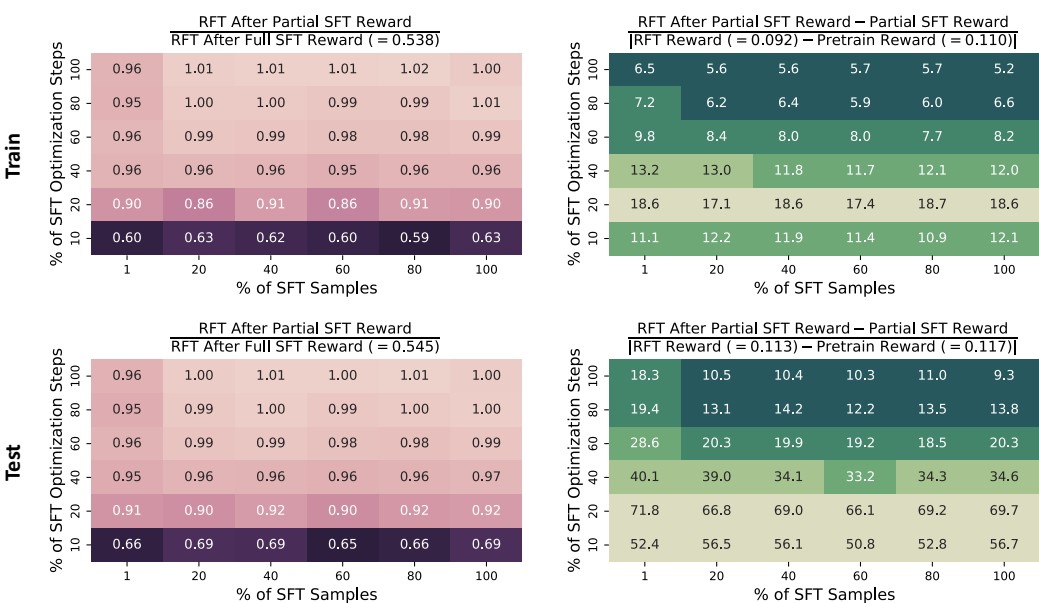

Figure 16: On datasets in which RFT suffers from vanishing gradients, a few initial SFT optimization steps on a small number of labeled inputs substantially boost the efficacy of RFT. This figure supplements Figure 4 by reporting metrics based on the mean test reward, in addition to on the mean train reward. The top row is identical to Figure 4, repeated here for ease of comparison.

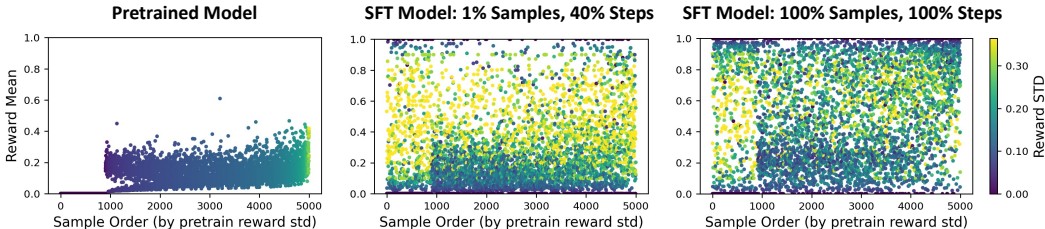

Figure 17: A few initial SFT optimization steps on a small number of labeled inputs effectively mitigate vanishing gradients in RFT. For the same randomly chosen NarrativeQA train samples from Figure 1, presented are the reward means and standard deviations (estimated based on ten generations per input) under the pretrained, a partial SFT, and the full (default) SFT models. The pretrained model (left) and SFT model (right) plots are identical to the respective plots in Figure 1, and are repeated here for ease of comparison. Observe that the number of samples with small reward standard deviation, *i.e.* with vanishing expected gradient, is considerably reduced after the partial SFT. We highlight that, despite the reward mean obtained by the partial SFT model being noticeably lower than that obtained by the full SFT model, after performing RFT the gap is closed almost completely. That is, as shown in Figure 4, RFT over the partial SFT model reaches roughly the same reward as RFT over the full SFT model. This highlights the effectiveness of a partial initial SFT phase for addressing vanishing gradients in RFT.

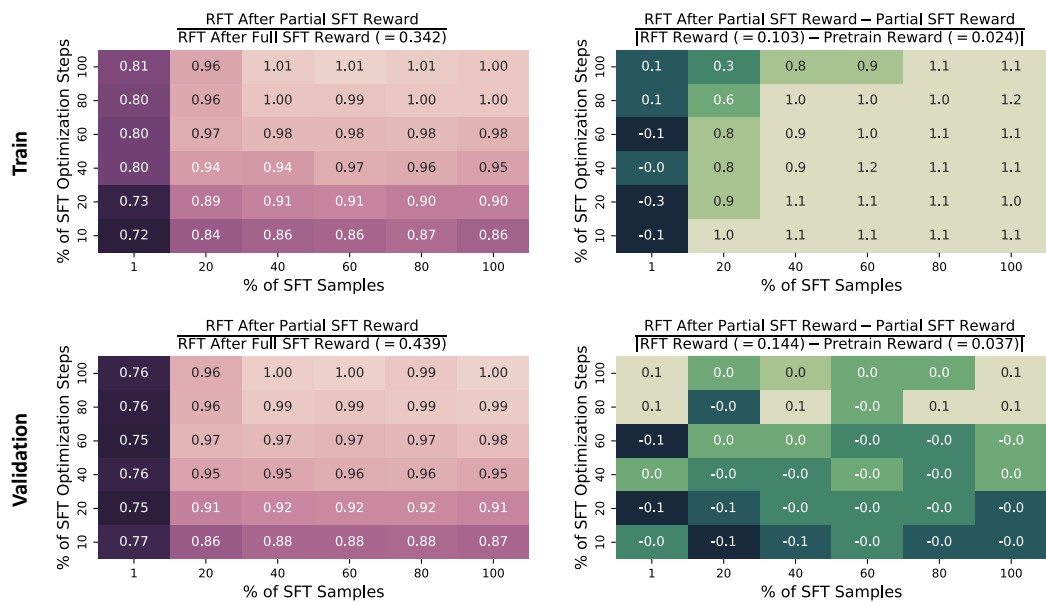

Figure 18: On datasets in which RFT suffers from vanishing gradients, a few initial SFT optimization steps on a small number of labeled inputs substantially boost the efficacy of RFT. This figure is identical to Figure 16, except that the experiments were carried out over the ToTTo dataset (as opposed to NarrativeQA).

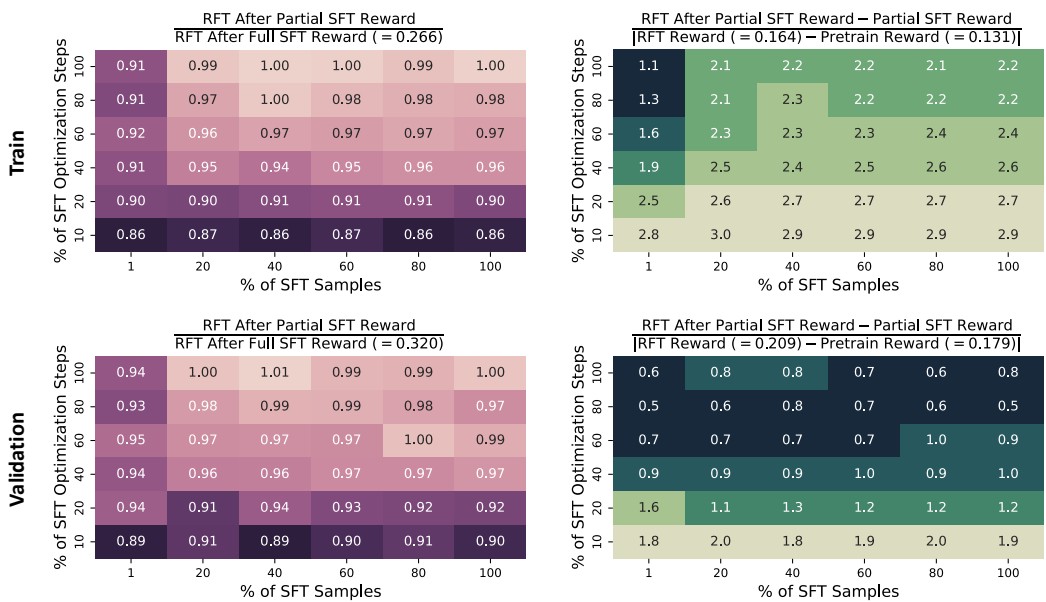

Figure 19: On datasets in which RFT suffers from vanishing gradients, a few initial SFT optimization steps on a small number of labeled inputs substantially boost the efficacy of RFT. This figure is identical to Figure 16, except that the experiments were carried out over the CommonGen dataset (as opposed to NarrativeQA).

