# OpenReview forum: "Vanishing Gradients in Reinforcement Finetuning of Language Models"
_ICLR.cc/2024/Conference — ICLR 2024 poster_

### Official Review · Reviewer_voez · 2023-10-30

**Soundness:** 3 good
**Presentation:** 3 good
**Contribution:** 3 good
**Rating:** 6
**Confidence:** 3

**Summary:**

This manuscript identifies a problem in policy-gradient-based Reinforcement Learning Fine-Tuning (RLFT): when the reward variance is small, the language model (LM) could be susceptible to vanishing gradients.  The authors provided both theoretical and empirical evidence to support their argument, and demonstrated that a supervised fine-tuning "jumpstart" could help alleviate the vanishing-gradient issue.

**Strengths:**

The authors identified an interesting issue (although rather self-explanatory), and advocated for a simple and straightforward solution.

- I admire the organization of this manuscript -- very clean and structured.
- The authors included a comprehensive set of empirical results, most of which are carefully presented and scrupulously discussed.

**Weaknesses:**

- The writing has room for improvement.  The authors tend to make heavy use of long sentences that are difficult to parse.
- The second paragraph on page 4, I am not sure how the sentence "the contribution of inputs... is still negligible..." is logically connected to the next sentence.
- The experiment setup in Section 4.1 is worth closer scrutiny.  It does not seem a fair comparison to me that RLFT optimizes for a task-specific metric while SFT is performed based on ground-truth completions, which means that RLFT *does not* have access to the ground-truth, whereas SFT *does*.
- It is worth noting that running RLFT with "task-specific" metrics, such as ROUGE, is no longer considered a good practice in the RLHF community.  Specifically for the GRUE benchmarks, I believe that the community has reached a consensus that it is not a suitable playground for RLFT.  I suppose that the authors would agree that the predominant approach in RLFT is to train a reward model using a preference (or ranking) dataset, and apply policy-gradient algorithms based on the reward model.  I could imagine that the same issue identified in this manuscript is also present in such scenarios, but the authors did not provide sufficient evidence.  (I wouldn't place too much weight on this weakness point, because, understandably, the authors might be short on compute resources.)
- Suppose that we are given a preference dataset: each prompt comes with two responses, one labelled as preferred.  How should we run SFT on such dataset?  Shall we only use the one preferred response for each prompt, or shall we also somehow leverage the other response as a negative sample?  This is subject to debate in the LLM community, and I hope the authors can shed more light on this point.

**Questions:**

Please refer to the "Weaknesses" section.

---

> ### Author Response · Authors · 2023-11-13
>
> Thank you for the thoughtful feedback, and for highlighting the interest in the identified problem, our comprehensive experimental analysis, and the organization of the manuscript. We address your comments below. If our response is satisfactory, we would greatly appreciate it if you would consider raising your score.
>
> > The writing has room for improvement. The authors tend to make heavy use of long sentences that are difficult to parse.
>
> We will make sure to further polish writing by rephrasing such sentences. We would greatly appreciate specific pointers to sentences you consider difficult to parse.
>
> > The second paragraph on page 4, I am not sure how the sentence "the contribution of inputs... is still negligible..." is logically connected to the next sentence.
>
> The sentence you refer to explains why Adam is not expected to mitigate the vanishing gradients problem. The following sentence states that the experiments of Section 4 affirm this expectation. We have rephrased the second paragraph on page 4 to clarify this point (modified text is marked in red). Thank you for the helpful comment!
>
> > The experiment setup in Section 4.1 is worth closer scrutiny. It does not seem a fair comparison to me that RLFT optimizes for a task-specific metric while SFT is performed based on ground-truth completions, which means that RLFT does not have access to the ground-truth, whereas SFT does.
>
> We emphasize that the goal of Section 4.1 is not to compare the performance of RFT with SFT, in order to advocate for the use of one over the other, rather to demonstrate the prevalence and detrimental effects of vanishing gradients in RFT. In particular, we contrast RFT with SFT to showcase how RFT struggles to increase the reward of inputs with small reward standard deviation, compared to its ability to increase the reward of other inputs. For example, in Figure 1 and Table 1, SFT poses a baseline that allows quantifying whether the correlation between reward standard deviation and reward change during RFT is high or low. The comparison to SFT verifies that it is relatively high, as expected.
>
> We hope that this addresses your concern. Please let us know otherwise and we will gladly elaborate.
>
> > It is worth noting that running RLFT with "task-specific" metrics, such as ROUGE, is no longer considered a good practice in the RLHF community. Specifically for the GRUE benchmarks, I believe that the community has reached a consensus that it is not a suitable playground for RLFT. I suppose that the authors would agree that the predominant approach in RLFT is to train a reward model using a preference (or ranking) dataset, and apply policy-gradient algorithms based on the reward model. I could imagine that the same issue identified in this manuscript is also present in such scenarios, but the authors did not provide sufficient evidence. (I wouldn't place too much weight on this weakness point, because, understandably, the authors might be short on compute resources.)
>
> The nature of our work is foundational, aiming to improve the understanding of existing methods. Accordingly, to allow for efficient experimentation, we mostly focused on the GRUE benchmark, which does not include reward functions learned from human feedback. We agree with the reviewer that a predominant approach in practice is to use reward functions learned from human preference data. Thus, as mentioned in the conclusion section, we consider studying the extent to which our empirical findings carry over to more complex finetuning pipelines, including human feedback, a valuable direction for future research.
>
> > Suppose that we are given a preference dataset: each prompt comes with two responses, one labelled as preferred. How should we run SFT on such dataset? Shall we only use the one preferred response for each prompt, or shall we also somehow leverage the other response as a negative sample? This is subject to debate in the LLM community, and I hope the authors can shed more light on this point.
>
> Our work focuses on a fundamental vanishing gradients problem in RFT. Among other contributions, we show that given access to a small number of ground truth labels (i.e. completions), an initial SFT phase can be effective in mitigating the problem. However, we do not treat the orthogonal question of how to best obtain ground truth labels for SFT. We believe investigating methods that use preference data to create SFT labels is an important subject, yet falls outside the scope of our findings.

---

### Official Review · Reviewer_xsTc · 2023-11-01

**Soundness:** 3 good
**Presentation:** 3 good
**Contribution:** 3 good
**Rating:** 6
**Confidence:** 3

**Summary:**

This paper studies the vanishing gradients in RLHF. Specifically, the authors argue that shrinkage in the standard deviation (std) of the reward would lead to vanishing gradients, leading to sub-optimal performance. The authors first show that norm of the gradient is upper bounded by the std of the reward plus some other factors, suggesting that as std goes to zero, training would stall due to vanishing gradients. On GRUE benchmark, the authors show that this is indeed the case; there are a substantial number of examples that have very small std under the pre-trained model. They show that RFT impacts these examples less than others; SFT, on the other hand, doesn't have this problem as ground truth labels force proper signal propagation. Using conventional heuristics such as larger learning rate, tuning temperature or entropy regularization is not effective in solving this problem. But, a few steps of SFT is shown to be effective in improving RFT performance.

**Strengths:**

The paper investigates a very important problem. It is well written with relevant experiments to support the claims.

**Weaknesses:**

Some parts of the paper need clarification.

1. Theorem-1 applies more broadly beyond autoregressive models. The only part that is relevant is the Jacobian which is easily separated from the rest using the triangle inequality in the proof. Take a simple non-autoregressive example with action space A ($|A|=2$), a linear policy $\theta^T x$ such that $p_\theta(y=0|x)=\theta_0 x_0$, $p_\theta(y=1|x)=\theta_1 x_1$, and $r(x, y=0)=r(x, y=1)=1$. The standard deviation should be zero and lead to vanishing gradients. But, the gradient is $\nabla_\theta(x; \theta)=\theta_0 x_0 [x_0, 0]/(\theta_0x_0) + \theta_1 x_1 [0, x_1]/(\theta_1x_1)=[x_0, x_1]$ which doesn't necessarily have zero gradient norm. Can you clarify if the gradient estimation in this example is incorrect or how I should interpret this result?

2. Can you discuss in more detail when small vanishing gradient becomes a problem? Assuming that an LLM is trained with RLHF to convergence, we would expect many of the completions to be of equal quality, i.e., similar rewards. Why would low standard deviation be a problem in this case? Similarly, at convergence, we expect gradient to be close to zero which also is not detrimental to training.

3. While not guaranteed (also not always in practice), ADAM update equations suggest O(1) updates to the weights. For a batch size of 1, would this solve the vanishing gradient problem? What is the maximal batch size where this problem is negligible? Can you also plot some of the statistics from ADAM training, like $\mu$, $\nu$, and $u$ (update) norms? What would be the impact of $\epsilon$ when the significant portion of a batch has vanishing gradients, i.e., would $\epsilon$ dominate second order moments?

4. Please clarify O-notation in Appendix D.4 w.r.t. concerned variable. Otherwise, it suggests negative value inside O or instant learning when STD goes to 1 in SFT.

**Questions:**

1. Can you please clarify the above example in the context of theorem-1?

2. Can you discuss when and why small vanishing gradient is a problem?

3. Can you discuss more on ADAM update and present more results?

---

> ### Author Response · Authors · 2023-11-13
> **Response (Part 1/2)**
>
> Thank you for the thoughtful feedback, for highlighting the importance of the investigated problem, and for noting that the paper is well-written. We treat your comments and questions below. If our response is satisfactory, we would greatly appreciate it if you would consider increasing your score.
>
> > 1 + Q1: Can you please clarify the above example in the context of theorem-1?
>
> Theorem 1 indeed applies also for non-autoregressive models, which can be obtained as a special case by considering an output sequence length of one. The important part missing from the example in the review is the softmax operator. We consider the standard practice of using softmax for converting the outputs of a model into a distribution over tokens (as stated in the preliminaries section, with the role of softmax in causing vanishing gradients mentioned in the introduction and paragraph following Theorem 1).
>
> We note that linear policies, or neural network policies using a linear head without softmax, do not define valid output distributions for all inputs (e.g., in the provided example the model defines a valid distribution only for inputs satisfying $\theta_0 x_0 + \theta_1 x_1 = 1$ and $x_0 \theta_0 \geq 0, x_1 \theta_1 \geq 0$). Thus, although they do not suffer from the vanishing gradients problem identified in this work, they are not directly applicable with policy gradient algorithms. Though, attempting to mitigate vanishing gradients in RFT by replacing softmax with alternative transformations may be an interesting topic for future work.
>
> > 2 + Q2: Can you discuss in more detail when small vanishing gradient becomes a problem? Assuming that an LLM is trained with RLHF to convergence, we would expect many of the completions to be of equal quality, i.e., similar rewards. Why would low standard deviation be a problem in this case? Similarly, at convergence, we expect gradient to be close to zero which also is not detrimental to training.
>
> Theorem 1 establishes that, if the reward standard deviation for an input is small, then its expected gradient vanishes even if the expected reward is far from optimal. While small reward standard deviation and vanishing gradients are expected when the model is near-optimal, they are problematic when the reward is suboptimal – they can hinder the ability of RFT to maximize the reward (as discussed both in the introduction and Section 3). Indeed, the experiments on the GRUE benchmark (Section 4.1) and controlled environments (Section 4.2) demonstrate that, in the presence of inputs with small reward standard deviation and low reward, RFT is unable to effectively maximize the reward.
>
> As to when vanishing gradients, due to small reward standard deviation, occur in RFT. We expect inputs with small reward standard deviation (and low reward mean) to be common when the text distribution of the downstream task markedly differs from that of the pretraining corpora. In such a case, over some inputs, the pretrained model is likely to give non-negligible probability only to outputs of roughly the same suboptimal reward. The reward standard deviation for these inputs is small, implying that their expected gradients are near-zero. We note that the above was also discussed in the introduction and Section 3. Section 4.1 corroborates this expectation using the GRUE benchmark.
>
> We hope this fully addresses your question. If not please let us know and we will happily expand!

---

> ### Author Response · Authors · 2023-11-13
> **Response (Part 2/2)**
>
> > 3 + Q3: While not guaranteed (also not always in practice), ADAM update equations suggest O(1) updates to the weights. For a batch size of 1, would this solve the vanishing gradient problem?
>
> Using a batch size of 1 is unlikely to effectively solve the problem since:
>
> - The update in Adam is based on moving averages of gradients and squared gradients, which are computed across batches (when the batch size is 1, they are computed across individual samples). Hence, similar to the batch size > 1 case, the effect of inputs with vanishing expected gradients on these averages, and thus on the update, will be negligible compared to those of other inputs.
>
> - Using a batch size of 1 is impractical since it does not allow efficient parallelism, and may lead to instability.
>
> To affirm that a batch size of 1 does not mitigate the vanishing gradients problem in RFT, we ran experiments over the NarrativeQA dataset. Indeed, the reward achieved does not improve upon the default batch size used in our experiments.
>
> **Results:**
> - Dataset: NarrativeQA, Model: T5-base, Batch size: 64 (default value in our experiments) – train reward 0.107, test reward 0.114
> - Dataset: NarrativeQA, Model: T5-base, Batch size: 1 – train reward 0.099, test reward 0.101
>
> Furthermore, we include in the following (anonymized) link a figure, analogous to Figure 1. It compares the reward means and standard deviations of training samples under the pretrained, RFT, and RFT with batch size 1 models. The figure demonstrates that using a batch size of 1 does not improve the ability of RFT to maximize the reward in the presence of inputs with small reward standard deviation.
>
> Link: https://ibb.co/F3JdLP9
>
> Technical note: To obtain results in time for the discussion period, we reduced the overall number of optimization steps to 25% of the original number of steps. This is done for both the run with a batch size of 1 and the run with the default batch size of 64. Since reducing the batch size led to lower train and test rewards, it is likely that increasing the number of iterations will not result in a major improvement (if at all) over the default batch size.
>
> > Can you also plot some of the statistics from ADAM training, like $\mu$, $\nu$, and $u$ (update) norms?
>
> Of course. In the additional experiments described above, we tracked the norms for the batch gradient (denoted $||\nabla||$), Adam update (denoted $||u||$), exponentially moving gradient average (denoted $||\mu||$), and the square root (taken entry-wise) of the exponentially moving squared gradients average (denoted $||\sqrt{\nu}||$). We include below (anonymized) links to plots reporting these quantities against the optimization step.
>
> **Results:**
> - Dataset: NarrativeQA, Model: T5-base, Batch size: 64 (default value in our experiments) – https://ibb.co/k3xRkmg
> - Dataset: NarrativeQA, Model: T5-base, Batch size: 1 – https://ibb.co/nCbBLNF
>
> As expected, the plots show that: (1) the Adam update and moving averages are not near-zero for both batch size 1 and 64; and (2) the gradient of individual samples, as can be observed in the leftmost plot for batch size 1, is zero for quite a few of the samples. Though, since the moving averages are not near-zero, the contribution of these samples (with near-zero gradients) to the update performed by Adam is negligible.
>
> > What would be the impact of $\epsilon$ when the significant portion of a batch has vanishing gradients, i.e., would $\epsilon$ dominate second order moments?
>
> If the second order moments average is practically zero (note that this requires multiple consecutive batches with a significant portion of vanishing gradients), then $\epsilon$ will indeed dominate the normalization term.
>
> > 4. Please clarify O-notation in Appendix D.4 w.r.t. concerned variable. Otherwise, it suggests negative value inside O or instant learning when STD goes to 1 in SFT.
>
> The $O$-notation in Appendix D.4 signifies that there exist constants $c_1 \in \mathbb{R}_{> 0}$ and $ c_2 \in \mathbb{R}$  (i.e., $c_1, c_2$ do not depend on the initialization $\theta^{(0)}$, which determines the initial standard deviation) such that $t_S \leq c_1 \cdot \ln (1 / STD) + c_2$, where STD stands for the reward standard deviation of the input under $\theta^{(0)}$ ($t_S$ is a shorthand for t_SFT). The constants ensure that the upper bound is always strictly greater than zero. Similarly, we use $\Omega$ to signify that there exist such constants for which $t_R \geq c_1 \cdot \frac{1}{STD^2} + c_2$ ($t_R$ is a shorthand for t_RFT).
>
> We have clarified this point in the manuscript (additional text marked by red in Theorem 2 of Appendix D.4). Thank you for the comment and helping us improve the clarity of our work!

---

> ### Comment · Reviewer_xsTc · 2023-11-21
> **Thank you for the update.**
>
> Thank you for the clarification and new results; especially ADAM state and gradient norms were helpful. Indeed, adding softmax resolves the issue for the example that I shared. I raised my score accordingly.

---

> > ### Author Response · Authors · 2023-11-22
> >
> > Thank you for your engagement and support! We greatly appreciate it.

---

### Official Review · Reviewer_4NxS · 2023-11-01

**Soundness:** 3 good
**Presentation:** 3 good
**Contribution:** 2 fair
**Rating:** 5
**Confidence:** 3

**Summary:**

The paper focuses on the challenge of vanishing gradients in the context of reinforcement finetuning (RFT) of pretrained language models. The authors identify that the expected gradient for an input tends to vanish when its reward standard deviation is small, regardless of the expected reward's optimality. This phenomenon can hinder the training process. The paper suggests a solution by introducing an initial supervised finetuning (SFT) phase where a small percentage of input samples may be needed.

**Strengths:**

1. The paper pinpoints a specific, nuanced problem in the domain of reinforcement learning finetuning of language models, i.e., vanishing gradients due to small reward standard deviation.
2. The proposed theory for small standard deviation leading to a small gradient norm is pretty intuitive, but the proof seems novel.

**Weaknesses:**

1. It is well-known that policy gradient could be suboptimal when the reward is pretty flat even though it's not the optimal solution. However, specific to the proposed scenario where the standard deviation is small, there could be many naive remedies not mentioned in the paper and it remains questionable whether the proposed problem could be easily solved by them. See  Q1.
2. If we use a learned reward model, whether SFT initialization is helpful should be highly related to the dataset that the reward model is trained on, and there seems to be no discussion on this part. See Q2.
3. To solve the suboptimal problem, it seems like designing a better reward function is more important. If there are many flat areas in the reward model, a different initialization might not necessarily help much.

**Questions:**

Q1: Consider simple reward scaling with gradient clipping, value function learning in [1], and reward normalization in [2]. Such tricks could help the model to escape the area when it is not exactly flat but the standard deviation is small since the exact value of the reward does not matter, but the relative one matters. I wonder whether such tricks could address the problem.

Q2. It could be easy to imagine that if the reward is trained on a dataset that is significantly better than the pretrained model, all outputs from the pretrained model will be bad and the reward std would be small. If we start fine-tuning with the SFT model, some of the outputs would be better after fine-tuning and the reward std would be larger. However, if the reward model is trained with human evaluation on the output of the pretrained model itself, it would be more likely that the reward std would be larger. I wonder whether the usefulness of SFT initialization is specific to some reward models trained on specific datasets.

[1] DPOK: Reinforcement Learning for Fine-tuning Text-to-Image Diffusion Models
Ying Fan, Olivia Watkins, Yuqing Du, Hao Liu, Moonkyung Ryu, Craig Boutilier, Pieter Abbeel, Mohammad Ghavamzadeh, Kangwook Lee, Kimin Lee

[2] Training Diffusion Models with Reinforcement Learning
Kevin Black, Michael Janner, Yilun Du, Ilya Kostrikov, Sergey Levine

---

> ### Author Response · Authors · 2023-11-13
> **Response (Part 1/2)**
>
> Thank you for your time and thoughtful feedback. We treat your comments and questions below. If our response is satisfactory, we would greatly appreciate it if you would consider increasing your score.
>
> > 1. …specific to the proposed scenario where the standard deviation is small, there could be many naive remedies not mentioned in the paper and it remains questionable whether the proposed problem could be easily solved by them. See Q1.
>
> > Q1: Consider simple reward scaling with gradient clipping, value function learning in [1], and reward normalization in [2]. Such tricks could help the model to escape the area when it is not exactly flat but the standard deviation is small since the exact value of the reward does not matter, but the relative one matters. I wonder whether such tricks could address the problem.
>
> Section 5.1 demonstrates that three common methods, which may seem suitable for combating vanishing gradients in RFT, are inadequate. Of course, our investigation of possible remedies is not exhaustive. One can come up with various other heuristics that may or may not help. Since our main contribution is the identification of the vanishing gradients problem and its detrimental effects (through a combination of theory and experiments), we believe further investigating ways to overcome it falls outside the scope of the current work, and poses a valuable direction for future research. We have updated the limitations paragraph in the conclusion section with a clarification per your remark (additional/modified text marked in red).
>
> Regarding the possible remedies raised in the review:
>
> - The PPO implementation we use, adopted from Ramamurthy et al. 2023, already includes learning a value function, as done in [1], and normalization of the reward, similar to [2] (in our implementation reward normalization is based on the rewards of inputs in a batch, while [2] normalize the rewards on a per-prompt basis). These heuristics are widely used in policy gradient implementations, yet as our experiments already show, they do not mitigate the difficulty of RFT to maximize the reward of inputs with small reward standard deviation.
>
> - To examine whether reward scaling with gradient clipping can aid in mitigating vanishing gradients in RFT, we ran preliminary experiments on the NarrativeQA dataset, where the rewards were scaled up by a factor of 10 or 100 (after normalization) and the maximum gradient norm was clipped to 0.5 to avoid instabilities. The results indicate that this method is inadequate as well. In particular, both reward scalings lead to lower or similar train and test rewards for RFT compared to the default configuration.
>
>     **Experimental Results:**
>     - No reward scaling (default configuration): train reward 0.107, test reward 0.114
>
>     - Reward scaling x10 with gradient clipping: train reward 0.102, test reward 0.110
>
>     - Reward scaling x100 with gradient clipping: train reward 0.108, test reward 0.111
>
>     Furthermore, we include in the following (anonymized) link a figure, analogous to Figure 1. It compares the reward means and standard deviations of training samples under the pretrained, RFT, and RFT with reward scaling x100 models. The figure demonstrates that simple reward scaling does not improve the ability of RFT to maximize the reward in the presence of inputs with small reward standard deviation.
>
>     Link: https://ibb.co/JvH8J5R
>
>     &nbsp;
>
>     Technical note: To obtain results in time for the discussion period, we reduced the overall number of optimization steps to 25% of the original number of steps. This is done for both the run without reward scaling and the runs with reward scaling. Since reward scaling led to lower or similar train and test rewards, it is likely that increasing the number of iterations will not result in a major improvement (if at all) over the default configuration with no reward scaling.

---

> > ### Author Response · Authors · 2023-11-13
> > **Response (Part 2/2)**
> >
> > > 2. If we use a learned reward model, whether SFT initialization is helpful should be highly related to the dataset that the reward model is trained on, and there seems to be no discussion on this part. See Q2.
> >
> > > Q2: It could be easy to imagine that if the reward is trained on a dataset that is significantly better than the pretrained model, all outputs from the pretrained model will be bad and the reward std would be small. If we start fine-tuning with the SFT model, some of the outputs would be better after fine-tuning and the reward std would be larger. However, if the reward model is trained with human evaluation on the output of the pretrained model itself, it would be more likely that the reward std would be larger. I wonder whether the usefulness of SFT initialization is specific to some reward models trained on specific datasets.
> >
> > When using a learned reward function, the data and objective used to train it may very well affect the severity of vanishing gradients in RFT and the benefits of an initial SFT phase. To facilitate efficient experimentation, we did not consider reward functions learned from human feedback (based on the outputs of the pretrained model). We believe an exciting next step is to study the extent to which our empirical findings carry over to more complex finetuning pipelines, including those with human feedback. Per your remark, we have clarified this point in the conclusion section (additional/modified text marked in red). Thank you for raising the matter and helping us improve our work!
> >
> > > 3. To solve the suboptimal problem, it seems like designing a better reward function is more important. If there are many flat areas in the reward model, a different initialization might not necessarily help much.
> >
> > This is an excellent point. While modifying the initialization through an SFT phase is a promising solution, we certainly do not believe that it is the only viable one (nor do we claim so in the paper). We believe further investigating ways to overcome vanishing gradients in RFT, e.g., by designing better reward functions, is an important direction for future work.

---

### Official Review · Reviewer_PKCD · 2023-11-01

**Soundness:** 4 excellent
**Presentation:** 4 excellent
**Contribution:** 4 excellent
**Rating:** 8
**Confidence:** 3

**Summary:**

In this paper, the authors identify an issue in the reinforcement finetuning phase of language models. Specifically, they first prove that the gradient norm of the value function is upper bounded by the standard deviation of the reward scores over the input sequences. They then empirically demonstrate on three datasets that small reward standard deviations are prevalent in the GRUE benchmark. Together, they suggest that the RLHF paradigm (i.e. preference reward + PPO) struggles to improve the quality on massive samples with small reward deviation. Finally, they suggest that simply using an initial supervised fine-tuning phase will greatly help mitigate the problem.

Overall, the motivation of this paper is very well justified and supported. In particular, they match the theory and practice quite well in a current fast-evolving paradigm. In addition, it provides a simple and effective solution to the discovered problem. I believe the paper could be potentially impactful in RLHF for language models.

My concerns are framed as questions below.

**Strengths:**

The motivation of this paper is very well justified and supported. In particular, they match the theory and practice quite well in a current fast-evolving paradigm. In addition, it provides a simple and effective solution to the discovered problem. I believe the paper could be potentially impactful in RLHF for language models.

**Weaknesses:**

My concerns are framed as questions below.

**Questions:**

1. Is there a potential explanation of why the reward model generates small standard deviations on the GRUE benchmark?
2. Could you list all the assumptions used in your proofs explicitly and discuss their realisticity?

---

> ### Author Response · Authors · 2023-11-13
>
> Thank you for the support, and specifically for emphasizing the potential impact of our work! We address your questions below. Please let us know if you have any further questions and we will happily expand.
>
> > 1. Is there a potential explanation of why the reward model generates small standard deviations on the GRUE benchmark?
>
> Certainly. As discussed in Sections 1 and 3, we expect inputs with small reward standard deviation (and low reward mean) to be common when the text distribution of the downstream task markedly differs from that of the pretraining corpora. In such a case, the pretrained model is likely to give non-negligible probability only to outputs of roughly the same suboptimal reward.
>
> Section 4.1 corroborates this expectation – see text under the “Inputs with small reward standard deviation are prevalent” header. Representative examples for inputs with small and inputs with large reward standard deviation under the pretrained model are given in Appendix E.
>
> > 2. Could you list all the assumptions used in your proofs explicitly and discuss their realisticity?
>
> The gradient norm bound in Theorem 1, and the analogous results for PPO variants (Propositions 1 and 2), are general and entail no assumptions, beyond requiring that the model $f$ is differentiable with respect to the parameters $\theta$ and that the softmax operator is used to produce next-token distributions (as stated in the preliminaries section). Both requirements are upheld by standard language models.

---

### Author Response · Authors · 2023-11-20

Dear reviewers,

We would like to thank you again for your effort and feedback. Since the discussion period is nearing its end, we kindly remind you to let us know whether we have fully addressed your questions. If there are any further points that you believe require clarification, we will gladly elaborate.

Authors

---

### Meta-Review · Area_Chair_QcRE · 2023-12-11

**Metareview:**

In this paper, the authors identify an issue in the reinforcement finetuning phase of language models. Specifically, they first prove that the gradient norm of the value function is upper bounded by the standard deviation of the reward scores over the input sequences. They then empirically demonstrate on three datasets that small reward standard deviations are prevalent in the GRUE benchmark. Together, they suggest that the RLHF paradigm (i.e. preference reward + PPO) struggles to improve the quality on massive samples with small reward deviation. Finally, they suggest that simply using an initial supervised fine-tuning phase will greatly help mitigate the problem.

There is consensus among the reviewers that this is a welcome contribution. I recommend acceptance as a poster.

Remark:
* In the appendix D.1., what does p(\mathcal{Y}_c | x) mean?

**Justification For Why Not Higher Score:**

* While insightful, Theorem 1 is relatively straightforward

**Justification For Why Not Lower Score:**

* The paper is well written
* Theorem 1 seems novel
* There is empirical evidence regarding small standard deviation of the reward

---

### Decision · Program_Chairs · 2024-01-16

Accept (poster)